# Expanding the cytokine receptor alphabet reprograms T cells into diverse states

Yang Zhao[1], Masato Ogishi[1], Aastha Pal[2], Leon L. Su[1], Pingdong Tao[1], Hua Jiang[1], Grayson E. Rodriguez[1], Xiaojing Chen[1], Qinli Sun[1], Lea Wenting Rysavy[2], Sam Limsuwannarot[2], Deepa Waghray[1], Anusha Kalbasi[2,3 ✉] & K. Christopher Garcia[1,4,5 ✉]

T cells respond to cytokines through receptor dimers that have been selected over the course of evolution to activate canonical JAK–STAT signalling and gene expression programs[1]. However, the potential combinatorial diversity of JAK–STAT receptor pairings can be expanded by exploring the untapped biology of alternative non-natural pairings. Here we exploited the common γ chain (γ_c) receptor as a shared signalling hub on T cells and enforced the expression of both natural and non-natural heterodimeric JAK–STAT receptor pairings using an orthogonal cytokine receptor platform[2–4] to expand the γ_c signalling code. We tested receptors from γ_c cytokines as well as interferon, IL-10 and homodimeric receptor families that do not normally pair with γ_c or are not naturally expressed on T cells. These receptors simulated their natural counterparts but also induced contextually unique transcriptional programs. This led to distinct T cell fates in tumours, including myeloid-like T cells with phagocytic capacity driven by orthogonal GSCFR (oGCSFR), and type 2 cytotoxic T (T_C2) and helper T (T_H2) cell differentiation driven by orthogonal IL-4R (o4R). T cells with orthogonal IL-22R (o22R) and oGCSFR, neither of which are natively expressed on T cells, exhibited stem-like and exhaustion-resistant transcriptional and chromatin landscapes, enhancing anti-tumour properties. Non-native receptor pairings and their resultant JAK–STAT signals open a path to diversifying T cell states beyond those induced by natural cytokines.

Cytokine receptors associated with the JAK-family kinases phosphorylate STAT-family transcription factors (TFs), which regulate specific target genes involved in haematopoiesis and cell differentiation. T cell responsiveness to cytokines is controlled by the expression of specific cytokine receptors that have presumably naturally evolved to support homeostatic functions. For example, IL-2, which signals through JAK1–JAK3 heterodimers and principally activates phosphorylated STAT5 (pSTAT5), has been the most widely studied cytokine for enhancing T cell functionality in adoptive cell therapy (ACT) of cancer[5,6]; however, its use is limited by toxicities and suboptimal efficacy[7,8]. Alternative cytokine signals may better equip T cells for ACT[9,10]. In fact, there exist approximately 35 different JAK–STAT cytokine receptors that elicit a spectrum of functions, yet only a few have been explored as alternatives to IL-2. Beyond their anti-tumour roles, cytokines impact regulatory nodes in cell specification[11], but their ability to drive cell identities remains underexplored. The ability of JAK–TYK kinases and STATs to engage in cross-talk downstream of different cytokine receptor chains opens the possibility of exploring this question by compelling new signalling outputs through enforced dimerization of non-canonical cytokine receptor pairings that do not encode natural cytokine ligands[12,13], effectively expanding the cytokine receptor pairing code.

Common γ_c acts as a JAK3-associated signalling hub for pairing with JAK1-associated private receptors (for example, IL-2Rβ, IL-4Rα, IL-7Rα, IL-9Rα and IL-21Rα) to induce a range of STATs that orchestrate survival, proliferation and differentiation across immune cell lineages[14]. Interferons (IFNs) predominantly activate STAT1 for immunomodulation and antiviral defence[15]. IL-10-family cytokines primarily trigger STAT3 and STAT1 activation and maintain tissue homeostasis[16]. GCSFR, which is expressed on neutrophils and their precursors, forms a homodimer and primarily activates STAT3 to regulate neutrophil function[17]. EPOR homodimers activate STAT5 signalling crucial for maintaining and differentiating erythroid progenitor cells[18]. However, the intrinsic effects of each cytokine family on T cells are obscured by both the pleiotropy of each cytokine in vivo and the selective expression of cytokine receptors in T cells.

Previously, we developed an orthogonal cytokine–receptor engineering platform to selectively and tunably activate cytokine receptor pathways on T cells that signal through the shared receptor γ_c[2–4] that is endogenously expressed (Fig. 1a). Here we extend this approach by swapping the orthogonal IL-2Rβ receptor intracellular domain (ICD) with the ICD of other cytokine receptors. This enabled us to sample a broad spectrum of JAK–STAT signalling outputs through both natural and non-natural γ_c heterodimeric pairings, including receptors for the

[1]Department of Molecular and Cellular Physiology, Stanford University School of Medicine, Stanford, CA, USA. [2]Department of Radiation Oncology, Stanford University School of Medicine, Stanford, CA, USA. [3]Stanford Cancer Institute, Stanford University School of Medicine, Stanford, CA, USA. [4]Howard Hughes Medical Institute, Stanford University School of Medicine, Stanford, CA, USA. [5]Department of Structural Biology, Stanford University School of Medicine, Stanford, CA, USA. ✉e-mail: akalbasi@stanford.edu; kcgarcia@stanford.edu

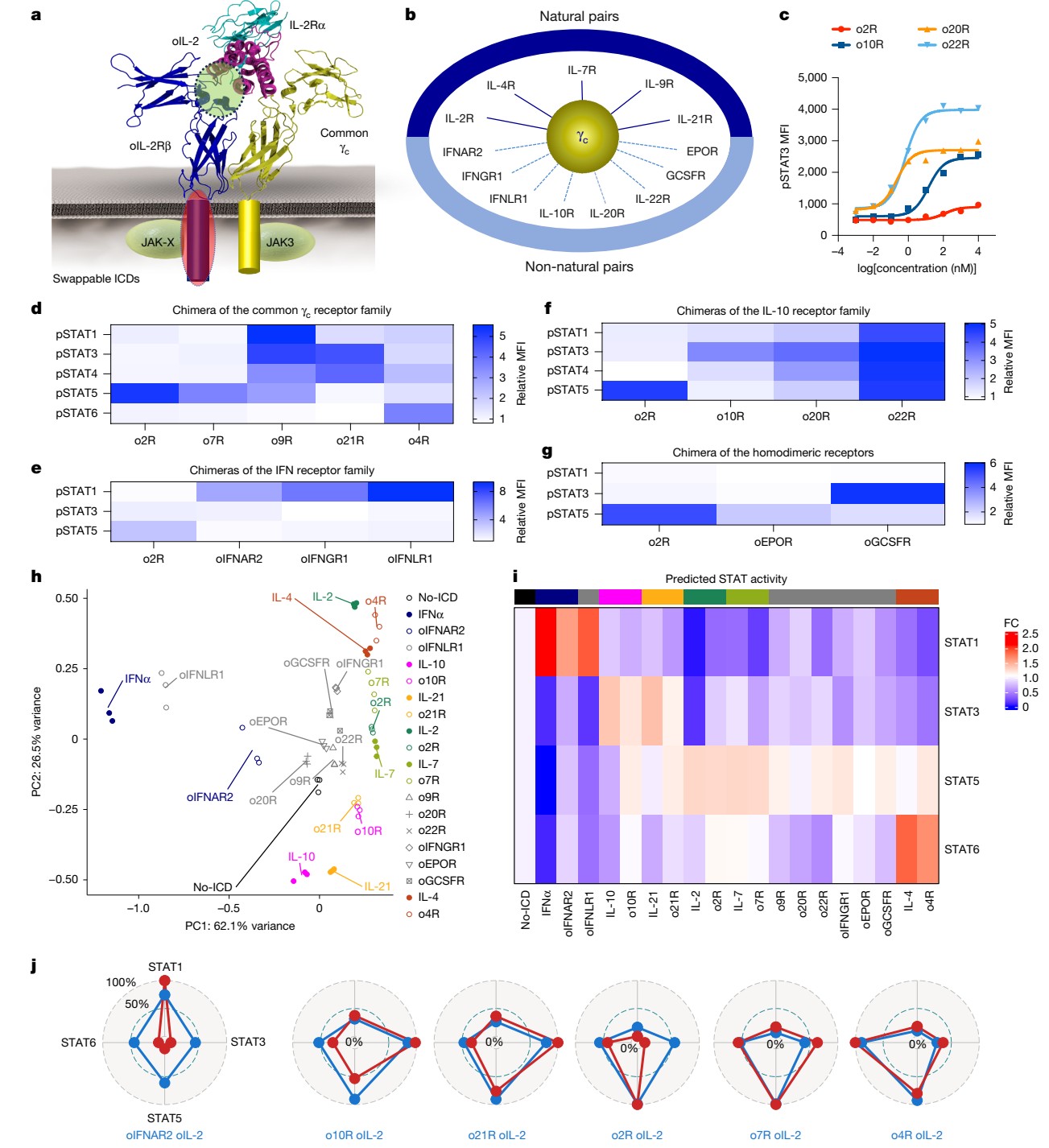

**Fig. 1 | Signal activation and gene expression profiles induced by orthogonal chimeric receptors. a**, Schematic of orthogonal IL-2–IL-2Rβ pairs, whereby a mutant IL-2 interacts with a mutant IL-2Rβ that pairs with the common γc to trigger swappable ICD signalling. **b**, Natural or non-natural ICDs that pair with the γc. **c**–**g**, Orthogonal-receptor-expressing (YFP[+]) C57BL/6 WT CD3[+] T cells were stimulated with MSA fusions of orthogonal mouse IL-2 (MSA–oIL-2) for 20 min. Cells were analysed for pSTAT. **c**, pSTAT3 signalling dose–response curves of o2R-, o10R-, o20R- and o22R-transduced WT T cells, plotted against the log10-transformed concentration of MSA–oIL-2. **d**, The relative pSTAT1, pSTAT3, pSTAT4, pSTAT5 and pSTAT6 mean fluorescence intensity (MFI) in T cells transduced with o2R, o4R, o7R, o9R and o21R treated with MSA–oIL-2 (10 μM), normalized to the YFP[−] controls. **e**, The relative pSTAT1, pSTAT3 and pSTAT5 MFI in T cells transduced with o2R, oIFNAR2, oIFNGR1 and oIFNLR1,

treated with MSA–oIL-2 (10 μM), normalized to the YFP[−] controls. **f**, The relative pSTAT1, pSTAT3, pSTAT4 and pSTAT5 MFI in T cells transduced with o2R, o10R, o20R and o22R and treated with MSA–oIL-2 (10 μM), normalized to the YFP[−] controls. **g**, The relative pSTAT1, pSTAT3 and pSTAT5 MFI in T cells transduced with o2R, oEPOR and oGCSFR and treated with MSA–oIL-2 (10 μM), normalized to the YFP[−] controls. **h**–**j**, C57BL/6 WT T cells transduced with orthogonal receptors were stimulated with MSA–oIL-2 (5 μM) or recombinant cytokines (10 nM) for 6 h, and then analysed using RNA-seq. *n* = 3 biologically independent samples. **h**, PCA of RNA-seq data. **i**, The expression of STAT-driven gene signatures. FC, fold change compared with the no-ICD controls. **j**, Comparison of the STAT gene signatures of orthogonal chimera-transduced T cells treated with MSA–oIL-2 versus those treated with the indicated native cytokines. Data are mean ± s.e.m.

IFN family, IL-10 family and homodimeric receptors GCSFR and EPOR. We find that orthogonal chimeric receptors containing ICDs for IL-4R (o4R), IL-20R (o20R), IL-22R (o22R) and GCSFR (oGCSFR) reprogram T cells to promote distinct natural and synthetic cell states with functional and therapeutic implications. Chimeric antigen receptor (CAR) or T cell receptor (TCR)-engineered T cells incorporating o22R show vastly enhanced stemness properties, and oGCSFR, a myeloid-specific receptor, induces a myeloid-like state and endows T cells with phagocytic properties. Both o22R and oGCSFR show superior efficacy across haematological and solid-tumour models, linked to epigenetic resistance to exhaustion. This approach expands the $\gamma_c$ family to non-natural JAK–STAT receptor pairings, lays the foundation for cell state manipulation through cytokine receptors and addresses limitations of engineered cell therapies that are not addressed by our naturally encoded repertoire of cytokines.

## Signal profiles of orthogonal receptors

We investigated whether natural and non-natural cytokine receptor pairings with $\gamma_c$ can form functional heterodimers to activate signalling in primary T cells. Using an orthogonal platform with mutant IL-2 (oIL-2) and IL-2Rβ (oIL-2Rβ) extracellular domain (ECD) that selectively bind to each other but not to their wild-type (WT) counterparts[2], we engineered mouse WT T cells with oIL-2Rβ or chimeric forms of oIL-2Rβ in which its transmembrane domain and ICD were replaced with those from other cytokine receptors—effectively extending the scope of $\gamma_c$ as a signalling hub (Fig. 1a). Binding of oIL-2 ligand induces heterodimerization of the chimeric oIL-2Rβ and endogenous $\gamma_c$[2], enforcing optimal receptor–JAK geometry for signalling[19] (Fig. 1a). We tested natural $\gamma_c$ parings (IL-4R, IL-7R, IL-9R, IL-21R)[4] as a benchmark for enforced non-natural pairings with JAK1-associated receptors (IFNAR2, IFNGR1, IFNLR1, IL-10R1, IL-20R, IL-22R), as well as homodimeric receptors GCSFR and EPOR (Fig. 1b and Extended Data Fig. 1a). The EPOR–$\gamma_c$ forms a JAK2–JAK3 pairing that is not found in nature.

Mouse-serum-albumin-linked oIL-2 (MSA–oIL-2) clone 3A10, selected for its strict orthogonality[2], induced dose-dependent pSTAT signalling in mouse T cells expressing orthogonal chimeric receptors (Fig. 1c and Extended Data Fig. 1b–e), confirming ligand-induced heterodimerization and activation of JAK1/2 (primary chain) and JAK3 ($\gamma_c$). Each $\gamma_c$ chimera showed expected canonical responses, with o9R and o21R also activating STAT4[20,21] (Fig. 1d and Extended Data Fig. 1b). IFN receptor chimeras predominantly triggered pSTAT1, with the most potent activation by oIFNLR1[22] (Fig. 1e and Extended Data Fig. 1c). o10R signalled mainly through STAT3 (Fig. 1c,f and Extended Data Fig. 1d). Notably, o22R broadly activated STAT1, STAT3, STAT4 and STAT5, while o20R induced dominant pSTAT3 with moderate pSTAT1, pSTAT4 and pSTAT5 (Fig. 1c,f and Extended Data Fig. 1d). Like IFNLR1, IL-20R and IL-22R are normally expressed on epithelial cells and not T cells[23]. oGCSFR induced pSTAT3 and moderate pSTAT1 and pSTAT5, while oEPOR showed weak STAT5 activation (Fig. 1g and Extended Data Fig. 1e). GCSFR and EPOR are not typically expressed in T cells, rather, they are expressed in myeloid and red blood cells, respectively. Cells did not induce pSTAT signalling in a $\gamma_c$-knockout (KO) cell line or when stimulated with $\gamma_c$-interface-mutant oIL-2 (Extended Data Fig. 2a–g). Furthermore, the rapid and robust orthogonal receptor-induced STAT signalling could not be generated by appending ICDs or motifs to a CAR terminus (Extended Data Fig. 2h). Thus, ligand-enforced natural and non-natural $\gamma_c$ heterodimers are critical for effective JAK–STAT signalling through chimeric orthogonal receptors.

We performed RNA-sequencing (RNA-seq) analysis to profile transcriptional responses to orthogonal receptors. Principal component analysis (PCA) showed distinct clustering, with chimeras sharing STAT profiles grouping near native cytokine-treated T cells: oIFNLR1 and oIFNAR2 grouped with IFNα-treated T cells, and STAT3 cytokines (IL-21 and IL-10) clustered closely with their orthogonal pairs (Fig. 1h).

STAT-driven gene expression mirrored known patterns: oIFNLR1 induced STAT1-driven genes; o10R and o21R, and less potently o22R, activated STAT3-driven genes; while o4R induced STAT6-driven genes (Fig. 1i). oIFNGR1 showed weak transcriptional output, probably due to suboptimal capture timing (Extended Data Fig. 2i). Most of the $\gamma_c$ chimeras triggered STAT5-driven genes at varying levels (Fig. 1i). Orthogonal chimeras resembled native cytokine-induced signatures but with attenuated strength, indicating partial agonism (Fig. 1i,j). Non-native pairings of $\gamma_c$ with JAK1-associated receptors (for example, IFNAR2 and IL-10R1) largely preserved the signalling fidelity of the natural heterodimer, but showed broader STAT activation when JAK3 replaced TYK2 from the natural heterodimer pair (IFNAR1 and IL-10R2; Fig. 1j). In summary, orthogonal receptors recapitulate native STAT programs while diversifying cellular responses by allowing $\gamma_c$-dependent activation of non-$\gamma_c$ receptors.

## Synthetic signals in T cell therapy

To assess how distinct STAT profiles affect T cell function, we used ACT in B16F10 melanoma. Thy1.1[+] pmel-1 CD8[+] T cells expressing orthogonal chimeras were transferred into lymphodepleted mice, followed by MSA–oIL-2 treatment (Fig. 2a). All treatments were well tolerated (Extended Data Fig. 3a,b). Notably, o4R (pSTAT6) conferred superior anti-tumour efficacy among $\gamma_c$ chimeras, with 1 out of 8 mice achieving complete remission (CR) (Fig. 2b,c and Extended Data Fig. 3c). IFN receptor chimeras also enhanced tumour control with oIFNLR1 yielding a curative response in 1 out of 8 mice (Fig. 2d,e and Extended Data Fig. 3c). IL-10 receptor family chimeras were highly effective, particularly o22R, which led to CR in 3 out of 9 mice (Fig. 2f,g and Extended Data Fig. 3c). Homodimeric receptors (oEPOR and oGCSFR) also controlled tumours; oGCSFR induced CR in 2 out of 8 mice (Fig. 2h,i and Extended Data Fig. 3c). These effects required oIL-2, enabling tunable in vivo signalling (Extended Data Fig. 3d).

We observed no association between in vitro proliferation driven by the orthogonal receptors and in vivo anti-tumour activity (Fig. 2 and Extended Data Fig. 3e). For example, o22R did not promote proliferation in vitro, but we noted significant anti-tumour effects (Fig. 2f,g). Notably, o22R T cells expanded substantially in the blood, spleen, tumour-draining lymph nodes (TDLNs) and tumour (Extended Data Fig. 4a–e), with increased BrdU incorporation and Ki-67 expression in tumours (Extended Data Fig. 4f,g), indicating an antigen-driven process in vivo. o20R and o22R increased intratumoural CD45.2[+] cells, suggesting enhanced endogenous immune cell infiltration (Extended Data Fig. 4h). Other effective chimeras also showed robust peripheral expansion (Extended Data Fig. 4b–e). Although o4R T cells were less prevalent in tumours, their abundance in peripheral organs may support tumour control (Extended Data Fig. 4b–e). Importantly, none of the receptors caused splenomegaly (Extended Data Fig. 4i). Together, orthogonal receptors variably influenced T cell expansion and tumour infiltration, leading to improved therapeutic outcomes.

## Divergent cell fates through synthetic receptors

To examine how synthetic signalling shapes the functionality of CD8[+] tumour-infiltrating lymphocytes (TILs), we performed single-cell RNA-seq (scRNA-seq) analysis of 29,008 CD8[+] TILs from four conditions with potent anti-tumour efficacy: o4R, o20R, o22R and oGCSFR pmel-1 T cells, and non-transduced (NT) control T cells (Fig. 3a,b). Among live single cells, the CD8[+] TIL frequency was highest for o22R (4.62%), followed by oGCSFR (2.03%), o20R (1.67%), o4R (0.7%) and NT (0.45%) T cells (Fig. 3c). TF-activity scores predicted by SCENIC were used for uniform manifold approximation and projection (UMAP) visualization (Extended Data Fig. 5a). Unsupervised clustering revealed seven transcriptionally distinct clusters (Fig. 3d). NT cells were primarily assigned to cluster 0 (C0) (Fig. 3e,f), which, along with C4, showed

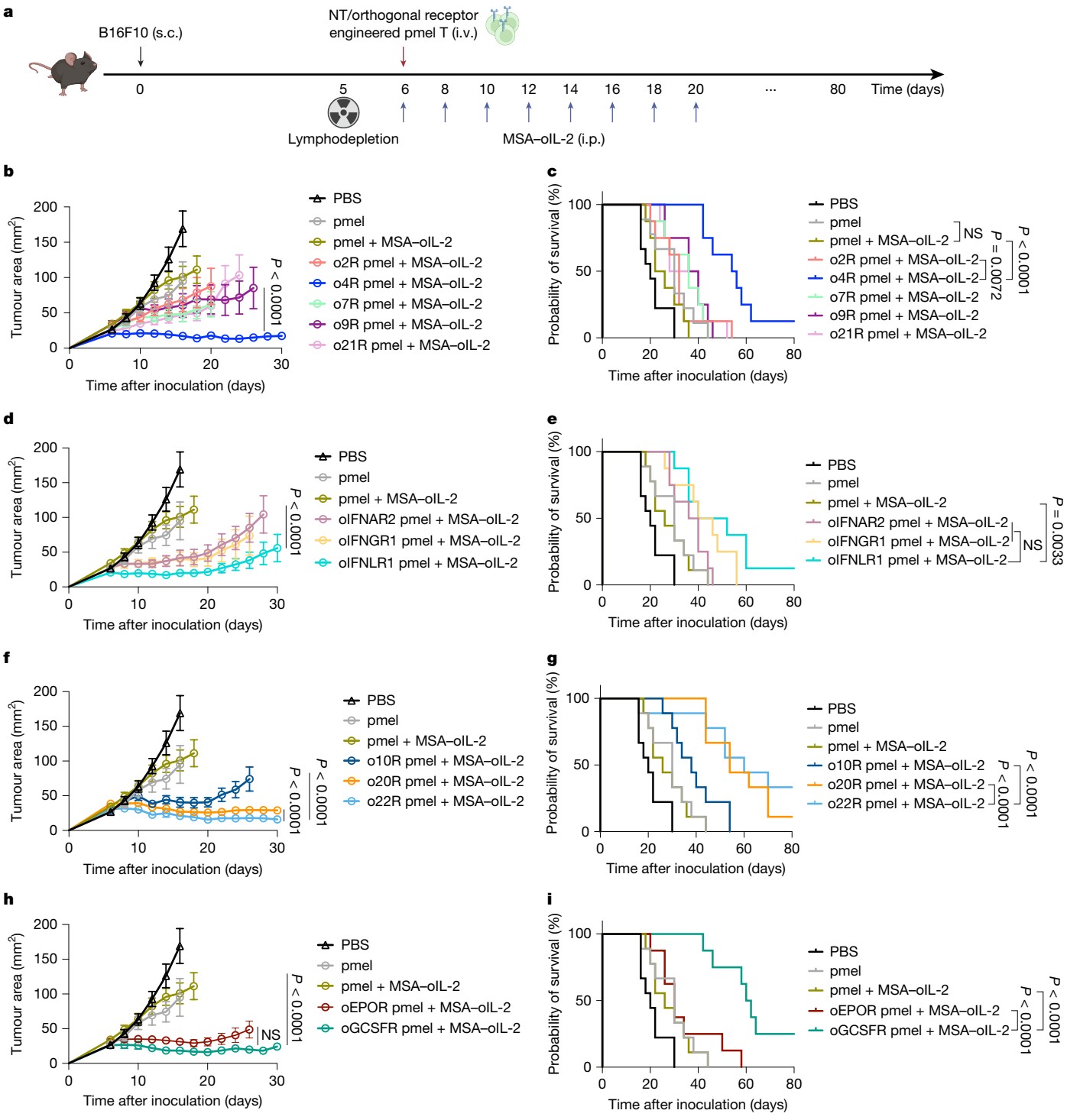

**Fig. 2 | Synthetic signalling drives enhanced anti-tumour efficacy of ACT immunotherapy. a–i**, Female C57BL/6 WT mice bearing established subcutaneous (s.c.) B16F10 tumours were sublethally lymphodepleted by total body irradiation (5 Gy) on day 5. On day 6, the mice received intravenous (i.v.) ACT of untransduced (NT) pmel-1 (pmel) T cells ($3 \times 10^6$ cells) or pmel-1 T cells transduced with the indicated orthogonal chimeric receptor ($\gamma_c$ receptor (**b,c**), IFN receptor (**d,e**), IL-10R (**f,g**) and homodimeric receptor (**h,i**); $3 \times 10^6$ cells) followed by intraperitoneal (i.p.) administration of MSA–oIL-2 ($2.5 \times 10^4$ U per day)

or PBS every other day until day 20. *n* = 8 (o4R, o7R, o21R, oIFNAR2, oIFNGR1, oIFNLR1, oEPOR and oGCSFR) and *n* = 9 (other groups) mice. **a**, The experimental timeline. **b–i**, The average tumour growth curves (**b**, **d**, **f** and **h**) and survival curves (**c**, **e**, **g** and **i**) of each treatment group. Data are mean ± s.e.m. Statistical analysis was performed using two-way analysis of variance (ANOVA) with Tukey's post-test (**b**, **d**, **f** and **h**) or log-rank (Mantel–Cox) tests (**c**, **e**, **g** and **i**). NS, not significant. The diagram in **a** was created using BioRender. Zhao, Y. (2025) https://BioRender.com/t1qoa7d.

high *Tox* expression, indicating an exhausted phenotype (Extended Data Fig. 5b–d). Multiple clusters were enriched for genes associated with cytotoxicity, with *Ifng* and *Prf1* enriched in C1, and *Gzma* and *Gzmb* in C2 (Extended Data Fig. 5e,f). Each orthogonal chimera resulted in

a distinct clustering profile (Fig. 3e,f), and all four exhibited elevated expression of cytotoxicity genes (Extended Data Fig. 5g). Although o22R produced moderate levels of cytotoxicity transcripts, flow cytometry revealed high densities of intratumoural IFNγ⁺TNF⁺ and

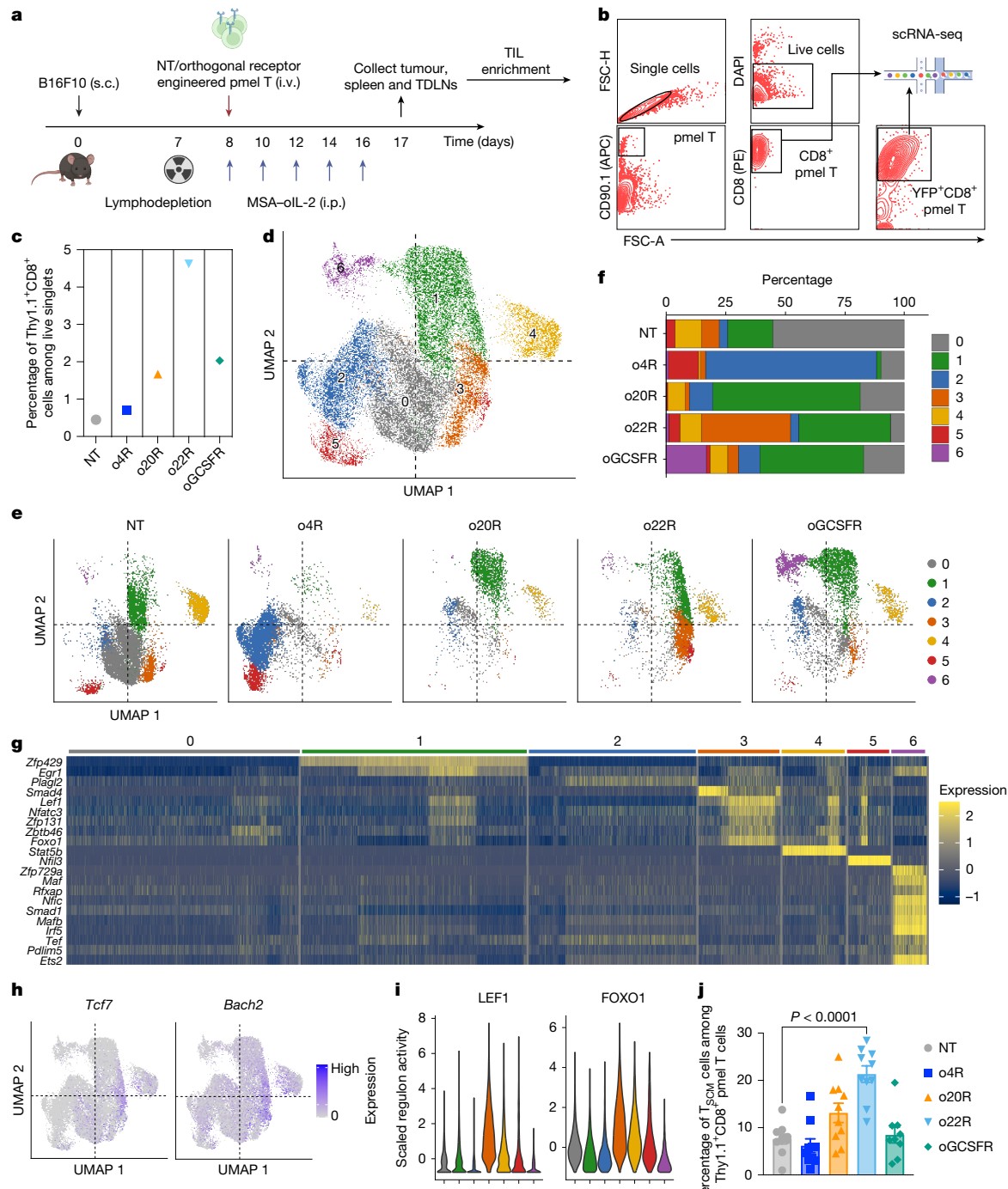

**Fig. 3 | Divergent T cell fates induced by synthetic cytokine receptors.**
**a–j**, Female C57BL/6 WT mice bearing established s.c. B16F10 tumours were lymphodepleted on day 7. On day 8, the mice received an i.v. adoptive transfer of 3 × 10⁶ pmel-1 T cells (NT or transduced with the indicated orthogonal chimeric receptor) followed by i.p. administration of MSA–oIL-2 (2.5 × 10⁴ U per day) or PBS every other day until day 20. On day 17, the mice were euthanized, and tumours, TDLNs and spleens were collected for flow cytometry analysis. For scRNA-seq analysis, Thy1.1⁺CD8⁺ TILs from ACT with NT pmel-1 T cells, and Thy1.1⁺CD8⁺YFP⁺ TILs from ACT with o4R, o20R, o22R and oGSCFR pmel-1 T cells were sorted for scRNA-seq analysis. **a**, The experimental timeline. **b**, The gating strategy for TILs used in scRNA-seq analysis. **c**, The frequencies of

Thy1.1⁺CD8⁺ cells for the live single-cell gate. *n* = 1 biologically independent sample. **d**, UMAP representation of NT pmel-1 T cells and those transduced with o4R, o20R, o22R and oGSCFR. **e**, UMAPs separated by the transduction conditions. **f**, The proportion of clusters in each transduction condition. **g**, The predicted activities of marker TFs. **h**, The expression of *Tcf7* and *Bach2* mRNA. **i**, The predicted activity of LEF1 and FOXO1. **j**, The frequencies of CD44⁻CD62L⁺ SCA1⁺ T_SCM cells among Thy1.1⁺CD8⁺ pmel-1 T cells in TDLNs. *n* = 8 (oEPOR and o9R), *n* = 9 (oGCSFR) and *n* = 10 (other groups) mice. Data are mean ± s.e.m. Statistical analysis was performed using one-way ANOVA with Tukey's post-test (**j**). The diagram in **a** was created using BioRender. Zhao, Y. (2025) https://BioRender.com/t2vr9gg.

GZMB⁺ cells, consistent with superior anti-tumour activity of o22R (Extended Data Fig. 5h–j). To examine molecular regulation, we identified marker TFs in each cluster (Fig. 3g). C3 showed high activity of

memory- and stemness-associated TFs (*Tcf7*, *Bach2*, *Lef1* and *Foxo1*)²⁴⁻²⁸ (Fig. 3g–i). o22R TILs were uniquely enriched in C3 (Fig. 3d,e), correlating with a 2.8-fold increase in canonical stem-cell-like memory

T ($T_{SCM}$; CD44−CD62L+SCA1+) cells in TDLNs compared with NT cells (Fig. 3j). Moreover, splenic o22R cells were enriched for IL-7Rα+KLRG1− memory precursors (Extended Data Fig. 5k). These findings suggest that synthetic cytokine signals shape diverse T cell profiles and drive functional heterogeneity in tumours.

## oGCSFR drives T cell to myeloid lineage

C6, with elevated activity of myeloid-lineage TFs such as NFIC, MAF and MAFB, was almost exclusively occupied by oGCSFR-transduced T cells (Figs. 3d–g and 4a). To examine this further, we projected our single-cell data onto a reference TIL atlas[29]. In contrast to other groups, a large proportion of the oGCSFR group did not efficiently map to known T cell populations (Extended Data Fig. 6a). To investigate myeloid lineage gene expression, we performed gene set enrichment analysis of oGCSFR differentially expressed genes. Among the 271 enriched gene sets (the top 20 are shown in Extended Data Fig. 6b), at least 89 (33%) were related to innate immunity or myeloid cell processes such as phagocytosis, neutrophil chemotaxis and myeloid leukocyte differentiation. In the SCENIC gene-regulatory network, MAFB was predicted to target another TF, C/EBPb, which in turn was predicted to drive multiple myeloid genes (*Tyrobp*, *Cd14* and *Plaur*) (Fig. 4b). A separate agnostic search for differentially upregulated genes identified several myeloid-lineage genes in C6 (Fig. 4c,d). Furthermore, C6 cells express diverse ligands known to interact with receptors expressed on conventional CD8+ T cells, consistent with the function of myeloid cells as an interacting partner of T cells (Fig. 4e,f).

Inspired by these observations, we examined whether oGCSFR can reprogram T cells into a myeloid-cell-like state in vitro. After 48 h in MSA−oIL-2, oGCSFR T cells showed markedly high expression of myeloid-lineage markers, including Mac-1 (encoded by *Itgam*; also known as CD11b), the pathogen recognition receptor CD14 and the Fc-gamma receptor 1 (FcγRI) CD64 (Fig. 4g–j), while also retaining T cell markers. About 39% expressed neutrophil-associated marker Gr-1, indicating a myeloid−neutrophil-like phenotype (Fig. 4k). For comparison, we overexpressed three highly activated TFs (NFIC, MAF and MAFB) in pmel-1 CD8+ T cells. Indeed, overexpression of a single TF or the trio in pmel-1 CD8+ T cells partially induced myeloid features (Fig. 4g–k and Extended Data Fig. 6c–f), albeit less substantially than oGCSFR, suggesting that the coordinated gene regulation orchestrated by oGCSFR is required for prominent T−myeloid transdifferentiation.

We next investigated whether receptors induced by oGCSFR in T cells function similarly to those in myeloid cells. Most CD8+Mac-1+ cells co-express CD14 (data not shown), a receptor that is essential for phagocytosis and bacterial lipopolysaccharide recognition[30]. To assess CD14 function, we sorted CD8+Mac-1− and CD8+Mac-1+ cells and performed a phagocytosis assay with *Escherichia coli*, in which J774A.1 macrophages demonstrate robust phagocytic activity. CD8+Mac-1+ T cells exhibited elevated phagocytic activity compared with CD8+Mac-1− and NT T cells, sustaining *E. coli* signals over time (Extended Data Fig. 6g,h). These findings were corroborated using live bacteria. Under high bacterial load, oGCSFR T cells demonstrated markedly enhanced phagocytic activity compared with NT T cells, efficiently internalizing GFP+ *Listeria monocytogenes* (Fig. 4l,m and Extended Data Fig. 6i). We further investigated whether oGCSFR-induced Fc receptors mediate antibody-dependent cellular phagocytosis (ADCP). FarRed-labelled anti-CD19-coated A20 cells were internalized by RAW264.7 macrophages, but showed minimal uptake in NT or o2R T cells lacking an ICD (Fig. 4n,o). By contrast, oGCSFR T cells demonstrated a marked increase in FarRed+ cells, resulting from both Fc-receptor dependent membrane capture and tumour component internalization (Fig. 4n,o and Extended Data Fig. 6j,k). oGCSFR T cells increased B16F10-CD19 cell killing from 57.8% to 70.8%

with CD19 antibodies, suggesting potential synergy with antibody therapies despite dominant direct cytotoxicity (Extended Data Fig. 6l). Furthermore, oGCSFR upregulated SIRPα expression in CD8+ T cells (Fig. 4p). Together, these results suggest that oGCSFR reprograms activated T cells into a synthetic myeloid-like state, serving as an effective lineage reprogramming tool compared with TFs.

## Robust $T_H2$/$T_C2$ cell induction through o4R signalling

While other orthogonal chimeras distributed across multiple clusters, o4R TILs concentrated almost exclusively in C2 (Fig. 3d–f). Differential gene expression analysis of C2 confirmed strong $T_C2$ cell signatures, including *Penk*, *Trat1*, *Lgals7*, *Csf2* and *Cyp11a1*, and key cytokine genes *Il4*, *Il5*, *Il13* and *Il10* (Fig. 5a,b), suggesting that o4R drives selective IL-4-dependent reprogramming in vivo.

A recent study linked enhanced type 2 function in CAR T cells to long-term responses in patients[31], but strategies to induce and maintain type 2 T cells in vivo are lacking. We engineered human T cells with human orthogonal chimeric IL-2Rβ–IL-4R (ho4R) and an NY-ESO-1-specific TCR (clone 1G4, HLA*0201) (Extended Data Fig. 7a). NT or ho4R NY-ESO-1 TCR-T cells were exposed to MSA-bound human orthogonal IL-2 (MSA−hoIL-2, containing SQVLKA substitutions at positions 15, 16, 19, 20, 22 and 23)[3] or IL-2. By day 14, ho4R significantly increased $T_H2$ cells, with increased IL-4+ (12.1% versus 5.4%), IL-5+ (39.4% versus 14.2%) and IL-13+ (25.7% versus 13.4%) CD4+ cells (Fig. 5c,d), and $T_C2$ cells marked by elevated IL-4+ (4.0% versus 2.8%) and IL-5+ (21.0% versus 11.6%) CD8+ cells (Fig. 5c,e). ho4R signalling did not restrain IFNγ production, and a subset of $T_H2$ or $T_C2$ ($T_H2$/$T_C2$) cells exhibited a hybrid phenotype, co-producing IFNγ with IL-4, IL-5 or IL-13 (Fig. 5f,g and Extended Data Fig. 7b). ho4R cells also showed increased GATA3 and CCR4 expression, while CXCR3 remain unchanged (Fig. 5h and Extended Data Fig. 7c). In NSG mice bearing HLA*0201+NY-ESO-1+ A375 tumours, ACT of ho4R NY-ESO-1 TCR-T cells along with MSA−hoIL-2 administration led to enhanced tumour suppression, whereas control TCR-T cells had minimal effect (Fig. 5i,j). Importantly, ho4R signalling strongly induced type 2 cells in vivo, with substantially increased IL-4+ (86.1% versus 25.8%), IL-5+ (62.8% versus 17.8%) and IL-13+ (9.2% versus 0.6%) cells (Fig. 5k and Extended Data Fig. 7d) without causing weight loss (Extended Data Fig. 7e). Blocking single or multiple type 2 cytokines in vitro did not impair the anti-tumour effect (Extended Data Fig. 7g), but *GATA3* KO completely abolished the anti-tumour activity mediated by ho4R (Extended Data Fig. 7f,g). These findings highlight synthetic IL-4R signalling as a potent approach to generate $T_H2$/$T_C2$ cells during cell manufacturing or in vivo with orthogonal IL-2.

## ho22R/hoGCSFR epigenetically rewire TILs

In a syngeneic melanoma model, mouse T cells equipped with o22R and oGCSFR demonstrated superior efficacy, leading us to examine their potential with human T cells in xenograft settings. We engineered human CD19 CAR T cells to express ho22R and hoGCSFR (Extended Data Fig. 8a). MSA−hoIL-2 activated STAT1, STAT3, STAT4 and STAT5 in ho22R, and mainly STAT3 in hoGCSFR cells (Extended Data Fig. 8b), consistent with the mouse receptors. Both ho22R and hoGCSFR enhanced tumour lysis at low effector-to-target (E:T) ratios when co-cultured with CD19+ NALM6 or Raji cells (Extended Data Fig. 8c,d), probably attributed to improved stemness (Extended Data Fig. 8e–g). Like its mouse counterpart, hoGCSFR also induced a myeloid-like phenotype in human T cells (Extended Data Fig. 8h–j). In a NALM6-luciferase xenograft model, CD19 CAR T cells expressing ho22R and hoGCSFR showed reduced leukaemic burden (Fig. 6a,b and Extended Data Fig. 9a). ho22R CAR T cells demonstrated the most durable control (Fig. 6b and Extended Data Fig. 9a), with a 72.6-fold increase in engraftment over control CAR T cells (Extended Data Fig. 9b). Against the Raji lymphoma model,

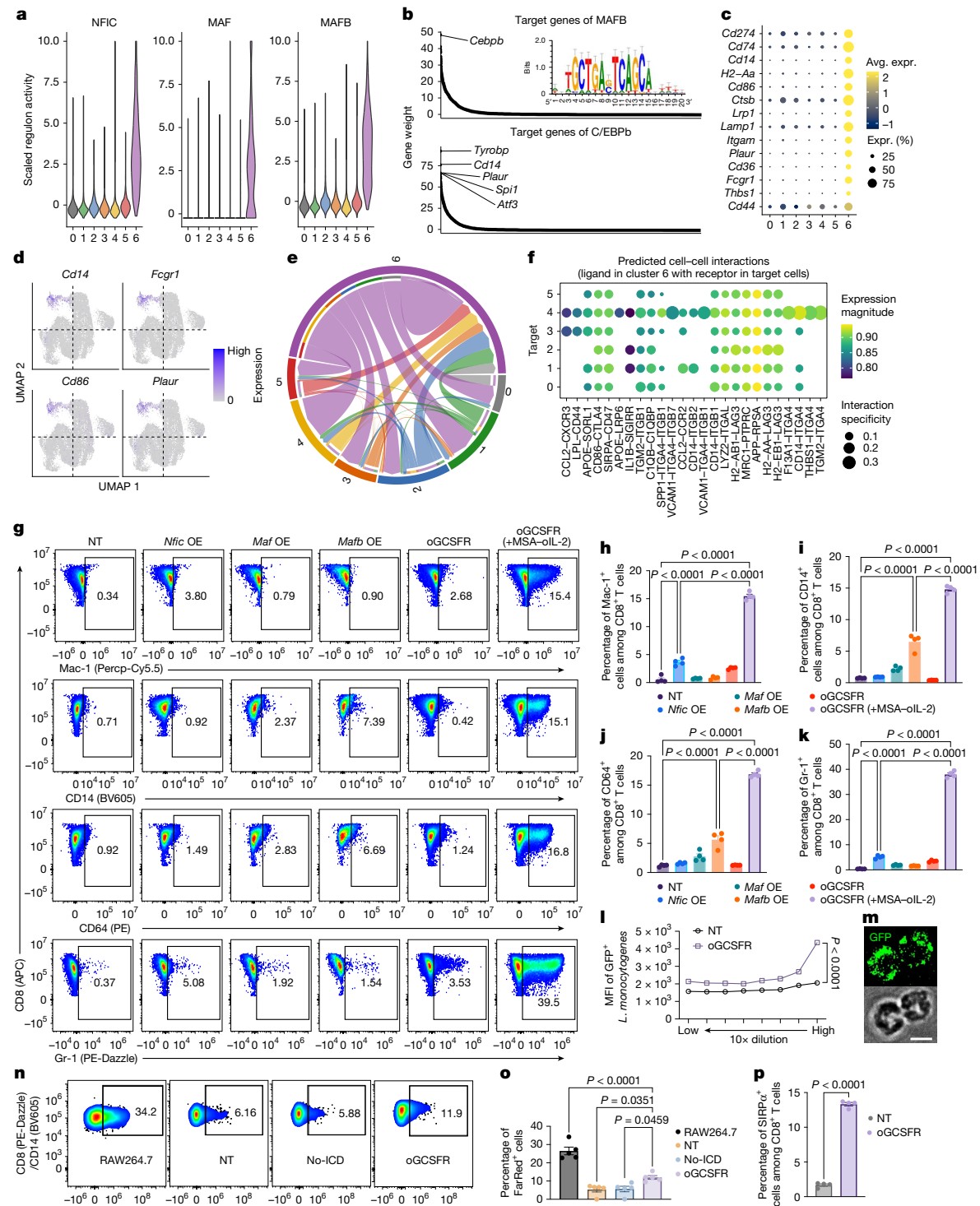

**Fig. 4 | oGCSFR expression endows CD8⁺ T cells with myeloid cell features.**
**a**, Mouse TIL scRNA-seq analysis (Fig. 3), showing the predicted activity of NFIC, MAF and MAFB. **b**, The top-ranked genes targeted by MAFB (top) and C/EBPb (bottom). **c**, Average expression (Avg. expr.) of myeloid-cell-associated surface-marker-encoding genes that are differentially upregulated in C6 cells. **d**, Expression of *Cd14*, *Fcgr1*, *Cd86* and *Plaur* mRNA. **e**, Prediction of intercellular ligand–receptor interactions between T cells in each cluster. **f**, Predicted interactions between ligands expressed in C6 cells and receptors expressed in cells in the other clusters. **g**–**k**, NT pmel-1 CD8⁺ T cells and those transduced with *Nfic*, *Maf*, *Mafb* or oGCSFR were cultured for 48 h, with oGCSFR pmel-1 CD8⁺ T cells also stimulated with MSA–oIL-2 (500 nM) during this period. *n* = 4 biologically independent samples. **g**, Representative flow cytometry plots showing the frequencies of Mac-1⁺, CD14⁺, CD64⁺ and Gr-1⁺ cells among pmel-1 CD8⁺ T cells. **h**–**k**, The frequencies of Mac-1⁺ (**h**), CD14⁺ (**i**), CD64⁺ (**j**) and Gr-1⁺ (**k**)

cells among pmel-1 CD8⁺ T cells. **l**, GFP⁺ *L. monocytogenes* was co-cultured with NT or oGCSFR pmel-1 CD8⁺ T cells in serial tenfold dilutions. The MFI of GFP⁺ *L. monocytogenes* is shown. *n* = 4 biologically independent samples. **m**, Representative fluorescence (top) and bright-field (bottom) images of GFP⁺ *L. monocytogenes* internalized by oGCSFR CD8⁺ T cells. Scale bar, 5 µm. **n**,**o**, FarRed-labelled A20 cells were pretreated with anti-mouse CD19 antibodies. Effector cells were co-cultured with the A20 cells for 2 h, then analysed using flow cytometry. *n* = 5 biologically independent samples. Representative flow cytometry plots (**n**) and frequencies (**o**) of FarRed⁺CD14⁺ cells (RAW 264.7 cells) or FarRed⁺CD8⁺ cells (T cells) are shown. **p**, The frequencies of SIRPα⁺ cells. *n* = 4 biologically independent samples. Data are mean ± s.e.m. Statistical analysis was performed using two-tailed Student's *t*-tests (**l** and **p**) and one-way ANOVA with Tukey's post-test (**h**–**k** and **o**).

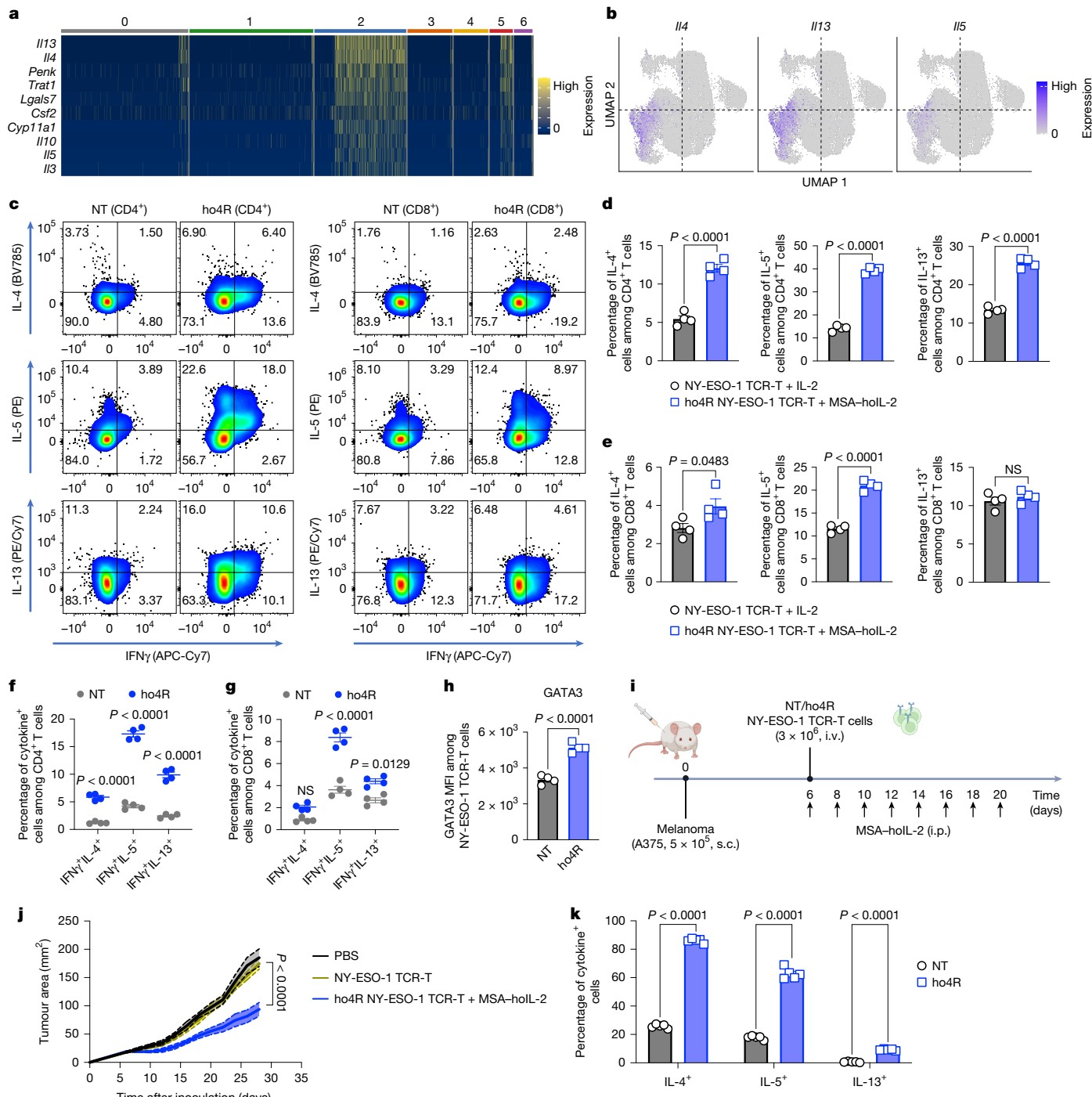

**Fig. 5 | Human orthogonal chimeric IL-4R signalling redirects T cell differentiation into a type 2 phenotype and improves anti-tumour activity in the human melanoma xenograft model. a**, Mouse TIL scRNA-seq analysis (Fig. 3) of the expression levels of marker genes for C2. **b**, Expression of *Il4*, *Il13* and *Il5* mRNA. **c**, Representative flow cytometry plots with quadrant gating showing the IL-4⁺, IL-5⁺, IL-13⁺ and IFNγ⁺ cytokine subpopulations in CD4⁺ and CD8⁺ NT or human orthogonal chimeric IL-4R (ho4R)-transduced NY-ESO-1 TCR-T cells. **d**, The frequencies of the IL-4⁺, IL-5⁺ and IL-13⁺ subpopulations among CD4⁺ TCR-T cells. *n* = 4 biologically independent samples. **e**, The frequencies of the IL-4⁺, IL-5⁺ and IL-13⁺ subpopulations among CD8⁺ TCR-T cells. *n* = 4 biologically independent samples. **f**, The frequencies of the IFNγ⁺IL-4⁺, IFNγ⁺IL-5⁺ and IFNγ⁺ IL-13⁺ subpopulations among CD4⁺ TCR-T cells. *n* = 4 biologically independent samples. **g**, The frequencies of the IFNγ⁺IL-4⁺, IFNγ⁺IL-5⁺ and IFNγ⁺IL-13⁺ subpopulations among CD8⁺ TCR-T cells. *n* = 4 biologically independent samples.

**h**, The GATA3 MFI. *n* = 4 biologically independent samples. **i,j**, Female NSG mice bearing s.c. HLA*0201⁺NY-ESO-1⁺ A375 tumours received i.v. transfer of 3 × 10⁶ NT or ho4R-expressing CD3⁺ NY-ESO-1 TCR-T cells on day 6, followed by i.p. administration of MSA–hoIL-2 (1 × 10⁵ U per day) every other day. *n* = 5 mice. **i**, The experimental timeline. **j**, Tumour growth curves. **k**, The experimental setting is described in Extended Data Fig. 7d. Female NSG mice bearing s.c. A375 tumours received an i.v. transfer of 2 × 10⁶ NT or ho4R-expressing CD3⁺ NY-ESO-1 TCR-T cells on day 10, followed by i.p. administration of MSA–hoIL-2 (2.5 × 10⁴ U, 3 doses, 1 × 10⁵ U, 3 doses) every other day. On day 21, mice were euthanized for flow cytometry analyses. *n* = 5 mice. The frequencies of the IL-4⁺, IL-5⁺, IL-13⁺ subpopulations among CD3⁺ TCR-T cells in spleens are shown. Data are mean ± s.e.m. Statistical analysis was performed using two-tailed Student's *t*-tests (**d**–**h** and **k**) and two-way ANOVA with Tukey's post test (**j**). The diagram in **i** was created using BioRender. Zhao, Y. (2025) https://BioRender.com/98rylzr.

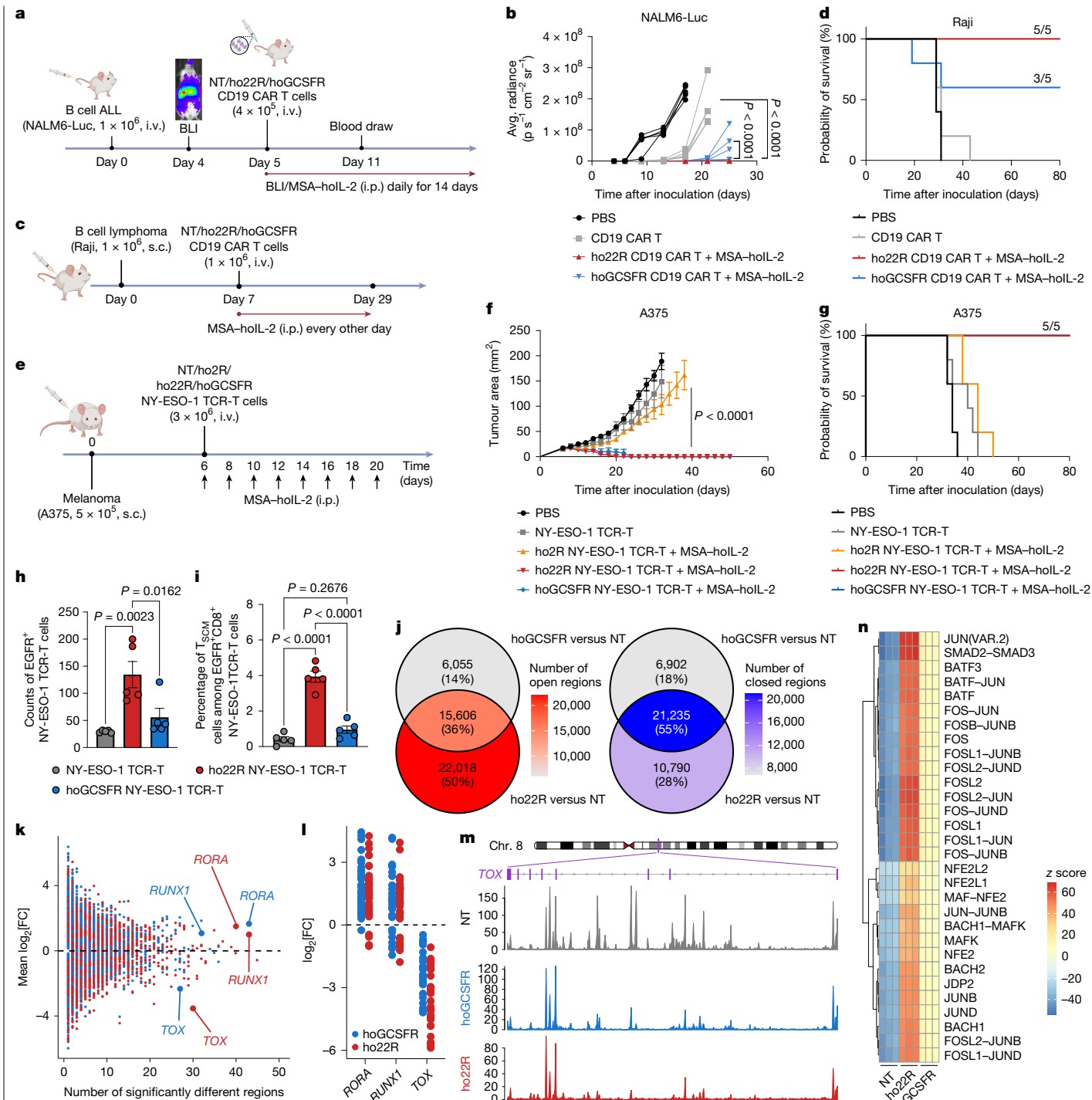

**Fig. 6 | Enhanced potency of CD19 CAR T cells and NY-ESO-1 TCR-T cells through ho22R and hoGSCFR signalling in xenograft tumour models. a,b**, Female NSG mice bearing NALM6-Luc B cell acute lymphoblastic leukaemia (ALL) received an i.v. transfer of $4 \times 10^5$ CD3$^+$ CD19 CAR T cells on day 5, followed by i.p. administration of MSA–hoIL-2 ($2.5 \times 10^4$ U) every day. $n = 5$ mice. The blood samples were analysed on day 11. **a**, The experimental timeline. **b**, Individual radiance measured using bioluminescence imaging (BLI). **c,d**, Female NSG mice bearing Raji lymphoma received an i.v. transfer of $1 \times 10^6$ CD3$^+$CD19$^+$ CAR T cells on day 7, followed by i.p. administration of MSA–hoIL-2 ($2.5 \times 10^4$ U) every other day. $n = 5$ mice. **c**, The experimental timeline. **d**, Survival curves. **e–g**, Female NSG mice bearing A375 melanoma received an i.v. transfer of $3 \times 10^6$ CD3$^+$ NY-ESO-1 TCR-T cells on day 6, followed by i.p. administration of MSA–hoIL-2 ($1 \times 10^5$ U) every other day. $n = 5$ mice. **e**, The experimental timeline. **f**, Tumour growth curves. **g**, Survival curves. **h,i**, The experimental setting is described in Extended Data Fig. 11c. $n = 5$ mice. **h**, Counts of CD3$^+$EGFR$^+$ TCR-T

cells in tumours. **i**, The frequencies of $T_{SCM}$ cells among EGFR$^+$CD8$^+$ TCR-T cells in tumours. **j–n**, The experimental setting is described in Extended Data Fig. 11c. $n = 3$ biologically independent samples. **j**, Overlap of differential chromatin accessibility regions between groups. False-discovery-rate (FDR)-adjusted $P < 0.05$. **k**, The number of significantly differentially accessible chromatin regions nearby versus their mean log$_2$-transformed fold change per gene. **l**, The log$_2$-transformed fold change in significantly differentially accessible chromatin regions nearby *RORA*, *RUNX1* and *TOX*. **m**, Representative chromatin accessibility profiles of *TOX*. Chr., chromosome. **n**, The most differentially enriched TF motifs in open chromatin regions. Data are mean ± s.e.m. Statistical analysis was performed using one-way ANOVA (**h** and **i**), two-way ANOVA with Tukey's post-test (**b** and **f**) or log-rank (Mantel–Cox) tests (**d** and **g**). The diagrams in **a**, **c** and **e** were created using BioRender. Zhao, Y. (2025). **a**, https://BioRender.com/giefd3b; **c**, https://BioRender.com/ulrtai9; **e**, https://BioRender.com/f6qd7eb.

control CAR T cells had modest effects (Fig. 6c,d and Extended Data Fig. 9c), whereas hoGCSFR CAR T cells controlled tumour growth (curing 60% of female and 100% of male mice) and ho22R CAR T cells eradicated tumours in all treated mice (Fig. 6d and Extended Data Fig. 9c–e). ho22R enhanced CAR T cell enrichment in tumours, where they exhibited reduced exhaustion (lower PD-1 and LAG3), and maintained IFNγ and GZMB secretion (Extended Data Fig. 9f–j). These findings demonstrate that ho22R and hoGCSFR CAR T cells produce robust and durable anti-tumour immunity against haematological malignancies.

To assess solid tumours, we next co-expressed ho2R, ho22R or hoGCSFR with NY-ESO-1 TCRs (Extended Data Fig. 10a). ho22R and hoGCSFR activated multiple STATs, while ho2R specifically activated STAT5 (Extended Data Fig. 10b). Despite better expansion with ho2R T cells (Extended Data Fig. 10c), ho22R and hoGCSFR showed more potent tumour cell killing and persistence after repeated stimulation with NY-ESO-1+ M407 cells (Extended Data Fig. 10d). Both ho22R and hoGCSFR retained a higher frequency of IFNγ+ cells, while hoGCSFR uniquely enriched GZMB+ cells (Extended Data Fig. 10e). Notably, ho22R preserved stem-like features during chronic stimulation (Extended Data Fig. 10f). In the A375 melanoma model (Fig. 6e), ACT with ho22R or hoGCSFR TCR-T cells resulted in substantial tumour control in both female (Fig. 6f,g and Extended Data Fig. 10g) and male (Extended Data Fig. 11a,b) mice, whereas ho2R and control TCR-T cells were ineffective. Notably, ho22R and hoGCSFR led to complete tumour remission in all treated female mice (Fig. 6g). ho22R TILs displayed enhanced expansion, an increased $T_{SCM}$ (CD45RO⁻ CCR7+CD45RA+CD95+CD27+) cell population and downregulated inhibitory receptors (PD-1, LAG3 and TIM3) (Fig. 6h,i and Extended Data Fig. 11c,d).

Epigenetic regulation has a crucial role in T cell fate decisions. Assay for transposase-accessible chromatin using sequencing (ATAC-seq) analysis of ho22R and hoGCSFR TILs revealed chromatin landscapes that were distinct from the NT control cells (Extended Data Fig. 11c,e). We identified 22,018 differentially open regions and 10,790 closed regions between ho22R and NT cells, and 6,055 differentially open regions and 6,902 closed regions between hoGCSFR and NT cells (Fig. 6j). We performed KEGG pathway enrichment analysis focusing on promoter-proximal regions (Extended Data Fig. 11f). The cytokine–cytokine receptor interaction pathway was enriched in both engineered cells compared with in the NT control cells (Extended Data Fig. 11g,h). We next identified key genes with the most pronounced chromatin remodelling. *RORA*, linked to $T_H17$ cell differentiation, and *RUNX1*, involved in T cell development, showed increased accessibility in ho22R and hoGCSFR cells (Fig. 6k,l), whereas the exhaustion-associated gene *TOX* exhibited more than 25 regions of reduced accessibility (Fig. 6k–m). Motif enrichment analysis identified enrichment of motifs bound by BACH2, a TF linked to restraining terminal differentiation[32], and those bound by AP-1 family TFs (JUN, FOS, JUNB, JUND, FOSL1, FOSL2 and BATF1 and BATF3) in the open chromatin regions in ho22R and hoGCSFR cells (Fig. 6n). *BACH2* KO in ho22R T cells led to a loss of stemness characteristics (CD62L, TCF1 and partial FOXO1) and a shift towards a CD44^high effector program (Extended Data Fig. 11i–m). These findings suggest that ho22R and hoGCSFR epigenetically rewire TILs to augment stemness and resistance to exhaustion, thereby facilitating efficient clearance of solid tumours.

## Discussion

Cytokines shape T cell fate, with the most well-characterized effects mediated by receptors naturally expressed on T cells that are activated through dimerization. Given that there are approximately 35–40 cytokine receptors, and only 4 shared JAK–TYK kinases, the ability of JAKs to engage in cross-talk is a biological necessity[13]. We leveraged this cross-talk, and the heterodimeric activation of JAK–STAT cytokine receptors[19], using the orthogonal IL-2 system that is reliant on the endogenous $\gamma_c$ on T cells. $\gamma_c$ signals through JAK3, a more promiscuous signalling hub than other JAKs[1]. Using this platform, we explored $\gamma_c$ cytokine–receptor pairings that either do not occur in nature, are rarely studied in anti-tumour properties (such as IL-10, IFNs) or involve receptors not naturally expressed on T cells (for example, IL-20R, IL-22R, GCSFR, EPOR). We assessed these natural and non-natural receptors on cell fate determination and anti-tumour efficacy in ACT.

$\gamma_c$ cytokines have been explored in ACT for supporting T cell proliferation and survival[33,34]. Among them, o4R induced the most potent anti-tumour responses, despite the recognized pro-tumour role of IL-4[35,36]. In support of this, local Fc–IL-4 delivery with ACT eradicated solid tumours by revitalizing terminally exhausted CD8+ T cells without inducing type 2 polarization[37]. By contrast, o4R selectively drove type 2 cell differentiation, positioning $T_C2/T_H2$ cell subsets as key effectors. Previous studies showed that antigen-specific $T_H2$ cells effectively treat myeloma and lymphoma[38], and memory $T_H2$ cells outperform $T_H1$ cells in tumour rechallenge[39]. Our findings, together with these, suggest that $T_C2/T_H2$ cells directly kill tumour cells, independent of type 2 cytokine secretion. Instead, secreted cytokines probably act through paracrine signalling. GATA3, a key $T_H2$ cell TF, is essential for o4R-driven tumour killing, indicating its broader role in T cell effector programs. Recent studies link a high proportion of type 2 cells in CAR T therapy to durable remissions[31,40]. Attempts to enhance this response through IL-4 priming improved leukaemia clearance but failed to induce a type 2 program[40], probably due to insufficient IL-4R expression. Building on these insights, engineering an orthogonal IL-4 receptor enables selectively in vivo generation of antigen-specific type 2 T cells. Beyond ACT, this approach may have applications in treating graft-versus-host disease[41,42].

Cell reprogramming through forced expression of lineage-instructing TFs has enabled lineage conversion[43,44] and, more recently, the therapeutic potential of reprogramming tumour cells into antigen-presenting cells has been investigated[45,46]. In the haematopoietic system, T cell precursors can be reprogrammed into myeloid lineage through C/EBPα or PU.1, giving rise to macrophage or dendritic cell populations[47]. However, reprogramming committed lymphocytes remains a challenge. Enforced C/EBPα and C/EBPβ expression disrupts PAX5 in B cells, enabling macrophage conversion through PU.1 cooperation, probably due to B cells being closer in lineage and transcriptionally similar to myeloid cells[48]. By contrast, T cells resist TF-induced reprogramming, consistent with our finding that NFIC, MAF and MAFB did not alter T cell identity. Cytokine receptor signalling regulates TF networks, but its role in lineage commitment remains underexplored. IL-2 and GM-CSF receptors can redirect common lymphoid progenitors toward myeloid lineages, demonstrating the potential of cytokine receptors in lineage convergence[49]. However, evidence for reprogramming mature T cells to a myeloid fate is lacking and still hypothetical. Ectopic expression of oGCSFR in activated CD8+ T cells reveals new plasticity, inducing a subset expressing myeloid markers (Mac-1+Gr-1+CD14+CD64+) resembling granulocytes or neutrophils with acquired phagocytic function. These cells retained CD8 expression and cytotoxicity, suggesting that GCSFR signalling induces myeloid transcription without antagonizing T cell commitment factors. These engineered cells exhibited dual functionality, maintaining cytotoxicity while acquiring macrophage-like engulfing capacity, potentially overcoming expansion limitations in CAR-macrophage engineering[50] and offering synergy with antibody-based cancer therapies.

Recognizing the pivotal role of the IL-10R–STAT3 in rejuvenating exhausted T cells and supporting memory, we explored other IL-10 family receptors that activate STAT3[51–53]. IL-20R and IL-22R, which are absent on native T cells, drove stronger pSTAT3 through o20R and o22R

compared with o10R. In particular, o22R also triggers STAT1, STAT4 and STAT5, and leads to superior efficacy in ACT immunotherapy. This outcome probably arises from coordinated STAT activation, promoting a polyfunctional cellular response: tumour infiltration, stemness and exhaustion resistance. The pSTAT patterns of o22R offering a blueprint to optimize STAT wiring in CAR and TCR-T cells. Transcriptional and epigenetic profiling of o22R T cells yields insights into the relationship between proximal signalling and the downstream mechanisms that maintain less-differentiated or $T_{SCM}$ cell states that contribute to anti-tumour immunity[54,55]. Despite persistent antigen exposure, o22R drives a subset of CD8+ T cells toward a stem-like fate, probably through strong STAT3 activation and induction of *Bach2*[53], consistent with the role of IL-22R in promoting epithelial stemness[56]. Consistent with this, ho22R T cells showed a chromatin landscape with enrichment of motifs bound by AP-1 and BACH2. In response to TCR signalling, AP-1 factors are phosphorylated to engage effector differentiation. BACH2 competes with AP-1 and, when overexpressed, enforces the transcriptional and epigenetic programs of stem-like CD8+ T cells[32,57]. *BACH2* KO disrupts ho22R-induced stemness and drives effector programs. It is therefore plausible that ho22R triggers *BACH2* expression to endow CD8+ T cells with stem-like properties and enhanced anti-tumour capacity, although the exact molecular cascades remain to be elucidated. Further systems-level studies of this orthogonal receptor library will identify shared JAK–STAT-driven programs that result in favourable anti-tumour functions.

In summary, our study demonstrates the use of an orthogonal cytokine–receptor engineering as a synthetic platform that exploits the promiscuity of $\gamma_c$ and JAK3 to redirect and optimize T cell states. In doing so, we have shown that the $\gamma_c$ cytokine–receptor pairing code on T cells can be expanded to generate new functions and cell states. This approach offers a promising strategy to potentiate T cells against multiple types of solid tumours and possibly in other clinical settings. It also suggests that differential combinatorial pairings between cytokine receptors that were not anticipated or used by nature could be a rich source of new immunological functions.

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

# Methods

## Mice

Female Thy1.2$^+$ C57BL/6 (C57BL/6J) mice (aged 5–6 weeks) and female and male *NOD.Cg-Prkdc$^{scid}$Il2rg$^{tm1Wjl}$/SzJ* (NSG) mice (aged 5–7 weeks) were purchased from Jackson Laboratory. TCR-transgenic Thy1.1$^+$ Pmel-1 mice (*B6.Cg-Thy1a/Cy Tg(TcraTcrb)8Rest/J*) were originally purchased from the Jackson Laboratory and maintained in the Stanford University-Lorry Lokey (SIM1) facility. Mice were housed in animal facilities approved by the Association for the Assessment and Accreditation of Laboratory Care, under a 12 h–12 h dark–light cycle, with ambient temperature maintained at 22 ± 2 °C and humidity at 40–60%. Experimental procedures in mouse studies were approved by the Institutional Animal Care and Use Committee (IACUC) at the Stanford University (animal protocol ID 32279) and performed in accordance with the guidelines from the animal facility of Stanford University.

## Cells and tumour models

The B16F10 mouse melanoma cell line, HEK293T cells, Raji cells, RAW 264.7 cells and J774A.1 cells were originally acquired from the American Type Culture Collection (ATCC). Platinum-E (Plat-E) and Platinum-GP (Plat-GP) retroviral Packaging Cell Line were purchased from Cell Biolabs. A375 and nRFP-M407 human melanoma cells were provided by A. Ribas. Mouse B cell lymphoma A20 cells were provided by R. S. Negrin and originally obtained from the ATCC. The NALM6 cell line was obtained from the ATCC. NALM6 cells were modified to express firefly luciferase (NALM6-Luc). Cell line authentication was performed by the supplier. All cell lines were confirmed to be mycoplasma free before use. B16F10, A375, HEK293T and Plat-E retroviral packaging cells were cultured in complete Dulbecco's modified Eagle's medium (DMEM) medium (Gibco/Thermo Fisher Scientific) supplemented with fetal bovine serum (FBS) (10% (v/v), Gibco/Thermo Fisher Scientific), penicillin–streptomycin (1% (v/v), Gibco/Thermo Fisher Scientific), L-glutamine (2 mM, Gibco/Thermo Fisher Scientific). Raji, nRFP-M407 and NALM6-Luc cells were maintained in RPMI-1640 supplemented with FBS (10% (v/v), Gibco/Thermo Fisher Scientific), and penicillin–streptomycin (1% (v/v), Gibco/Thermo Fisher Scientific). B16F10 tumour cells ($3 \times 10^5$) were implanted s.c. into the right flanks of C57BL/6 WT mice to establish the syngeneic s.c. tumour models. Raji tumour cells ($1 \times 10^6$) and A375 tumour cells ($5 \times 10^5$) were inoculated s.c. into the right flanks of NSG mice to establish the xenograft tumour models. NALM6-Luc tumour cells ($1 \times 10^6$) were injected i.v. into NSG mice to establish leukaemia xenograft models.

## Mammalian expression vectors

cDNA encoding mouse orthogonal IL-2Rβ and geneblock cDNA encoding mouse ICDs of IL-2R, IL-4R, IL-7R, IL-9R, IL-21R, IFNAG2, IFNGR1, IFNLR1, IL-10R, IL-20R, IL-22R, EPOR and GCSFR (IDT) were cloned into the retroviral vector pMSCV-MCS-IRES-YFP by PCR and isothermal assembly (ITA). Geneblock cDNA encoding mouse NFIC, MAF and MAFB (IDT) were cloned into the retroviral vector pMSCV-MCS-IRES-YFP by ITA. Geneblock cDNA encoding mouse chimeric orthogonal IL-2Rβ-ECD–GCSFR (IDT) were cloned into the retroviral vector pMSCV-truncated EGFR(tEGFR) by ITA. Similarly, human orthogonal IL-2Rβ (ho2R), chimeric orthogonal IL-2Rβ-ECD–IL-4R-ICD (ho4R), chimeric orthogonal IL-2Rβ-ECD–IL-22R-ICD (ho22R) and chimeric orthogonal IL-2Rβ-ECD–GCSFR-ICD (hoGCSFR) constructs were inserted into the pMSCV vector using the same methods. The STAT3-binding motif (YRHQ) and C-terminal domain (CTD) of IL-22R were fused at the C terminus of the mouse CD19 CAR construct (1D3) by ITA. The human CD19 CAR construct (FMC63) was acquired from Addgene (127889). The constructs for TCRs targeting NY-ESO-1 (1G4) in the pMSGV-tEGFR vector were provided by A. Ribas.

## Retrovirus production

HEK293T cells were seeded at $3 \times 10^6$ cells per 10 cm tissue culture dish. After incubation overnight, the medium was replaced with prewarmed DMEM with FBS (5%, v/v), penicillin–streptomycin (0.5%, v/v). For each transfection, 8.3 µg of plasmid (5 µg of plasmid containing the gene of interest plus 3.3 µg of pCL-Eco packaging plasmid) was added to 830 µl of Opti-MEM I reduced rerum medium (Gibco/Thermo Fisher Scientific), followed by the gradual addition of 25 µl of TurboFect transfection reagent (Thermo Fisher Scientific) with gentle vortexing. After incubation for 20 min, the transfection mixture was gently added to HEK293T cells. The culture medium was replaced 16 h later and, after an additional 24 h, the medium was collected, clarified by centrifugation and passed through a 0.45-µm filter. If not used immediately, virus was frozen at −80 °C for later use. To generate retrovirus for the transduction of human T cells, Plat-GP packaging cells (Cell Biolabs) were transfected with relevant plasmids using FuGENE HD transfection reagent (Promega). For each transfection, 20 µg of plasmid (consisting of 15 µg of expression plasmid plus 5 µg of pLTR-RD114A envelope plasmid) was added to 1 ml of Opti-MEM I medium, followed by addition of 60 µl of FuGENE HD transfection reagent. The culture medium was replaced 24 h later and, after an additional 24 h, the medium was collected, clarified by centrifugation and passed through a 0.45-µm filter before storage at −80 °C.

## Activation and transduction of primary mouse T cells

Spleens from C57BL/6 WT, pmel-1 mice were disintegrated mechanically and filtered through a 70-µm strainer (Thermo Fisher Scientific). Red blood cells (RBCs) were lysed with ACK lysis buffer (2 ml per spleen, Gibco/Thermo Fisher Scientific) for 5 min at 25 °C. The splenocytes were washed once with T cell medium, which contained RPMI-1640 (Gibco), FBS (10% (v/v), Gibco/Thermo Fisher Scientific), HEPES (25 mM, Gibco/Thermo Fisher Scientific), penicillin–streptomycin (1% (v/v), Gibco/Thermo Fisher Scientific), sodium pyruvate (1% (v/v), Gibco/Thermo Fisher Scientific), MEM non-essential amino acids solution (1% (v/v), Gibco/Thermo Fisher Scientific) and 2-mercaptoethanol (0.1% (v/v), Gibco/Thermo Fisher Scientific), and were then resuspended at a cell density of $2 \times 10^6$ cells per ml in a complete RPMI-1640 medium supplemented with MSA–mIL-2 (100 U ml$^{-1}$). For WT T cell activation, cells were activated with plate-bound anti-mouse CD3ε (2.5 µg ml$^{-1}$, 145-2C11, BioLegend) and soluble anti-mouse CD28 (5 µg ml$^{-1}$, 37.51, BioXCell) for 24 h. Pmel-1 T cells were activated with hgp100$_{25–33}$ peptide (1 µM, GenScript). Cells were cultured at 37 °C for 24 h. T cells were enriched by using Ficoll-Paque PLUS (Cytiva) and seeded onto precoated six-well plates at $3 \times 10^6$ cells per well in complete RPMI-1640 medium (3 ml per well). Then, 1 day before transduction, 12-well tissue culture plates were coated with retronectin (25 µg ml$^{-1}$, Takara) and placed into a 4 °C refrigerator overnight. The next day, the plates were blocked with 0.5% FBS in PBS for 30 min and washed with PBS. Activated T cells ($2 \times 10^6$ in 1 ml T cell medium) and viral supernatant (3 ml) were added to each well, along with MSA–mIL-2 (100 U ml$^{-1}$), and centrifuged at 2,500 rpm for 1.5 h at 32 °C. After incubating overnight, T cells were collected and expanded in a T cell medium supplemented with MSA–mIL-2 (100 U ml$^{-1}$). On day 3, the transduction efficiency was assessed on the basis of the expression of YFP using flow cytometry. Untransduced (NT) T cells activated and cultured in parallel were used as a control. Cells were collected for in vitro assays or i.v. injection 2 days after spinfection.

## Activation and transduction of primary human T cells

Primary human peripheral blood mononuclear cells isolated from a healthy human donor by leukapheresis were thawed and resuspended at a cell density of $2 \times 10^6$ cells per ml in T cell medium supplemented with human IL-2 (10 ng ml$^{-1}$, PeproTech). For human T cell activation, cells were activated with plate-bound anti-human CD3ε (1 µg ml$^{-1}$,

OKT-3, BioXCell) and soluble anti-human CD28 (5 µg ml⁻¹, 9.3, BioXCell) for 48 h. Activated T cells were collected and transduced on 12-well plates coated with retronectin (25 µg ml⁻¹, Takara) and loaded with 1.5 ml per well of each retrovirus (encoding NY-ESO-1 TCR clone 1G4 and ho2R, ho22R or hoGCSFR, or CD19 CAR clone FMC63 and ho22R or hoGCSFR) by spinfection. After incubating overnight, T cells were collected and expanded in a T cell medium supplemented with human IL-2 (10 ng ml⁻¹). Cells were stained with anti-human EGFR antibody (BioLegend, recognizes tEGFR in the NY-ESO-1 TCR construct) or anti-MYC tag antibody (Cell Signaling, recognizes the MYC-tag in the CD19 CAR construct). The transduction efficiency was detected on the basis of the expression of YFP, EGFR and MYC. NT T cells activated and cultured in parallel were used as control. Cells were expanded and collected for in vitro assays or i.v. injection 10–12 days after activation.

## CRISPR–Cas9 gene editing
Activated ho22R T cells were transduced with lentiCRISPR vectors encoding sgRNAs targeting BACH2 (5′-CATCCTTCCGGCACACAAAC-3′, 5′-CTAGCAACAGCCTCAAGCCG-3′, 5′-ACGTGACTTTGATCGTGGAG-3′) and expressing mCherry as a reporter. Transduction was performed by spinfection as described above. mCherry⁺ cells were sorted 72 h later for flow cytometry analysis and quantitative PCR. For ho4R T cells, sgRNAs targeting GATA3 (5′-CCTACTACGGAAACTCGGTC-3′, 5′-GGAGCTGTACTCGGGCACGT-3′, 5′-TCGACGAGGAGGCTCCACCC-3′) were complexed with Alt-R Cas9 protein to form ribonucleoproteins, followed by electroporation using the Lonza 4D-Nucleofector system. Cells were immediately transferred to IL-2 supplemented T cell medium, and the KO efficiency was assessed using flow cytometry.

## Flow cytometry analyses
For surface marker staining, cells were collected into U-bottom 96-well plates (Thermo Fisher Scientific), blocked with anti-mouse CD16/32 antibody (BioLegend) or human TruStain FcX (BioLegend), and incubated with the indicated antibodies at 4 °C for 20 min, followed by live/dead staining by 4′,6-diamidino-2-phenylindole (DAPI, Thermo Fisher Scientific). Cells were then washed and resuspended with FACS buffer (PBS containing 0.2% BSA, Sigma-Aldrich) for flow cytometry analyses. For pSTAT staining, primary mouse or human T cells were rested in T cell medium lacking IL-2 for 24 h before signalling assays. Cells were plated in a 96-well round-bottom plate in 50 µl T cell medium. Cells were stimulated by addition of MSA fused to mouse orthogonal IL-2 (MSA–oIL-2) or MSA fused to human orthogonal IL-2 (MSA–hoIL-2) for 15 min at 37 °C, and the reaction was terminated by fixation with 1.5% paraformaldehyde for 15 min at room temperature with agitation. Cells were washed and permeabilized with ice-cold 100% methanol for 60 min on ice. Next, cells were washed with FACS buffer before staining with pSTAT antibodies for 1 h at 4 °C in the dark. Cells were washed and resuspended in FACS buffer for flow cytometry analyses. For intracellular cytokine staining, cells were first stimulated by a cell stimulation cocktail (protein transport inhibitors included, Invitrogen/Thermo Fisher Scientific) at 37 °C for 5 h. After stimulation, cells were first stained for surface markers and Zombie Violet Fixable Dye (BioLegend), then fixed and permeabilized using the Cytofix/Cytoperm Fixation/Permeabilization Solution Kit (BD Biosciences). Intracellular staining with the indicated antibodies was performed according to the manufacturer's protocol. For TF or BrdU staining, cells were first stained for surface markers and Zombie Violet Fixable Dye. Next, cells were fixed and permeabilized using the FOXP3/Transcription Factor Staining Buffer Set (eBioscience) for TFs, or the BD Pharmingen APC BrdU Kit (BD Biosciences) for BrdU according to the manufacturer's instructions, followed by incubation with the indicated antibodies for intracellular staining. Cells were detected using the CytoFlex (Beckman Coulter) system. Analyses were performed using FlowJo (v.10.10.0).

## Antibodies and reagents for flow cytometry
The following antibodies or staining reagents were purchased from BioLegend: mouse CD16/32 (93, 101302), mouse CD90.1/Thy-1.1 (OX-7, 202533), mouse CD8β (YTS156.7.7, 126606), mouse CD8β (YTS156.7.7, 126614), mouse CD45.2 (104, 109808), mouse Ki-67 (16A8, 652413), mouse SCA1 (D7, 108142), mouse CD44 (IM7, 103030), mouse CD62L (MEL-14, 104428), mouse PD-1 (29F.1A12, 135220), mouse IFNγ (XMG1.2, 505850), mouse TNF (MP6-XT22, 506329), mouse/human GZMB (QA16A02, 372214), mouse/human KLRG1 (2F1/KLRG1, 138419), mouse IL-7Rα (A7R34, 135022), mouse CD122 (TM-β1, 123210), mouse SLAMF6 (330-AJ, 134606), mouse Mac-1 (M1/70, 101228), mouse CD14 (Sa14-2, 123335), mouse CD64 (X54-5/7.1, 139303), mouse Gr-1 (1A8, 127648), mouse SIRPα (P84, 144011), human EGFR (AY13, 352906), human CD3 (HIT3a, 300308), human CD4 (SK3, 344646), human CD8 (SK1, 344724), human IL-13 (JES10-5A2, 501914), human IL-4 (MP4-25D2, 500845), human IL-5 (TRFK5, 504304), mouse/human GATA3 (W19195B, 386906), human CXCR3 (G025H7, 353737), human CCR4 (L291H4, 359443), human CD62L (DREG-56, 304830), human CD95 (DX2, 305622), human CD45RA (HI100, 304120), human CD27 (O323, 302832), human CCR7 (G043H7, 353214), human CD45RO (UCHL1, 304228), human CD66b (G10F5, 305121), human LAG3 (11C3C65, 369322), human IFNγ (4S.B3, 502530), human Ki-67 (Ki-67, 350526), human CD39 (A1, 328240), human TIM3 (F38-2E2, 345026), human TruStain FcX (422302) and Zombie Violet Fixable Viability Kit (423114). The following antibodies or staining reagents were purchased from BD Biosciences: pSTAT3 (4/pSTAT3, 557815), pSTAT4 (38/pSTAT4, 558137), pSTAT5 (47/STAT5, 612599), mouse pSTAT6 (J71-773.58.11, 558252), human pSTAT6 (23/STAT6, 612701) and the BD Pharmingen APC BrdU Kit (552598). The following antibodies were purchased from Cell Signaling: pSTAT1 (58D6, 8009S) and MYC-tag (9B11, 3739/2233S). DAPI was purchased from Thermo Fisher Scientific. For flow cytometry staining, surface marker antibodies were used at a 1:200 dilution, intracellular antibodies at 1:100 and pSTAT antibodies at 1:50.

## Protein production
The cDNAs encoding mouse and human orthogonal IL-2 was cloned into the mammalian expression vector pD649, which includes a C-terminal 8×His tag for affinity purification. DNA encoding MSA was purchased from Integrated DNA Technologies (IDT) and cloned into pD649 as an N-terminal fusion. Mammalian expression DNA constructs were transfected into HEK293F cells using the Expi293 Expression System (BD Biosciences) for secretion and purified from the clarified supernatant by nickel affinity resin (Ni-NTA, Qiagen) followed by size-exclusion chromatography with a Superdex-200 column (Cytiva) and formulated in sterile PBS for injection. Endotoxin was removed using the Proteus NoEndo HC Spin column kit according to the manufacturer's recommendations (VivaProducts) and endotoxin removal was confirmed using the Pierce LAL Chromogenic Endotoxin Quantification Kit (Thermo Fisher Scientific). Proteins were concentrated, flash-frozen in liquid nitrogen and stored at −80 °C until ready for use.

## In vivo anti-tumour therapy studies
C57BL/6 WT mice bearing established s.c. B16F10 tumours were sub-lethally lymphodepleted by total body irradiation (5 Gy) on day 5. On day 6, mice received i.v. adoptive transfer of 3 × 10⁶ pmel-1 T cells (either NT or transduced with o2R, o4R, o7R, o9R, o21R, o10R, o20R, o22R, oIFNAR2, oIFNGR1, oIFNLR1, oEPOR or oGCSFR) followed by i.p. administration of MSA–oIL-2 (2.5 × 10⁴ U per day) or PBS control every other day until day 20. For the s.c. Raji tumour model, NSG mice received i.v. adoptive transfer of 1 × 10⁶ CD19 CAR T cells, ho22R CD19 CAR T cells or hoGCSFR CD19 CAR T cells on day 7 after tumour inoculation followed by i.p. administration of MSA–hoIL-2 (2.5 × 10⁴ U per day) or PBS control every other day until day 29. For the s.c. A375 tumour model, NSG mice received i.v. adoptive transfer of 3 × 10⁶ NY-ESO-1 TCR-T cells

(WT NY-ESO-1 TCR-T cells, or ho2R-, ho4R-, ho22R- or hoGCSFR-expressing NY-ESO-1 TCR-T cells) on day 6 after tumour inoculation followed by i.p. administration of MSA−hoIL-2 ($1 \times 10^5$ U per day) or PBS control every other day until day 20. For the s.c. tumour model, the tumour area was calculated using the formula area = length × width from calliper measurements of two orthogonal diameters. For the NALM6-Luc leukaemia model, NSG mice received i.v. adoptive transfer of $4 \times 10^5$ CD19 CAR T cells, ho22R CD19 CAR T cells or hoGCSFR CD19 CAR T cells on day 5 after tumour inoculation followed by i.p. administration of MSA−hoIL-2 ($2.5 \times 10^4$ U per day) or PBS control every day until day 18. Mice were anaesthetized and i.p. injected with bioluminescent substrate D-luciferin potassium salt (30 µg ml$^{-1}$, 100 µl, GoldBio) prediluted in PBS. Then, 10 min after injection, the mice were subjected to luminescence imaging using the Xenogen IVIS fluorescence/bioluminescence imaging system for tumour growth monitoring. Sample size was not predetermined statistically but was based on standards commonly used in previous studies. Mice were assigned to experimental groups to ensure uniform starting tumour sizes. No blinding was performed during the study. Post-therapy survival of mice was monitored for at least 90 days after tumour inoculation. Mice were euthanized when body weight loss was beyond 15% of the baseline weight, the tumour area reached 200 mm$^2$ or any signs of discomfort were detected by the investigators or as recommended by the caretaker who monitored the mice every day.

## Immunophenotyping by flow cytometry
C57BL/6 mice were inoculated s.c. with B16F10 tumour cells ($1 \times 10^6$), sublethally lymphodepleted by irradiation on day 7 and received i.v. adoptive transfer of $3 \times 10^6$ pmel-1 T cells (either NT or transduced with o2R, o4R, o7R, o9R, o21R, o10R, o20R, o22R, oIFNAR2, oIFNGR1, oIFNLR1, oEPOR or oGCSFR) on day 8 after tumour inoculation followed by i.p. administration of MSA−oIL-2 ($2.5 \times 10^4$ U per day) or PBS control every other day until day 16. Mice received i.p. administration of bromodeoxyuridine (BrdU) (1 mg per 100 µl, BD Pharmingen) on day 16. On day 17, mice were euthanized, and tumours, TDLNs and spleens were collected. NSG mice were inoculated s.c. with A375 cells ($1 \times 10^6$) and received i.v. ACT of $3 \times 10^6$ NY-ESO-1 TCR-T cells (WT NY-ESO-1 TCR-T cells, or ho2R-, ho22R- or hoGCSFR-expressing NY-ESO-1 TCR-T cells) on day 10 after tumour inoculation followed by i.p. administration of MSA−hoIL-2 ($2.5 \times 10^4$ U, 3 doses, $1 \times 10^5$ U, 3 doses) or PBS control every other day until day 20. On day 21, mice were euthanized, tumours and spleens were collected. For the Raji tumour model, NSG mice were inoculated (s.c.) with Raji cells ($2 \times 10^6$) and received an i.v. adoptive transfer of $1 \times 10^6$ NT, ho22R or hoGCSFR CD19 CAR T cells on day 11, followed by i.p. administration of MSA−hoIL-2 ($2.5 \times 10^4$ U) or PBS every other day until day 25. On day 26, the mice were euthanized, and tumours were collected for flow cytometry analysis. Collected tumours were weighed, mechanically minced and digested in RPMI-1640 medium supplemented with collagenase type IV (1 mg ml$^{-1}$, Gibco/Thermo Fisher Scientific), dispase II (100 µg ml$^{-1}$, Sigma-Aldrich), hyalurondase (100 µg ml$^{-1}$, Sigma-Aldrich) and DNase I (100 µg ml$^{-1}$, Sigma-Aldrich) at 37 °C for 60 min. RBC lysis was performed on the digested tumour samples with ACK lysing buffer. Tumour-infiltrating leukocytes were then enriched by Percoll (Cytiva) density-gradient centrifugation, resuspended in PBS with BSA (0.2%, w/v), stained with the indicated antibodies and analysed by flow cytometry. Spleens were ground and filtered through a 70-µm strainer (Thermo Fisher Scientific). RBC lysis was performed on the spleen samples with ACK lysing buffer (2 ml per spleen, Gibco/Thermo Fisher Scientific) and then resuspended in PBS with BSA (0.2%, w/v). TDLNs were ground and filtered through a 70-µm strainer (Thermo Fisher Scientific) and then resuspended in PBS with BSA (0.2%, w/v). Cells collected from blood, spleen, tumour and TDLNs were stained with the indicated antibodies and analysed on the CytoFLEX flow cytometer (Beckman Coulter).

## Phagocytosis assay
oGCSFR-transduced pmel-1 CD8$^+$ T cells were maintained in MSA−oIL-2 (500 nM) for 72 h. Live CD8$^+$Mac-1$^-$ and CD8$^+$Mac-1$^+$ cells were sorted using the Aria II sorter (BD Biosciences) at the Stanford Shared FACS Facility, with NT cells sorted as controls. Phagocytic activity was assessed using the pHrodo Red *S. aureus* BioParticles Conjugate (Thermo Fisher Scientific) according to the manufacturer's protocol. In brief, sorted cells and J774A.1 cells were seeded into a 96-well U-bottom plate and incubated with a suspension of pHrodo Red *S. aureus* BioParticles for 30 min. After incubation, extracellular fluorescent probes were washed off, and the cells were stained for surface markers and with DAPI. Fluorescence was measured on the CytoFLEX flow cytometer (Beckman Coulter). For real-time phagocytosis analysis, sorted cells were seeded into a 96-well flat-bottom plate (10,000 cells per well) and incubated with pHrodo Red *S. aureus* BioParticles (Thermo Fisher Scientific). Fluorescence images of each well were captured every 20 min using the IncuCyte live imaging system (Essen Bioscience), and phagocytosis was quantified as the percentage confluence. For the live bacterial phagocytosis assay, NT or oGCSFR-tEGFR-transduced CD8$^+$ T cells were seeded into a 96-well flat-bottom plate (200,000 cells per well) and incubated with GFP$^+$ *L. monocytogenes* (1/2a) derived from FDA LS808 (Microbiologics) at a series of concentrations, with 10× dilution for 2 h. Next, the supernatant was removed, and the cells were cultured in gentamicin-containing RPMI complete medium at 37 °C for an additional 22 h. GFP signals from each well were analysed on the CytoFLEX flow cytometer (Beckman Coulter). Live-cell imaging without sorting was performed using the Leica TCS SP8 Confocal system with a white-light laser. Fixed cells were sorted and visualized using the Zeiss Elyra7 lattice SIM microscope.

## ADCP assay
A20 cells were labelled with FarRed using the Invitrogen CellTrace Far Red Cell Proliferation Kit (Thermo Fisher Scientific) according to the manufacturer's instructions. Labelled A20 cells ($5 \times 10^6$) were then incubated with InVivoMAb anti-mouse CD19 (5 µg ml$^{-1}$, ID3, BioXCell) in 1.25 ml RPMI medium at 37 °C for 10 min. No-ICD o2R and oGCSFR-transduced pmel-1 CD8$^+$ T cells were maintained in MSA−oIL-2 (500 nM) for 72 h. For the blocking experiment, T cells were pretreated with a blocking cocktail containing anti-mouse CD64 (BioLegend, X54-5/7.1) and anti-mouse CD16/32 (BioLegend, 93) at 1 µg ml$^{-1}$. RAW264.7 cells, NT pmel-1 CD8$^+$ T cells, no-ICD o2R-transduced pmel-1 CD8$^+$ T cells and oGCSFR-transduced pmel-1 CD8$^+$ T cells were seeded into a 96-well round-bottom plate at $5 \times 10^4$ cells per well. FarRed$^+$ A20 target cells were added at a target:effector ratio of 2:1, and co-cultures were incubated for 2 h. After incubation, cells were stained for surface markers, then with DAPI (Sigma-Aldrich) and then washed with FACS buffer and analysed using the CytoFLEX flow cytometer (Beckman Coulter). Confocal imaging (Leica TCS SP8 with a white-light laser) was performed at a effector:target (E:T) ratio of 1:1, and co-cultures were incubated for 24 h.

## Human T cell differentiation assay
Human T cells were activated on day 0 using ImmunoCult Human CD3/CD28 T Cell Activator (StemCell) and retrovirally transduced to generate NY-ESO-1 TCR-T cells or ho4R NY-ESO-1 TCR-T cells on day 2. After transduction, ho4R NY-ESO-1 TCR-T cells or NY-ESO-1 TCR-T cells were cultured in RPMI complete medium supplemented with MSA−hoIL-2 (100 nM) or recombinant human IL-2 (10 nM, Peprotech), respectively. The cell density was adjusted to $1 \times 10^6$ cells per ml every 2 days as needed with fresh complete medium supplemented with MSA−hoIL-2 or recombinant human IL-2. After 7 days of incubation, cells were reactivated using ImmunoCult Human CD3/CD28 T Cell Activator and cultured for an additional 7 days with cell density adjustments performed as previously described. On day 14, differentiated cells were

stained for the indicated markers and analysed using the CytoFlex flow cytometer (Beckman Coulter).

## Cell killing assays

CD19 CAR T cells (either NT or transduced with ho22R or hoGCSFR) were co-cultured with target cells (NAML6-Luc or Raji) at the different E:T ratios in the presence or absence of MSA–hoIL-2 (100 nM). After co-culture for 24 h, the cells in the plates were collected and analysed by flow cytometry to determine the viability of tumour cells. NT T cells, ho4R NY-ESO-1 TCR-T cells, ho4R NY-ESO-1 TCR-T cells (*GATA3* KO) or ho4R NY-ESO-1 TCR-T cells with anti-IL-4 (BioXCell, MP4-25D2, 2 µg ml$^{-1}$), anti-IL-5 (BioXCell, TRFK5, 2 µg ml$^{-1}$), anti-IL-13 (BioXCell, tralokinumab, 2 µg ml$^{-1}$) or their combination (2 µg ml$^{-1}$ each) were co-cultured with A375 cells at an E:T ratio of 1:2 for 48 h. Viable tumour cell counts were assessed by flow cytometry. Human melanoma cells (nRFP-M407, 1 × 10$^6$ per well) were plated in six-well plates. NT T cells, NY-ESO-1 TCR-T cells or NY-ESO-1 TCR-T cells co-transduced with ho2R, ho22R or hoGCSFR were added in duplicate at a 1:1 E:T ratio with MSA–hoIL-2 (100 nM). Every 48 h, the cells were collected, washed, resuspended in fresh MSA–hoIL-2 and added to the nRFP-M407 tumour cells at a 1:1 ratio. For real-time cell killing analysis, after each 48 h co-culture, the T cells were collected from the six-well plates, resuspended in fresh MSA–hoIL-2 (100 nM) and added to nRFP-M407-preseeded flat-bottom 96-well plates (40,000 cells per well) at a 1:1 E:T ratio and fluorescence images were obtained for each well every 3 h using the IncuCyte live imaging system (Essen Bioscience) and quantified by percentage confluence.

## Human T cell repetitive stimulation assays

Human melanoma cells (A375, 1 × 10$^6$ per well) were plated in six-well plates. NT T cells, NY-ESO-1 TCR-T cells or NY-ESO-1 TCR-T cells co-transduced with either ho22R or hoGCSFR were added in duplicate at a 1:1 E:T ratio with MSA–hoIL-2 (100 nM). Every 48 h, the cells were collected, washed, resuspended in fresh MSA–hoIL-2 and were rechallenged with A375 tumour cells (1 × 10$^6$). After the three rounds of stimulation with target cells, the T cells were collected for phenotyping by flow cytometry.

## RNA-seq sample preparation and data analysis

WT mouse T cells transduced with orthogonal receptors were sorted using the Aria II sorter (BD Biosciences) at the Stanford Shared FACS Facility. No-ICD o2R-transduced cells were sorted as controls. After sorting, cells were recovered in MSA–mIL-2 (100 U ml$^{-1}$) for 24 h, subjected to IL-2 starvation for another 24 h and then stimulated with 5 µM MSA–oIL-2 or 10 nM recombinant cytokines (IL-2, IL-4, IL-7, IL-21, IL-10 and IFNα; Peprotech) for 6 h. Total RNA was extracted using the Quick-RNA 96 Kit (Zymo Research). Libraries were synthesized using the mRNA library preparation (poly A enrichment) kit according to the manufacturer's instructions. Libraries were pooled and sequenced on the NovaSeq X Plus Series (PE150). Reads were aligned to the mouse reference genome (mm10) using Rsubread (v.2.18.0)[58]. Gene expression was quantified with featureCounts. We first conducted differential expression analysis between the no-ICD o2R plus MSA–oIL-2 condition and cytokine-treated conditions (IFNα, IL-10, IL-21, IL-2, IL-7 and IL-4) to define differentially regulated genes induced or repressed by these canonical cytokines. DESeq2 (v.1.48.1) was used for this analysis, and genes with FDR-adjusted *P* values < 0.05 were considered to be differentially regulated. IFNα- and IL-4-driven genes were interpreted as STAT1- and STAT6-driven genes. For STAT3-driven genes, intersection was taken from IL-10- and IL-21-driven genes. For STAT5-driven genes, intersection was taken from IL-2- and IL-7-driven genes. We next computed the transcripts per million (TPM) for all genes across all samples. We then computed Pearson's correlation coefficients to each of the reference samples (for example, samples treated with IL-10 and IL-21 in the case of STAT3) by using only TPM values of genes driven by the STAT

of interest (for example, STAT3) but not driven by other STATs (such as STAT1, STAT5 or STAT6). As there are multiple reference samples, we took the mean of those correlation coefficients for each individual sample. We considered these averaged correlation coefficients to be STAT scores. PCA was performed with STAT scores. Heat-map and radar charts were generated using the fold-change values of these STAT scores compared with the no-ICD o2R plus MSA–oIL-2 condition.

## scRNA-seq sample preparation

C57BL/6 mice were inoculated s.c. with B16F10 tumour cells (1 × 10$^6$), sublethally lymphodepleted by irradiation on day 7 and received i.v. adoptive transfer of 3 × 10$^6$ pmel-1 T cells (NT or transduced with o2R, o4R, o20R, o22R or oGCSFR) on day 8 after tumour inoculation followed by i.p. administration of MSA–oIL-2 (2.5 × 10$^4$ U per day) or PBS control every other day until day 16. On day 17, the mice were euthanized, tumours were minced and dissociated using a mouse tumour dissociation kit (Miltenyi Biotec) and the gentleMACS Octo Dissociator (Miltenyi Biotec). TILs were first enriched by density-gradient centrifugation against Percoll (Cytiva), and then stained for surface markers and with DAPI (Thermo Fisher Scientific). Thy1.1$^+$CD8$^+$ or Thy1.1$^+$CD8$^+$YFP$^+$ T cells were sorted using the Aria II sorter (BD Biosciences) at the Stanford Share FACS Facility. Sorted cells were subjected to single-cell encapsulation using the Chromium Single Cell Instrument and reagents. A Chromium Next GEM Chip G was loaded with the appropriate number of cells, and the sequencing libraries were prepared using 10x Genomics reagents according to the manufacturer's instructions and passed quality control. In brief, an emulsion encapsulating single cells into droplets with reagents and gel beads containing a unique molecular identifier, reverse transcription reagents and cell barcoding oligonucleotides was generated. cDNAs were obtained and amplified after droplets broke. For the 3′ Gene Expression library, the cDNA was fragmented, ligated to a sequencing adaptor and PCR amplified. The generated 3′ Gene Expression libraries were sequenced using the NovaSeq 6000 system with a sequencing depth of >20,000 paired-end reads per cell. The fastq files were generated by Cell Ranger (v.7.1.0) mkfastq from 10x Genomics, and primary data analysis using a custom reference package based on the mm10 reference genome.

## Analysis of scRNA-seq data

The gene expression matrix was first processed using Seurat (v.5.1.0). For each dataset, we excluded cells containing fewer than 200 genes (to remove debris, empty droplets and low-quality cells) and also cells in which >20% of transcripts were derived from mitochondrial RNA, leaving 32,075 cells. All expression data were normalized by log transformation. Seurat was used to first normalize gene expression count data using the NormalizeData and ScaleData functions. Then, the FindVariableFeatures function was used to select the top 2,000 variable genes and PCA was performed. Cell clusters were identified using the FindNeighbors and FindClusters functions. After clustering, we omitted two clusters (*n* = 597 cells) with low *Cd8* and *Thy1* expression to exclude non pmel-1 T cell contaminants. We also omitted one cluster with low *Ptprc* and high *Hbb* expression and high ribosomal RNA content (*n* = 2,346 cells) to further exclude any non-leukocytes, and one cluster with negligible cell count (*n* = 124 cells). The final Seurat dataset contained 29,008 cells.

TF activity prediction was conducted using the pySCENIC (v.0.12.1) docker distribution with the default parameter settings. A total of 241 regulons was identified, with corresponding TFs annotated. The area under the receiver-operator curve (AUC) scores for all TFs were stored in a designated assay slot (termed scenic) and used for UMAP computation with the Seurat RunUMAP function with the default parameters. For unsupervised clustering, cells were first embedded into a ten-dimensional UMAP space, and clusters were then identified using the FindCluster function with the resolution parameter set to 0.02. For the search for marker TFs, the FindAllMarkers function was used with

the min.pct parameter set to 0.50 (that is, at least 50% of cells have positive TF activity), and TFs with FDR-adjusted $P$ value < 0.05 and average $\log_2$-transformed fold-change > 1 were considered to be markers. Up to ten TFs, sorted based on the average $\log_2$-transformed fold-change values, are shown in the main figure. For marker gene analysis, the FindAllMarkers function was used with min.pct set to 0.10 for the gene expression matrix. Standard Seurat functionalities were used for data visualization. Intercellular communication prediction was performed using LIANA (v.1.5.1)[59]. The built-in MouseConsensus ligand–receptor database was used. Five algorithms (natmi, connectome, logfc, sca and cellphonedb) were used, and only LR interactions concordant between all algorithms were retained.

### ATAC-seq sample preparation
NSG mice were inoculated s.c. with A375 cells ($1 \times 10^6$) and received i.v. ACT of $3 \times 10^6$ NY-ESO-1 TCR-T cells (WT NY-ESO-1 TCR-T cells, or ho22R- or hoGCSFR-expressing NY-ESO-1 TCR-T cells) on day 10 after tumour inoculation followed by i.p. administration of MSA–hoIL-2 ($2.5 \times 10^4$ U, 3 doses, $1 \times 10^5$ U, 3 doses) or PBS control every other day until day 20. On day 21, the mice were euthanized and tumours were minced and dissociated using the human tumour dissociation kit (Miltenyi Biotec) and the gentleMACS Octo Dissociator (Miltenyi Biotec). TILs were first enriched by density-gradient centrifugation against Percoll (GE healthcare), and were then stained for surface markers and with DAPI (Sigma-Aldrich). $CD3^+EGFR^+$ or $CD3^+EGFR^+YFP^+$ T cells were sorted using the Aria II sorter (BD Biosciences) at the Stanford Share FACS Facility. For ATAC-seq library preparation, intact nuclei from sorted cells were treated with a hyperactive Tn5 transposase mutant. This transposase simultaneously tags the target DNA with sequencing adapters and fragments the DNA. The tagmented DNA is then purified and amplified using indexed primers to generate libraries. Equal amounts of each sample were pooled and subjected to 50 bp paired-end sequencing on the NovaSeqX sequencer.

### Chromatin accessibility analysis
After routine quality control with FastQC, reads were mapped onto the reference genome (hg19) using RSubread (v.2.18.0) in the DNA mode[58]. The aligned reads were then subjected to peak calling using MACS3 (v.3.0.1)[60]. Downstream analyses were performed in R (v.4.4). Peaks were annotated with their genomic locations and associated genes using ChIPseeker (v.1.44.0)[61]. A set of non-redundant peaks across all samples was defined using ChIPQC. The reads aligned with each region were then counted using the summarizeOverlaps function, and the count matrix was analysed with DESeq2 (v.1.48.1)[62]. Count data normalized by variance-stabilizing transformation were subjected to PCA for visualization. Differential chromatin accessibility analysis was conducted according to the standard DESeq2 workflow between each transduction group (biological triplicates). Differential TF motif enrichment analysis was performed using chromVAR (v.1.30.1)[63]. Here, all samples were analysed simultaneously, and the TF motifs most significantly variably regulated among transduction group were chosen for heat-map analysis. *Cis*-regulatory element enrichment analysis were performed with GREAT (v.2.10.0)[64] for each sample separately. Gene Ontology terms significantly enriched in at least one sample in a given transduction group were considered. Terms enriched in the hoGCSFR group but not in the ho22R or the NT group were manually inspected.

### Statistical analysis
Statistical analysis was performed using GraphPad Prism 9 (GraphPad software), except bulk RNA-seq, scRNA-seq and ATAC-seq data, which were analysed with R (described above). All values and error bars are shown as mean ± s.e.m. Comparisons of two groups were performed by using two-tailed unpaired Student's $t$-tests. Comparisons of multiple groups were performed using one-way ANOVA with Tukey's multiple-comparison test unless otherwise indicated. Experiments that involved repeated measures over a time course, such as tumour growth, were performed using two-way ANOVA with Tukey's multiple-comparison post-test. Survival data were analysed using the log-rank (Mantel–Cox) test. No statistically significant differences were considered when $P$ values were larger than 0.05.

### Statistics and reproducibility
Animal studies were independently repeated 2–3 times. For groups with more than five mice, the total number of animals was pooled from two independent experiments. In vitro experiments were independently repeated 2–4 times with consistent results.

### Ethics statement
Experimental procedures in mouse studies were approved by the Institutional Animal Care and Use Committee (IACUC) at the Stanford University (animal protocol ID 32279) and performed in accordance with the guidelines from the animal facility of Stanford University. Primary T lymphocytes from healthy donors were provided by the Stanford Blood Center. Ethical approval pertaining to T cell donors was obtained by the Stanford Blood Center.

### Reporting summary
Further information on research design is available in the Nature Portfolio Reporting Summary linked to this article.

## Data availability
The raw and processed bulk RNA-seq data are publicly available at the Gene Expression Omnibus (GEO) under accession code GSE282973 (C57BL/6 WT T cells; aligned to the mm10 mouse reference genome). The scRNA-seq data are available at the GEO (GSE272444; TCR transgenic pmel-1 T cells in melanoma tumours; mm10 reference genome). ATAC-seq data generated in this study have been deposited at the GEO under accession code GSE272385 (human T cells in melanoma tumours grown in NSG mice; aligned to the hg19 human reference genome). Normalized gene expression matrices and associated sample metadata were downloaded from the GEO (https://www.ncbi.nlm.nih.gov/geo/) using the accession numbers above. All other raw data are available at Figshare[65] (https://doi.org/10.6084/m9.figshare.26322181). Additional supporting information is available from the corresponding author on reasonable request. Source data are provided with this paper.

## Code availability
The R code used to analyse the scRNA-seq data is publicly at Zenodo[66] (https://doi.org/10.5281/zenodo.15702063). Analysis details are provided in the Methods.

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

**Acknowledgements** We thank A. Ribas for providing NY-ESO-1 TCR plasmid, A375 cells and M407-nRFP cells; R. S. Negrin for providing A20 cells; and the staff at the Animal care facilities

at Stanford, Stanford Shared FACS Facility and Stanford Cell Sciences Imaging Facility for technical assistance. This work was supported by the Parker Institute of Cancer Immunotherapy (PICI) (K.C.G. and A.K.), National Institutes of Health (NIH) 3 U54 CA 244711 01 (K.C.G.), NIH RO1 AI-51321 (K.C.G.), Yosemite Innovation Fund, Ludwig Institute (K.C.G.), and NIH R37CA273074 (A.K.). K.C.G. is an investigator with the Howard Hughes Medical Institute. A.K. is a CRI Lloyd J. Old STAR (CRI14104). Y.Z. is a CRI-Margaret Dammann Eisner Postdoctoral Fellow and supported by the CRI Irvington Postdoctoral Fellowship (315511). M.O. is supported by NCI Predoctoral to Postdoctoral Fellow Transition (K00) award (4K00CA274708). G.E.R. was funded by PICI Rise Postdoctoral Scholarship.

**Author contributions** K.C.G. conceived the study, supervised the experiments and collaborated with the other authors on the manuscript. Y.Z. designed and conducted the experiments, analysed the data and wrote the manuscript. L.L.S. performed the signalling experiments shown in Fig. 1. A.K. supervised experiment design, analysed scRNA-seq data and collaborated on writing the manuscript. M.O. analysed bulk RNA-seq, scRNA-seq and ATAC-seq data, and contributed to experimental design and manuscript writing. A.P. assisted with scRNA-seq data analysis. D.W. and L.L.S. produced the orthogonal IL-2 proteins used in the study. H.J., G.E.R. and Q.S. assisted with in vivo immune phenotyping experiments. P.T. helped with human T cell transduction and orthogonal IL-2 variant experiments. X.C. assisted with CRISPR–Cas9 gene editing. L.W.R. and S.L. supported the Incucyte experimental setup and data analysis.

**Competing interests** K.C.G. is the founder of Synthekine and co-founder of Dispatch Therapeutics, which are developing cytokine receptor-based therapeutics. The chimeric receptors in this study are described in a pending patent application (PCT/US2016/050511). A.K. serves on the advisory board for and holds stock in Dispatch Therapeutics and Certis Oncology and consults for Sastra Cell Therapy. The other authors declare no competing interests.

**Additional information**
**Correspondence and requests for materials** should be addressed to Anusha Kalbasi or K. Christopher Garcia.

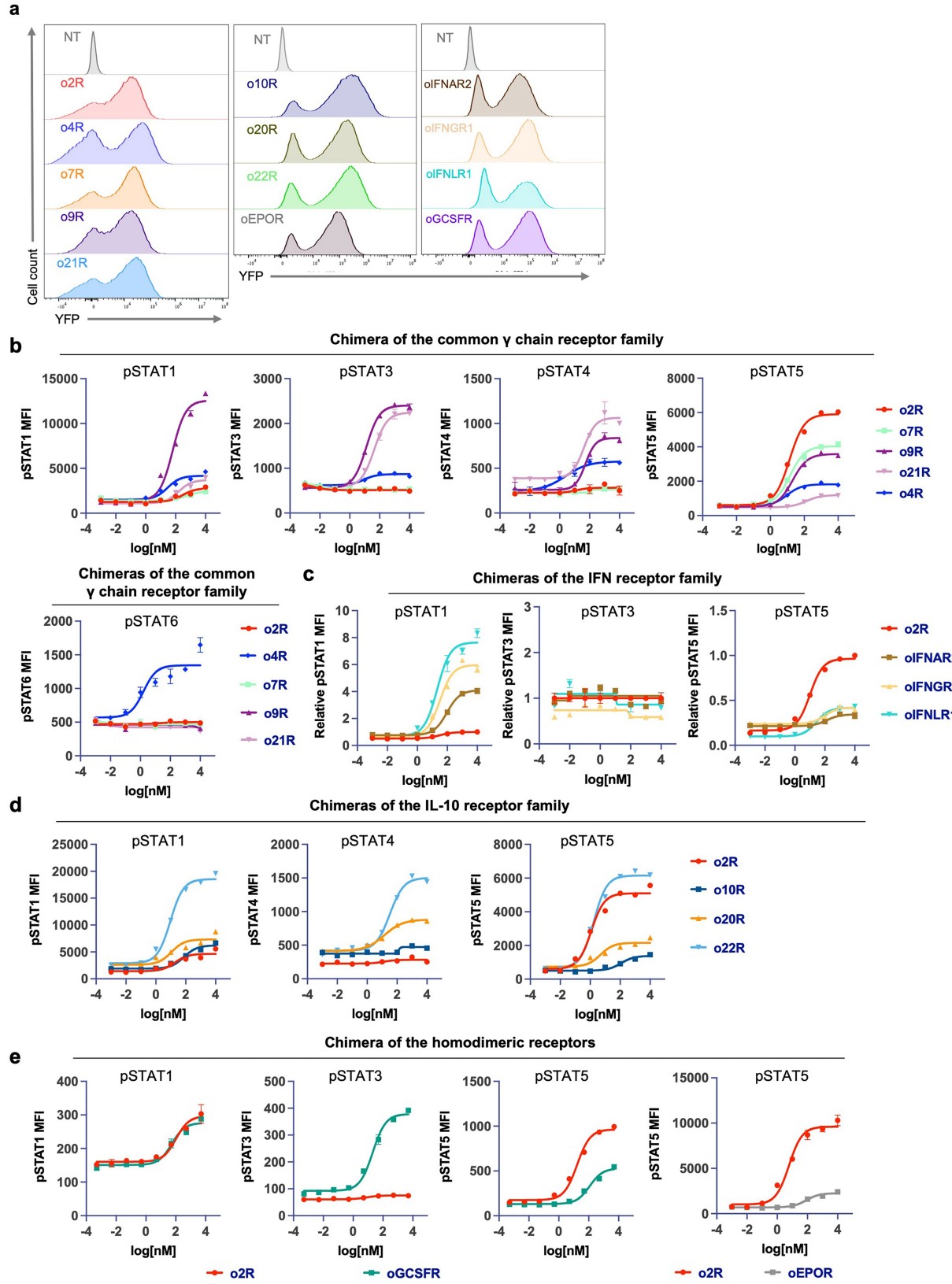

**Extended Data Fig. 1** | See next page for caption.

**Extended Data Fig. 1 | Characterization and signalling profiles of orthogonal receptor expressing WT mouse T cells. a**, Orthogonal chimeric receptor constructs were introduced into WT mouse T cells via YFP-encoding retroviral vectors. Expression levels of each receptor were assessed by YFP fluorescence using flow cytometry. **b**–**e**, Orthogonal receptor expressing (YFP⁺) C57BL/6 WT CD3⁺ T cells were stimulated with MSA fusions of MSA-oIL2 for 20 min. Cells were analysed for pSTAT using flow cytometry. Data are presented as the MFI of pSTAT from YFP⁺ T cells plotted against the $\log_{10}$ concentration of MSA-oIL2. **b**, pSTAT-1, -3, -4, -5, and -6 signalling dose-response curves of orthogonal IL-2Rβ ECD–ICD from o4R, o7R, o9R, and o21R transduced T cells (n = 2 biologically independent samples). **c**, pSTAT-1, -3, and -5 signalling dose-response curves of orthogonal IL-2Rβ ECD–ICD from oIFNAR2, oIFNGR1, and oIFNLR1 transduced T cells (n = 2 biologically independent samples). **d**, pSTAT-1, -4, and -5 signalling dose-response curves of o10R, o20R, and o22R transduced T cells (n = 2 biologically independent samples). **e**, pSTAT-1, -3, and -5 signalling dose-response curves of orthogonal IL-2Rβ ECD–ICD from oGCSFR and oEPOR transduced T cells (n = 3 biologically independent samples). All data represent mean ± s.e.m.

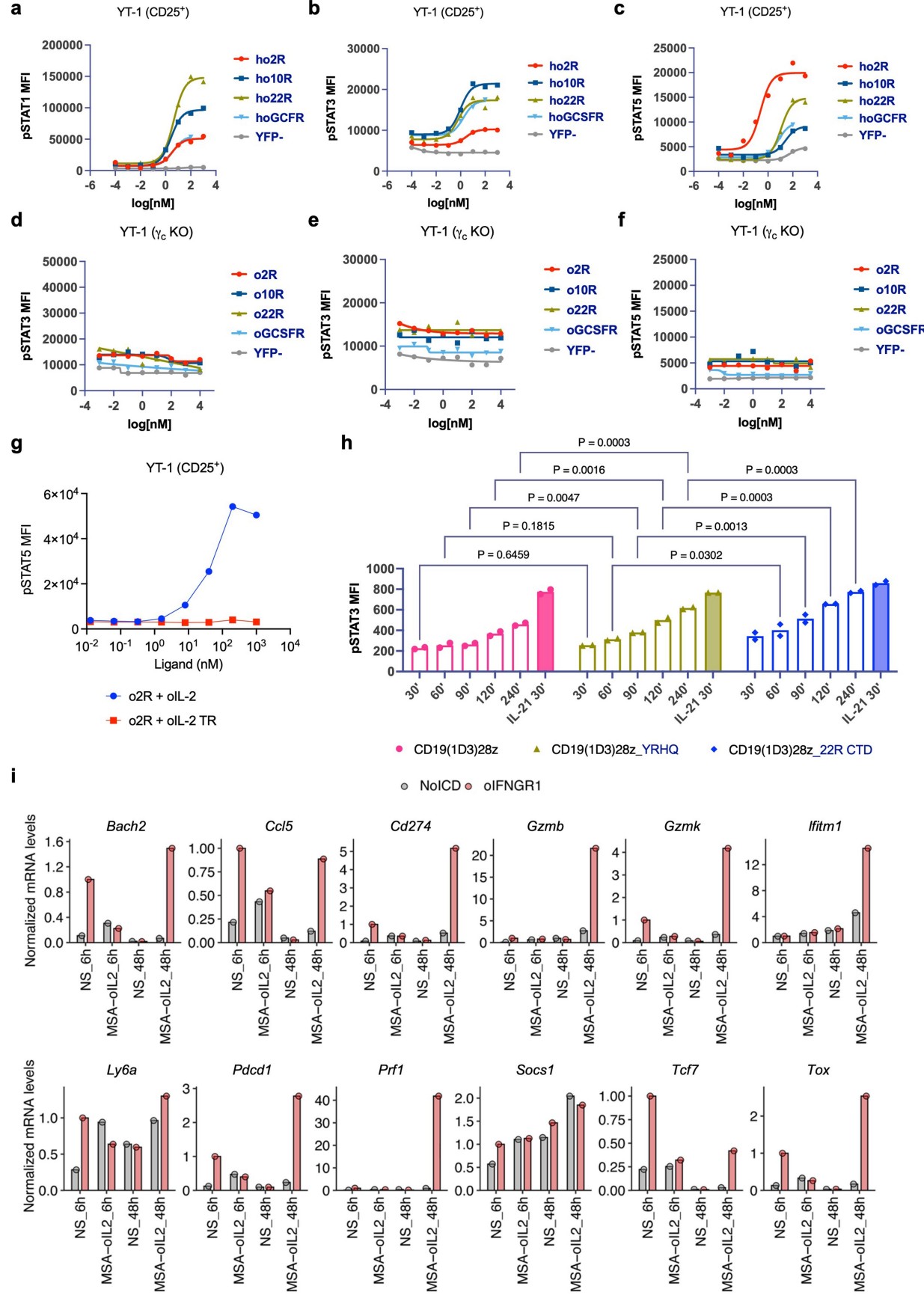

**Extended Data Fig. 2** | See next page for caption.

**Extended Data Fig. 2 | Orthogonal cytokine receptor system dependent on ligand-induced heterodimerization with $\gamma_c$. a–f**, Orthogonal chimeric receptors were introduced into YT-1 cells using YFP-encoding retroviral vectors. Cells were stimulated with MSA-human orthogonal IL2 (MSA-hoIL2) for 20 min and analysed for pSTAT via flow cytometry (n = 2 biological independent samples). Data are presented as the MFI of pSTAT from YFP$^+$ T cells, plotted against the $\log_{10}$ concentration of MSA-hoIL2. **a–c**, pSTAT1 (**a**), pSTAT3 (**b**), and pSTAT5 (**c**) signalling dose-response curves for human o2R (ho2R), ho10R, ho22R, and hoGCSFR-transduced YT-1 (CD25$^+$) cells. **d–f**, pSTAT1 (**d**), pSTAT3 (**e**), and pSTAT5 (**f**) signalling dose-response curves for ho2R, ho10R, ho22R, and hoGCSFR-transduced $\gamma_c$ knockout (KO) YT-1 cells. **g**, ho2R were transduced into CD25$^+$ YT-1 cells. Cells were stimulated with orthogonal IL-2 (oIL-2) or oIL-2-TR (carrying the Q126T and S130R $\gamma_c$ interface mutations) for 20 min and analysed for pSTAT via flow cytometry. Data are presented as the MFI of pSTAT from YFP$^+$ T cells. **h**, CD19 CAR T cells transduced with the indicated CAR-encoding construct (n = 2 biological independent samples). CD19_28z: a murine CD19 CAR (1D3 clone); CD19_28z_YRHQ: CD19 CAR with a STAT3-binding motif (YRHQ) at the C-terminus; CD19_28z_22 R CTD: CD19 CAR with the C-terminal domain (CTD) of IL-22R fused at the C-terminus. CAR T cells were stimulated with CD19$^+$ B cells or treated with IL-21 (50 ng/mL) served as a control. Shown are MFI of pSTAT3 over time. **i**, Normalized mRNA expression levels of indicated genes in mouse T cells transduced with o2R lacking ICD (NoICD) or oIFNGR1, following 6- or 48-hour stimulation (unstimulated [NS] or stimulated with MSA-oIL-2) (n = 1 biological independent sample). All data represent mean ± s.e.m. and are analysed by one-way ANOVA with Tukey's post-test (**h**).

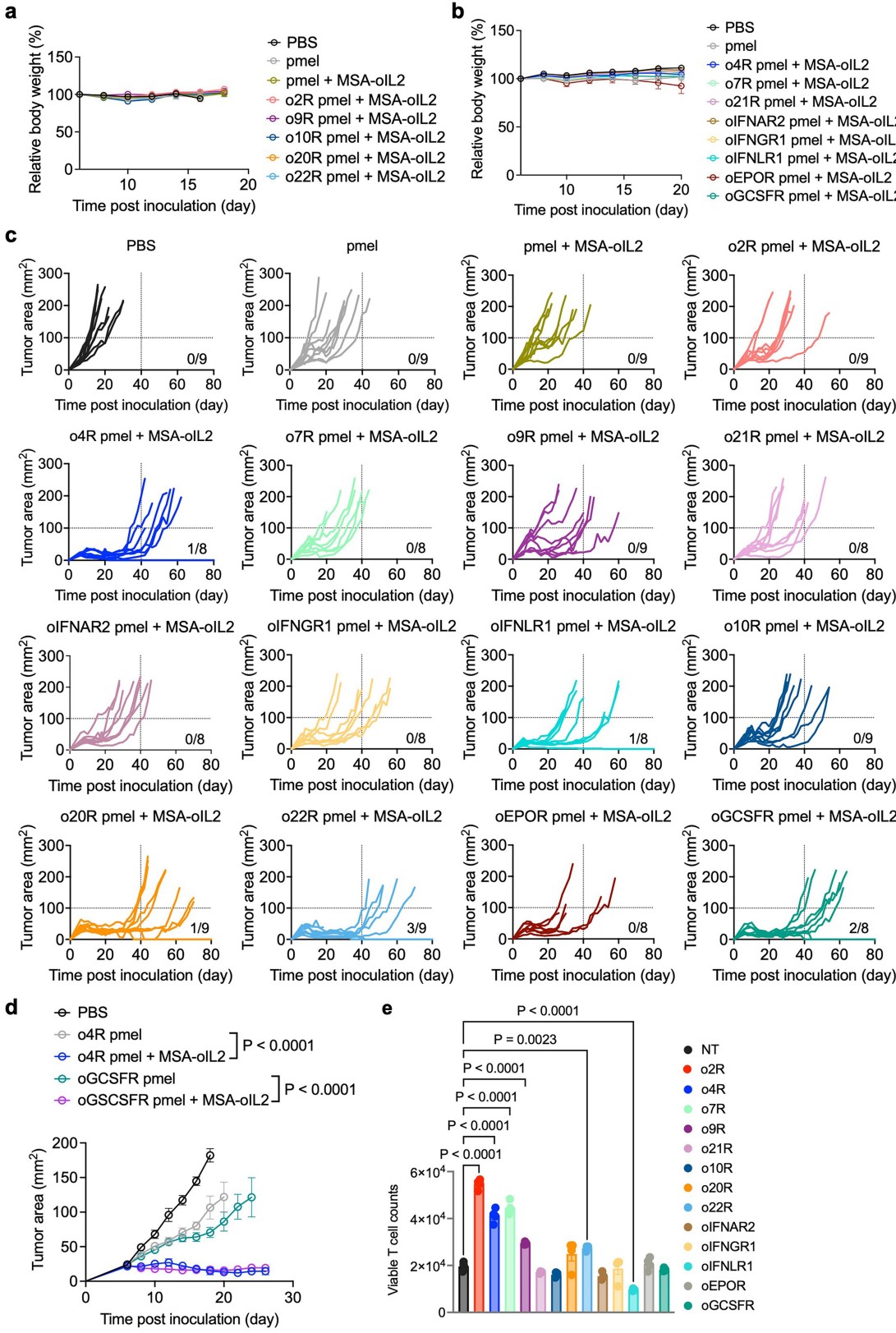

**Extended Data Fig. 3 | Orthogonal cytokine receptor signalling enhances antitumor efficacy of pmel T cells in B16F10 tumour model without causing toxicity. a,b,** Experimental setting is described in Fig. 2a. Relative body weight of mice post indicated treatment. **c,** Experimental setting is described in Fig. 2a. Shown are individual tumour growth curves. Indicated are numbers of tumour-free mice among the total number of mice in the group. **d,** Experimental setting is described in Fig. 2a (n = 5 animals). **e,** WT mouse T cells transduced with the indicated orthogonal chimeric receptor were cultured in MSA-oIL2 (100 nM) for two days (n = 4 biologically independent samples). Shown are viable T cell counts. Shown are average tumour growth curves. All data represent mean ± s.e.m. and are analysed by one-way (**e**) or two-way ANOVA with Tukey's post-test (**d**).

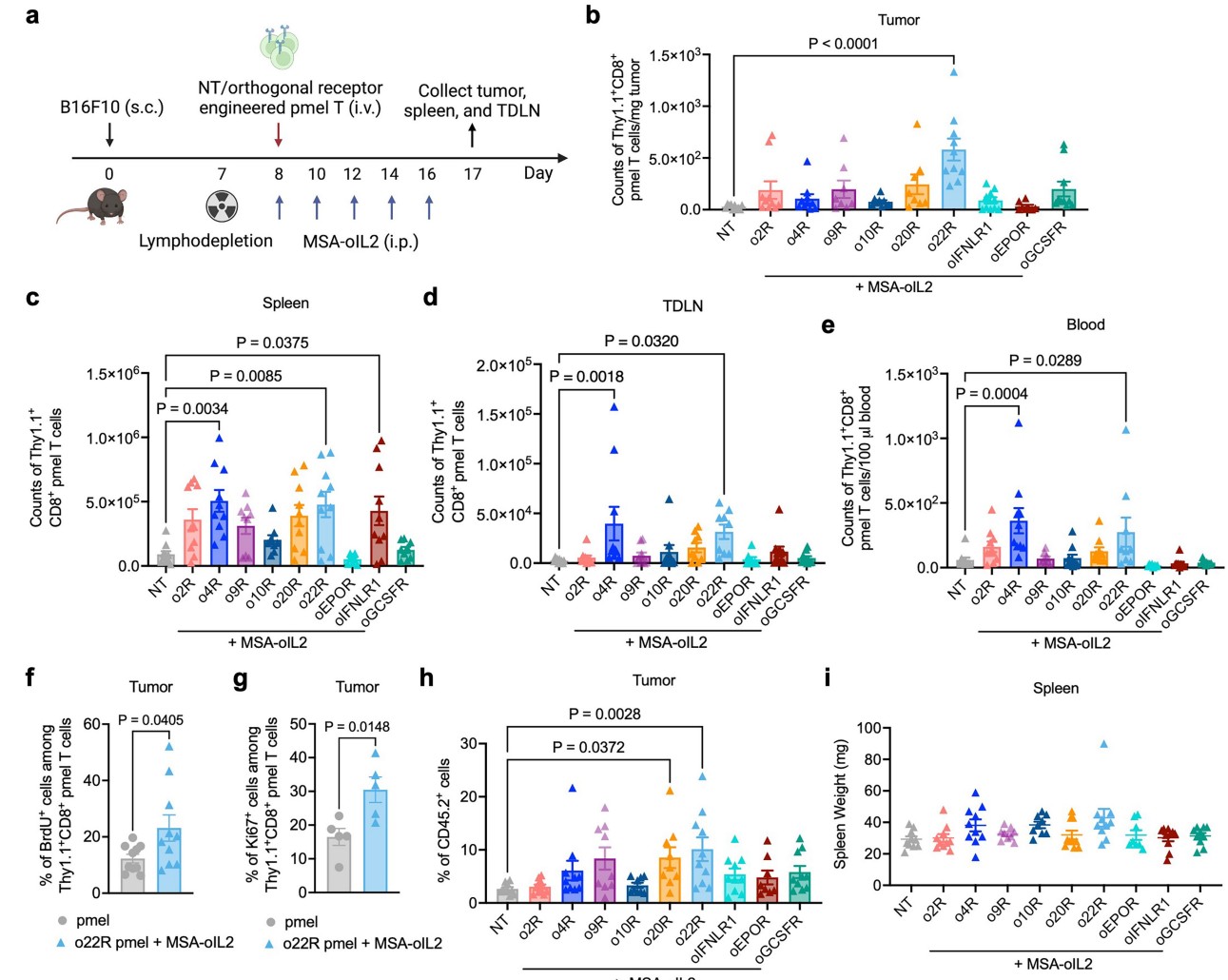

**Extended Data Fig. 4 | o22R pmel T cells exhibited the most prominent cell expansion in the B16F10 tumours. a–i**, C57BL/6 mice bearing B16F10 tumours were lymphodepleted on day 7. On day 8, mice received an i.v. adoptive transfer of pmel T cells by i.p. administration of MSA-oIL2 every other day. Mice received i.p. administration of Bromodeoxyuridine (BrdU) on day 16. On day 17, mice were sacrificed, indicated organs were collected for flow cytometry analysis. **a**, The experimental timeline. **b**, Counts of Thy1.1⁺CD8⁺ pmel T cells per mg tumour (n = 8 for oEPOR, o9R, o20R; n = 9 for o10R; n = 10 for all other groups). **c**, Counts of Thy1.1⁺CD8⁺ pmel T cells per spleen (n = 8 for oEPOR; n = 9 for o9R, o10R, o20R and oGCSFR; n = 10 for all other groups). **d**, Counts of Thy1.1⁺CD8⁺ pmel T cells per TDLN (n = 8 for oEPOR; n = 9 for o9R, o10R, o20R, and o22R;

n = 10 for all other groups). **e**, Counts of Thy1.1⁺CD8⁺ pmel T cells per 100 μl blood (n = 8 for oEPOR and o9R; n = 9 for o22R; n = 10 for all other groups). **f**, Frequencies of BrdU⁺ cells among Thy1.1⁺CD8⁺ pmel T cells in tumours (n = 10). **g**, Frequencies of Ki67⁺ cells among Thy1.1⁺CD8⁺ pmel T cells in tumours (n = 5). **h**, Frequencies of CD45.2⁺ cells in tumours (n = 8 for oEPOR; n = 9 for o9R, o10R, and o20R; n = 10 for all other groups). **i**, Spleen weight of mice from each treatment group (n = 8 for oEPOR; n = 9 for o9R, and o10R; n = 10 for all other groups). All data represent mean ± s.e.m. and are analysed by two-tailed Student's t-test (**f, g**) or one-way ANOVA with Tukey's post-test (**b–e, h, i**). The diagram in **a** was created using BioRender.

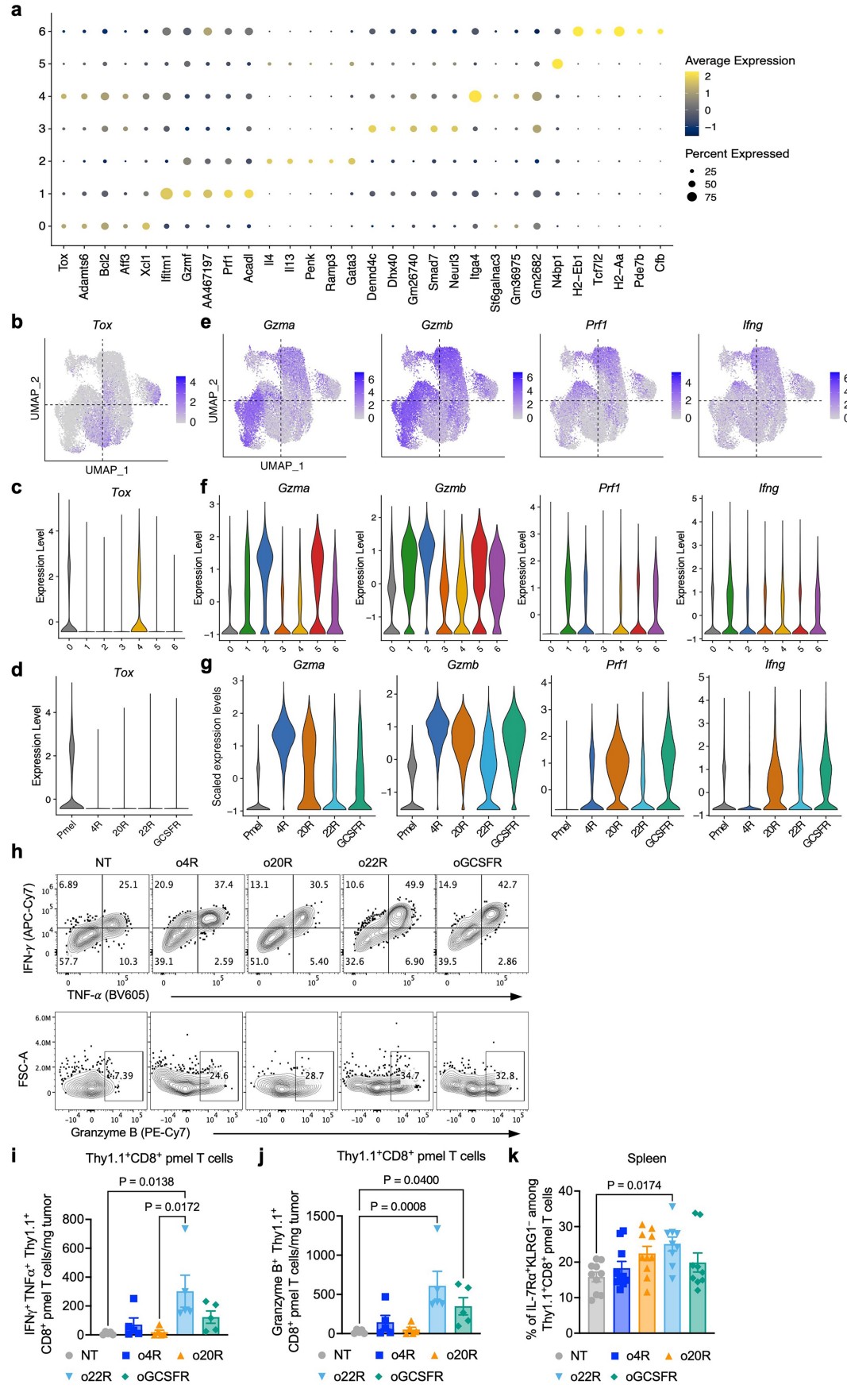

**Extended Data Fig. 5** | See next page for caption.

**Extended Data Fig. 5 | TILs with orthogonal chimeras exhibited enhanced cytotoxicity. a–k,** Experimental setting is described in Fig. 3a. **a,** Average expression levels of top-ranked genes across clusters. **b,** *Tox* expression over the UMAP. **c,** Violin plots showing expression levels of *Tox* cross clusters. **d,** Violin plots showing expression levels of *Tox* cross treatment groups. **e,** *Gzma, Gzmb, Prf1,* and *Ifng* expression over the UMAP. **f,** Violin plots showing expression levels of *Gzma, Gzmb, Prf1,* and *Ifng* cross clusters. **g,** Violin plots showing expression levels of *Gzma, Gzmb, Prf1,* and *Ifng* cross treatment groups. **h,** Representative flow cytometry plots showing IFN-γ⁺TNFα⁺ (top) and Granzyme B⁺ (bottom) subpopulations among Thy1.1⁺CD8⁺ pmel T cells in tumours. **i,** Counts of IFNγ⁺ TNFα⁺ Thy1.1⁺CD8⁺ T cells per mg tumour (n = 5 animals). **j,** Counts of Granzyme B⁺Thy1.1⁺CD8⁺ T cells per mg tumour (n = 5 animals). **k,** Frequencies of IL-7Rα⁺ KLRG1⁻ cells among Thy1.1⁺CD8⁺ pmel T cells in spleens (n = 8 for oEPOR; n = 9 for o9R, o10R, and oGCSFR; n = 10 for all other groups). All data represent mean ± s.e.m. and are analysed by one-way ANOVA with Tukey's post-test (**i–k**).

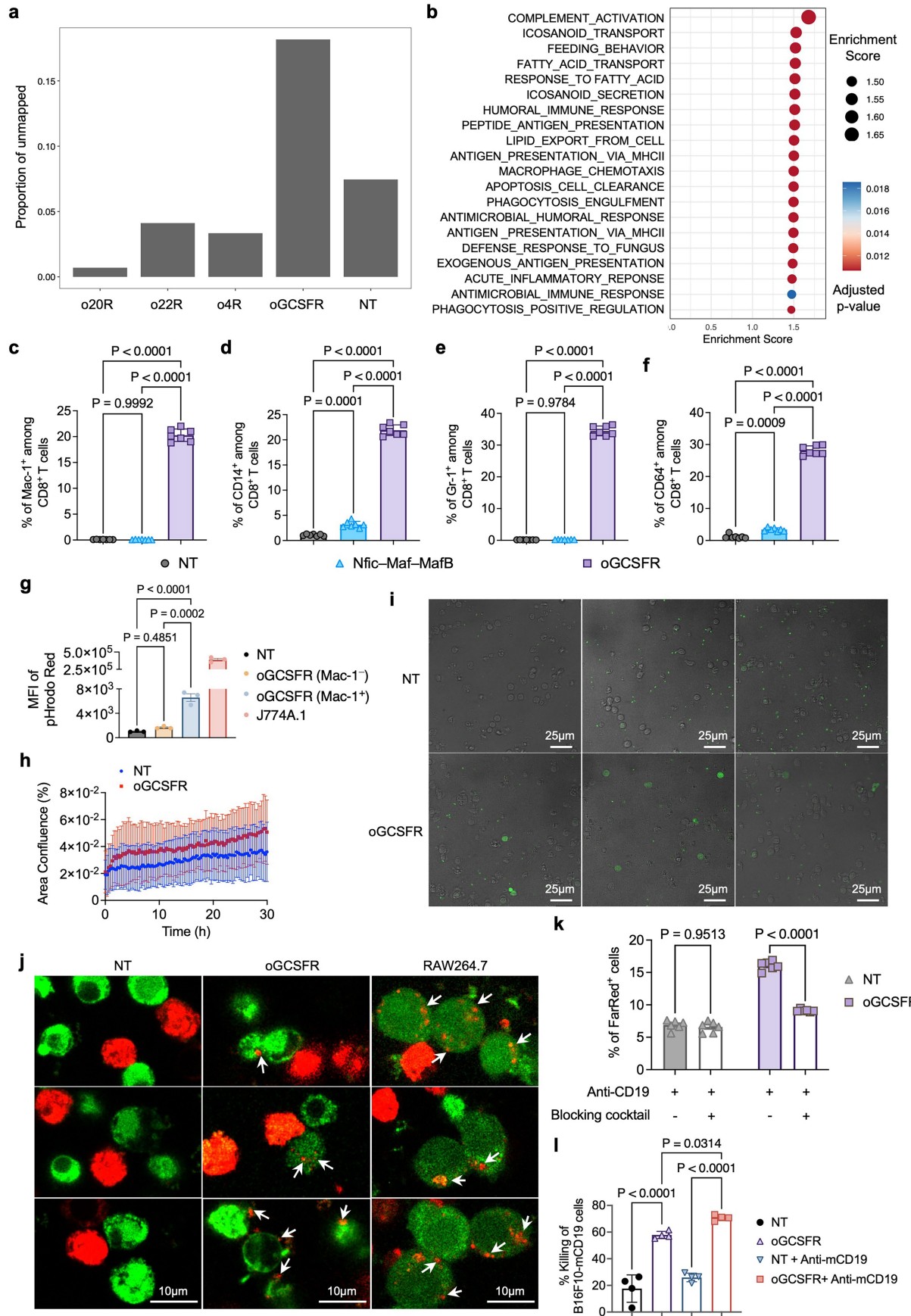

**Extended Data Fig. 6** | See next page for caption.

**Extended Data Fig. 6 | Orthogonal GCSFR drives myeloid-like T cell states.**
**a**, Proportion of each treatment group unmapped to the reference atlas of TILs[29]. **b**, Gene Ontology enrichment analysis of oGCSFR-upregulated genes identified the top 20 enriched pathways (enrichment score ≥1.2, adjusted p < 0.05). **c–f**, NT pmel CD8[+] T cells and those transduced with Nfic–Maf–MafB, or oGSCFR were cultured for 72 h, with oGSCFR pmel CD8[+] T cells stimulated with MSA-oIL2 (500 nM) during this period, followed by flow cytometry analysis (n = 7 biological independent samples). Shown are the frequencies of Mac-1[+] (**c**), CD14[+] (**d**), CD64[+] (**e**), and Gr-1[+] (**f**) cells. **g**, MFI of pHrodo Red E. coli bioparticles (n = 3 biologically independent samples). **h**, Shown are the pHrodo Red confluences (n = 8 biologically independent samples). **i**, The experimental setup is described in Fig. 4l. Shown are the representative fluorescence images of GFP[+] L. *monocytogenes* at the highest concentration. **j**, Experimental setting

is described in Fig. 4n,o. Representative fluorescence images of CSFE[+] (green) effector cells cocultured with FarRed[+] (red) A20 target cells at a 1:1 E:T ratio. **k**, Pmel CD8[+] T cells were cocultured for 2 h with FarRed[+] A20 cells pretreated with mouse CD19 (mCD19) antibody, followed by flow cytometry analysis (n = 6 biologically independent samples). Shown are the frequencies of FarRed[+]CD8[+] cells. **l**, oGCSFR pmel CD8[+] T cells were preserved in MSA-oIL2 (500 nM) for 24 h. B16F10-mCD19 cells were co-cultured with pmel CD8[+] T cells at a 1:1 E:T ratio with MSA-hoIL2 (200 nM), in the presence or absence of mCD19 antibodies for 48 h. Cells were collected for flow cytometry analysis (n = 4 biologically independent samples). Shown are the percentages of B16F10-mCD19 cell killing. All data represent mean ± s.e.m. and are analysed by two-tailed Student's t-test (**k**) or one-way ANOVA with Tukey's post-test (**c–f**, **g**, **l**).

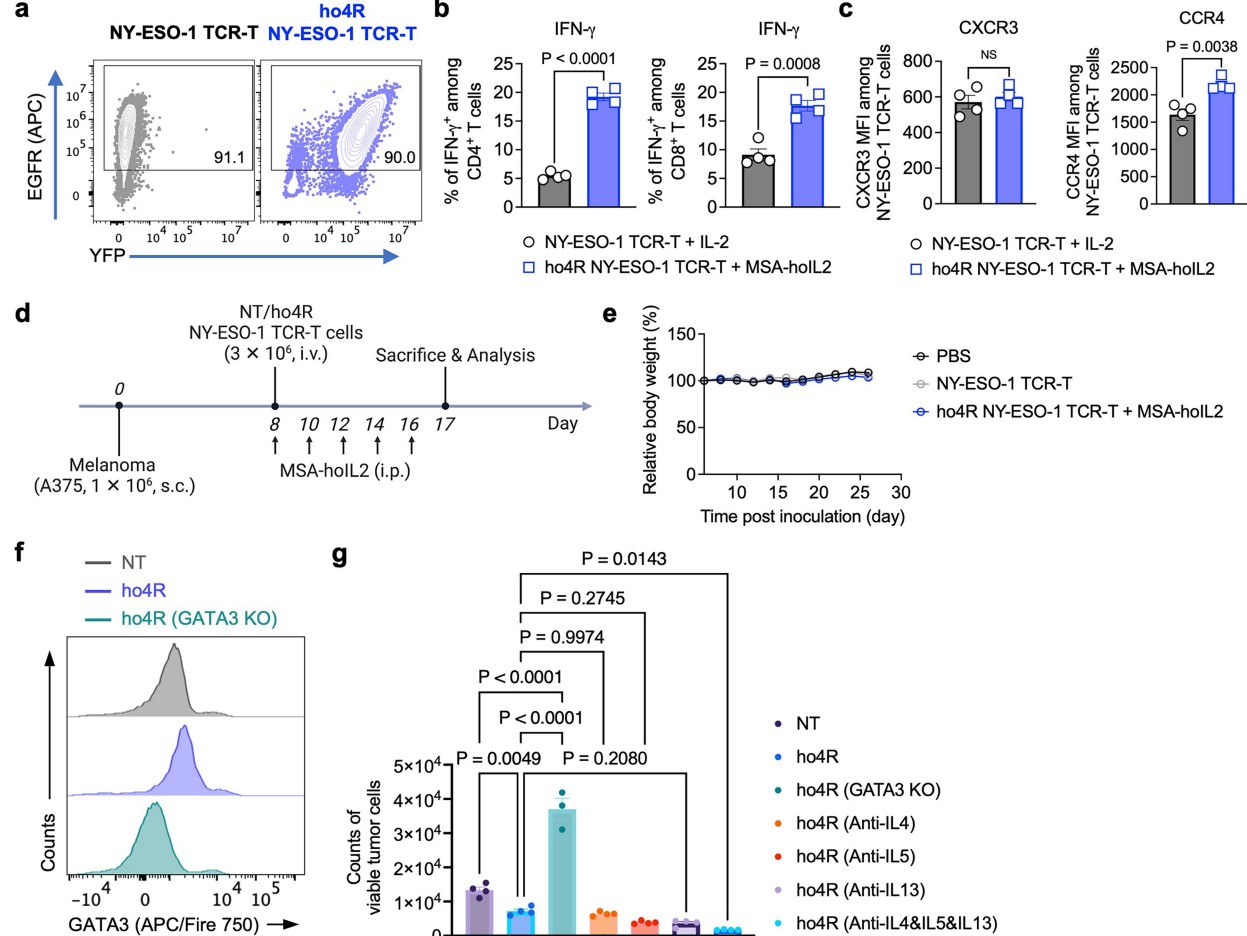

**Extended Data Fig. 7 | Human orthogonal IL-4R enhances type 2 function and antitumor activity. a**, NY-ESO-1 TCR and ho4R constructs were introduced via retroviral vectors. Expression levels of NY-ESO-1 TCR and ho4R were assessed by EGFR and YFP expression using flow cytometry, respectively. Shown are representative flow cytometry plots of transduction efficiency. **b**, Frequencies of IFN-γ[+] subpopulation among CD4[+] (left) and CD8[+] (right) NY-ESO-1 TCR-T cells (n = 4 biologically independent samples). **c**, MFI of CXCR3 (left) and CCR4 (right) among CD3[+] NY-ESO-1 TCR-T cells (n = 4 biologically independent samples). **d**, The experimental timeline for Fig. 5k. **e**, Experimental setting is described in

Fig. 5i. Relative body weight of mice post indicated treatment. **f**, Representative flow cytometry histograms showing GATA-3 expression level. **g**, NT or ho4R NY-ESO-1 TCR-T cells (unmodified, GATA3 knockout, or treated with the indicated antibodies) were co-cultured with A375 cells at an E:T ratio of 1:2 for 48 h (n = 3 biologically independent samples). Shown are the counts of viable A375 tumour cells. All data represent mean ± s.e.m. and are analysed by two-tailed Student's t-test (**b**,**c**) or one-way ANOVA with Tukey's post-test (**g**). The diagram in **d** was created using BioRender.

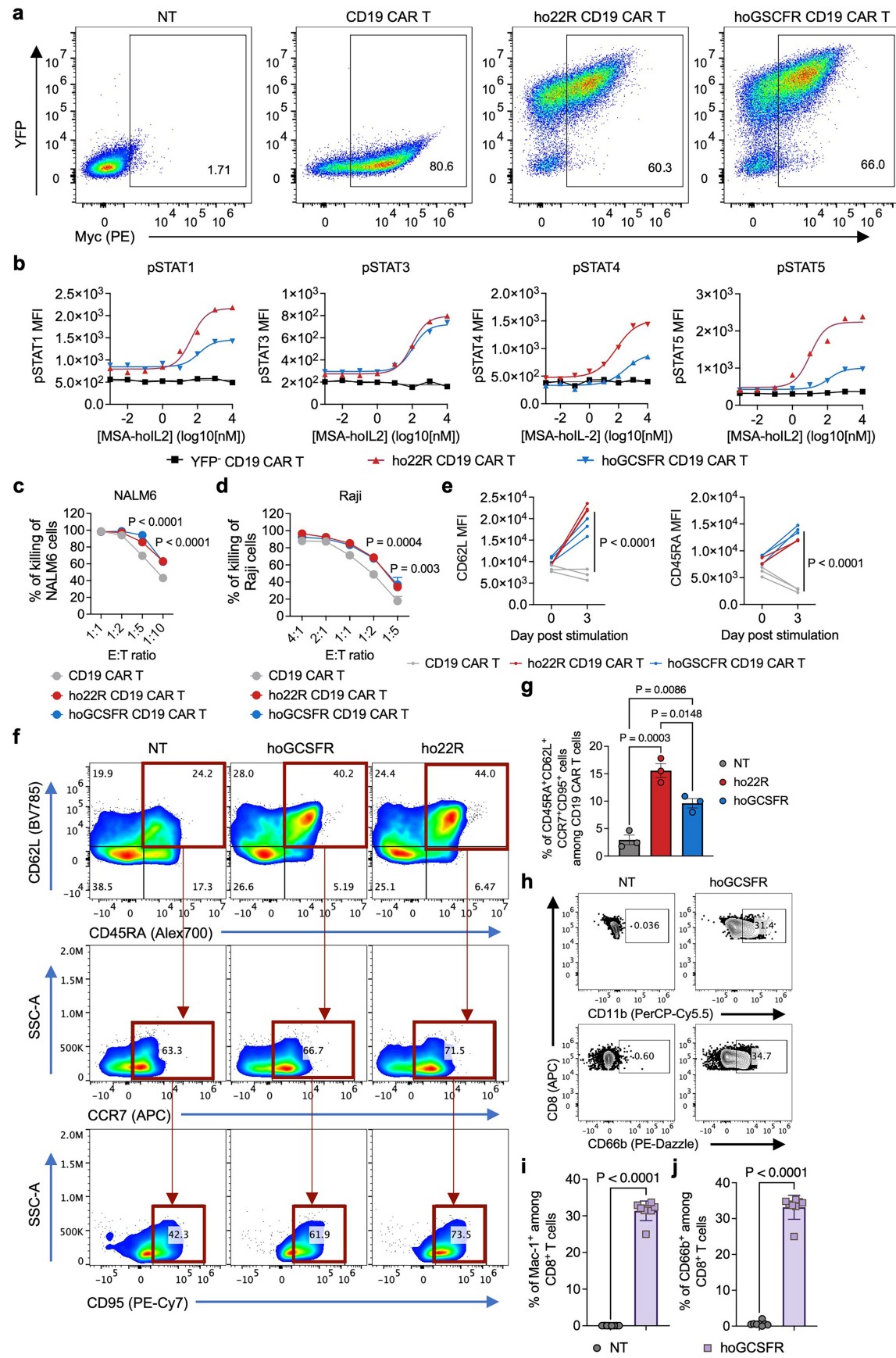

**Extended Data Fig. 8** | See next page for caption.

**Extended Data Fig. 8 | ho22R and hoGCSFR signalling enhance the antitumor potency of CD19 CAR-T cells and promote T cell stemness. a**, CD19 CAR, ho22R, hoGCSFR constructs were introduced via retroviral vectors. Expression levels of CD19 CAR were measured by Myc-tag fluorescence using flow cytometry. Expression levels of ho22R and hoGCSFR were measured by YFP fluorescence using flow cytometry. Shown are representative flow cytometry plots of transduction efficiency. **b**, pSTAT-1, -3, -4, and -5 signalling dose-response curves (n = 2 biologically independent samples). Data are presented as the MFI of pSTAT against the $\log_{10}$ concentration of MSA-hoIL2. **c,d**, CD19 CAR T, ho22R CD19 CAR T, and hoGCSFR CD19 CAR T cells were cocultured with NALM6-Luciferase (NALM6-Luc) cells or Raji cells at different E:T ratios in the presence of MSA-hoIL2 (100 nM) for 24 h (n = 3 biologically independent samples). Shown are the percentages of killing of NALM6-Luc cells (**c**) and Raji cells (**d**).

**e**–**g**, CD19 CAR T, ho22R CD19 CAR T, and hoGCSFR CD19 CAR T cells were cultured in MSA-hoIL2 (100 nM) supplemented T cell medium for 3 days (n = 3 biologically independent samples). **e**, Shown are MFI of CD62L and CD45RA among CD19 CAR T cells. **f**, Representative flow cytometry plots showing the gating strategy for CD45RA$^+$ CD62L$^+$ CCR7$^+$ CD95$^+$ subpopulations in NT, hoGCSFR, and ho22R CD3$^+$ NY-ESO-1 TCR-T cells. **g**, Frequencies of CD45RA$^+$ CD62L$^+$ CCR7$^+$ CD95$^+$ cells. **h**–**j**, NT and oGCSFR CD8$^+$ human T cells were stimulated with MSA-hoIL2 (500 nM) for 72 h, followed by flow cytometry analysis (n = 6 biologically independent samples). **h**, Representative flow cytometry plots of Mac-1$^+$ and CD66b$^+$ cells among CD8$^+$ T cells. **i,j**, Shown are the frequencies of Mac-1$^+$ (**i**) and CD66b$^+$ (**j**) cells among CD8$^+$ T cells. All data represent mean ± s.e.m. and are analysed by two-tailed Student's t-test (**i,j**), one-way ANOVA (**g**), or two-way (**c**–**e**) ANOVA with Tukey's post-test.

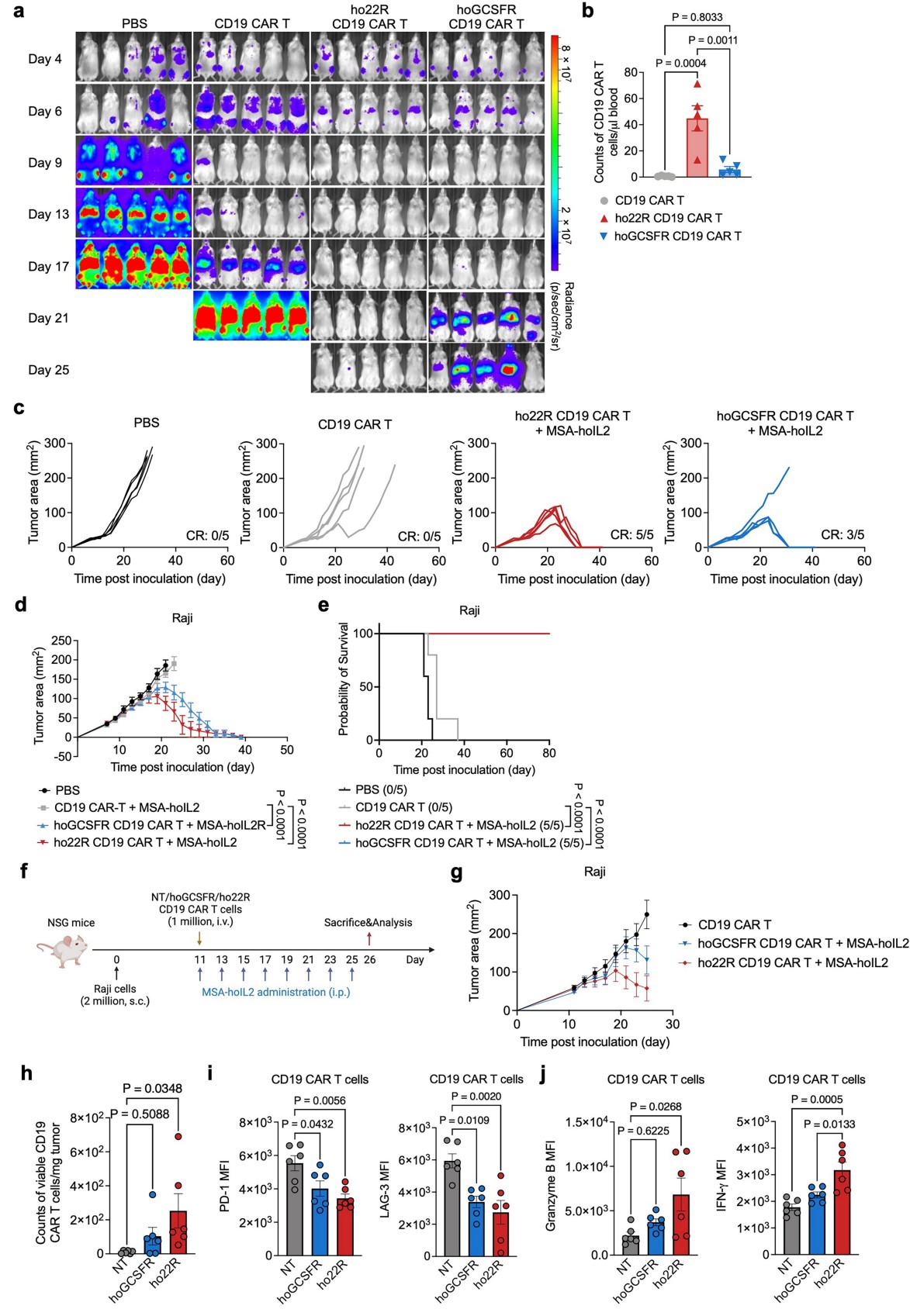

**Extended Data Fig. 9** | See next page for caption.

**Extended Data Fig. 9 | ho22R and hoGCSFR signalling enhances the anti-tumour potency of CD19 CAR T cells against haematological malignances.**
**a**, Experimental setting is described in Fig. 6a. Show are bioluminescence images. **b**, Experimental setting is described in Fig. 6a. Counts of CD19 CAR T cells per µl blood (n = 4 animals). **c**, Experimental setting is described in Fig. 6c. Individual tumour growth curves. **d**,**e**, NSG male mice were inoculated (s.c.) with Raji cells ($1 \times 10^6$) and received an i.v. adoptive transfer of $1 \times 10^6$ NT, ho22R, or hoGCSFR CD3$^+$ CD19 CAR T cells on Day 7, followed by i.p. administration of MSA-hoIL2 ($2.5 \times 10^4$ unit) or PBS every other day until day 29 (n = 5 animals). **d**, Average tumour growth curves. **e**, Survival curves. Indicated are the numbers of tumour-free mice per total number of mice in each group. **f–j**, NSG female mice were inoculated (s.c.) with Raji cells ($2 \times 10^6$) and received an i.v. adoptive transfer of $1 \times 10^6$ NT, ho22R, or hoGCSFR CD3$^+$ CD19 CAR T cells on Day 11, followed by i.p. administration of MSA-hoIL2 ($2.5 \times 10^4$ unit) or PBS every other day until day 25 (n = 6 animals). On day 26, the mice were sacrificed, and tumours were collected for flow cytometry analysis. **f**, The experimental timeline. **g**, Average tumour growth curves. **h**, Counts of CD3$^+$ Myc$^+$ CD19 CAR T cells in tumours. **i**, MFI of PD-1 (left) and LAG-3 (right) among CD3$^+$ Myc$^+$ CD19 CAR T cells in tumours. **j**, MFI of Granzyme B (left) and IFN-γ (right) among CD3$^+$ Myc$^+$ CD19 CAR T cells in tumours. All data represent mean ± s.e.m. and are analysed by one-way (**b**, **h–j**), or two-way (**d**) ANOVA with Tukey's post-test, or log-rank (Mantel-Cox) test (**e**). The diagram in **f** created using BioRender.com.

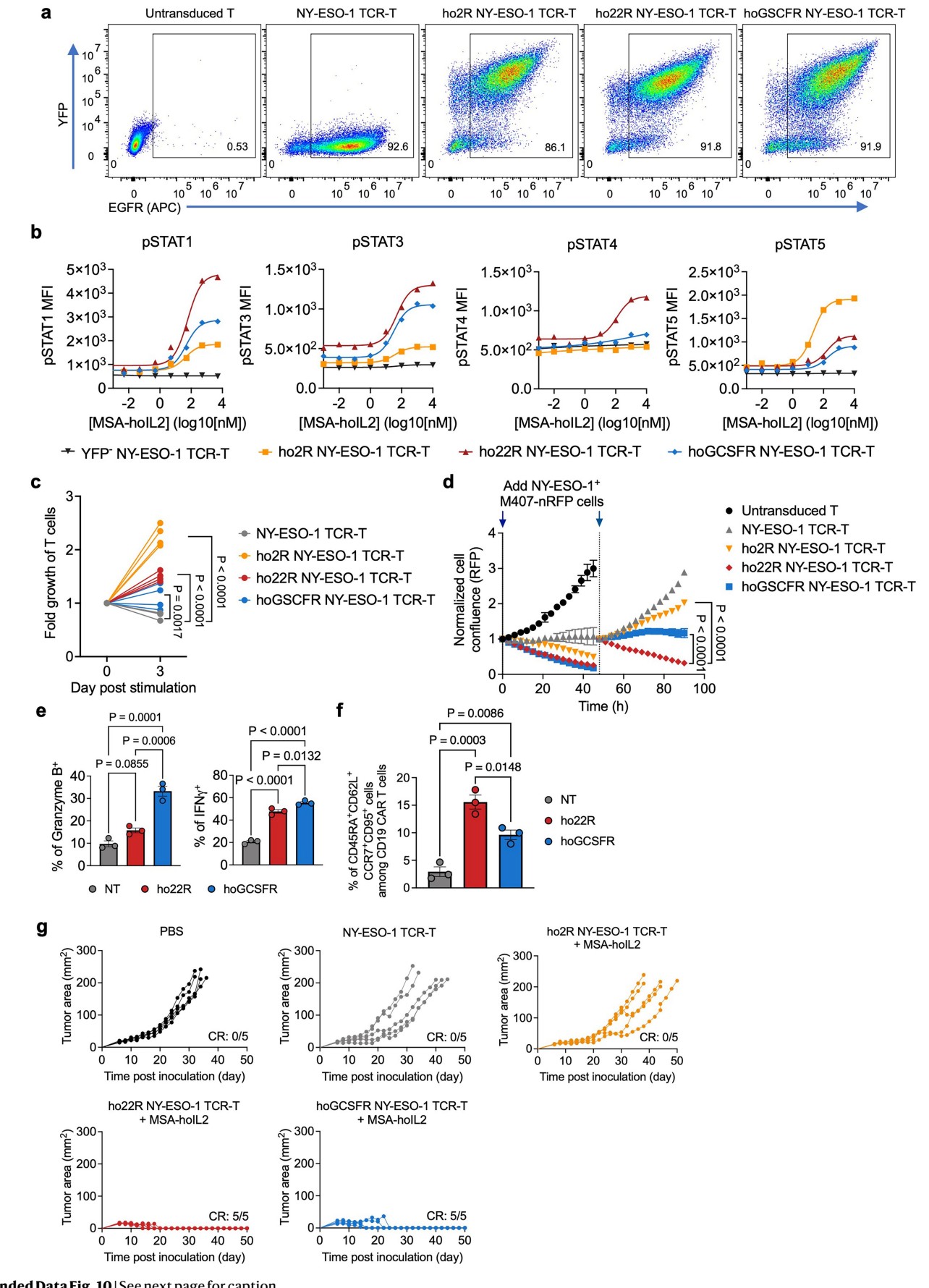

**Extended Data Fig. 10** | See next page for caption.

**Extended Data Fig. 10 | ho22R and hoGCSFR NY-ESO-1 TCR-T cells exhibit enhanced antitumor efficacy in HLA*0201+ NY-ESO-1+ A375 xenograft tumour model. a**, NY-ESO-1 TCR, ho2R, ho22R, and hoGCSFR constructs were introduced via retroviral vectors. Expression levels were measured by EGFR (NY-ESO-1 TCR) or YFP fluorescence (ho2R, ho22R, and hoGCSFR) using flow cytometry. Shown are representative flow cytometry plots of transduction efficiency. **b**, pSTAT-1, -3, -4, and -5 signalling dose-response curves (n = 2 biologically independent samples). Data are presented as the MFI of pSTAT against the $\log_{10}$ concentration of MSA-hoIL2. **c**, Fold growth of CD3+ NY-ESO-1 TCR-T cells post 3-day culture in the presence of MSA-hoIL2 (100 nM) (n = 4 biologically independent samples). **d–f**, NT T cells, NY-ESO-1 TCR-T cells, or NY-ESO-1 TCR-T cells transduced with ho2R, ho22R, or hoGCSFR were cocultured at a 1:1 E:T ratio with the HLA*0201+ NY-ESO-1+ melanoma cell line (nRFP-M407)

in the presence of MSA-hoIL2 (100 nM). nRFP-M407 cells were reintroduced after 48 h of coculture (blue arrows) in the presence of MSA-hoIL2 (100 nM). After three rounds of tumour challenge, cells were collected for flow cytometry analyses. **d**, Shown is the normalized tumour cell confluence (n = 4 biologically independent samples). **e**, Frequencies of Granzyme B+ (left) and IFN-γ+ (right) among CD3+ NY-ESO-1 TCR-T cells (n = 3 biologically independent samples). **f**, Frequencies of CD45RA+CD62L+CCR7+CD95+ among CD3+ NY-ESO-1 TCR-T cells (n = 3 biologically independent samples). **g**, Experimental setting is described in Fig. 6f. Shown are the individual tumour growth curves. Indicated are the numbers of tumour-free mice per total number of mice in each group. All data represent mean ± s.e.m. and are analysed by one-way ANOVA (**e**, **f**) or two-way (**c**, **d**) ANOVA with Tukey's post-test.

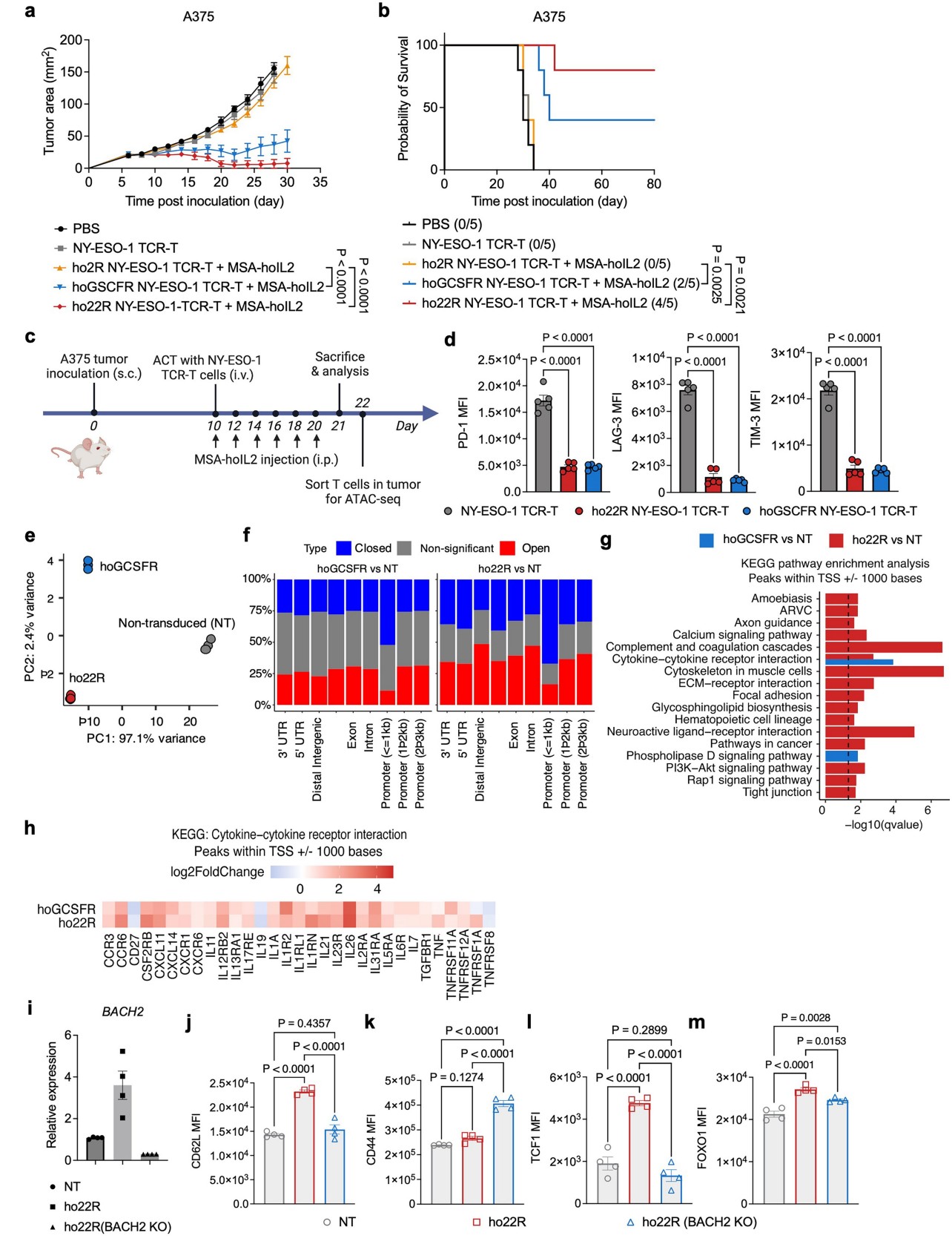

**Extended Data Fig. 11 | ho22R and hoGCSFR epigenetically rewire TILs in A375 xenograft tumour model. a,b**, Male NSG mice bearing A375 tumours e received an i.v. adoptive transfer of $3 \times 10^6$ CD3$^+$ NY-ESO-1 TCR-T cells on Day 6, followed by i.p. administration of MSA-hoIL2 every other day (n = 5 animals). **a**, Tumour growth curves. **b**, Survival curves. Indicated are the numbers of tumour-free mice per total number of mice. **c**, The experimental timeline for Fig. 6(h–n) and Extended Data Fig. 11d–h. Female NSG mice bearing A375 melanoma received an i.v. adoptive transfer of $3 \times 10^6$ CD3$^+$ NY-ESO-1 TCR-T cells on day 10, followed by i.p. administration of MSA-hoIL2 every other day. On day 21, the tumours were analysed by flow cytometry (n = 5 animals). On day 22, TCR-T cells in tumours were sorted for ATAC-seq analysis (n = 3 biological independent samples). **d**, MFI of PD-1, LAG–3, and TIM-3 among EGFR$^+$CD8$^+$ TCR-T cells in tumours (n = 5 animals). **e**, Principal component analysis (PCA).

**f**, Proportions of differential or non-significant chromatin regions. **g**, KEGG pathway enrichment was performed on accessible promoter regions (TSS ± 1 kb) using one-tailed hypergeometric tests, with FDR-adjusted q values (Benjamini-Hochberg). **h**, Genes underlying the enrichment for the "KEGG cytokine-cytokine receptor interaction" term (differential chromatin regions within transcription start sites +/−1 kb of known genes). **i–m**, NT, ho22R, and ho22R (BACH2 knockout) CD3$^+$ human T cells were restimulated with soluble CD3 antibodies (1 μg/mL) in the presence of MSA-hoIL2 (200 nM) for 48 h (n = 4). Cells were then collected for quantitative PCR and flow cytometry analysis. **i**, Relative expression of *BACH2* (n = 4 biologically independent samples). **j–m**, MFI of CD62L (**j**), CD44 (**k**), TCF-1 (**l**), FOXO1 (**m**). All data represent mean ± s.e.m. and are analysed by one-way ANOVA (**d**, **j–m**), or two-way (**a**) ANOVA with Tukey's post-test, or log-rank (Mantel-Cox) test (**b**). The diagram in **c** created using BioRender.com.

# Reporting Summary

## Statistics

For all statistical analyses, confirm that the following items are present in the figure legend, table legend, main text, or Methods section.

| n/a | Confirmed | |
|---|---|---|
| ☐ | ☒ | The exact sample size (*n*) for each experimental group/condition, given as a discrete number and unit of measurement |
| ☐ | ☒ | A statement on whether measurements were taken from distinct samples or whether the same sample was measured repeatedly |
| ☐ | ☒ | The statistical test(s) used AND whether they are one- or two-sided *Only common tests should be described solely by name; describe more complex techniques in the Methods section.* |
| ☐ | ☒ | A description of all covariates tested |
| ☐ | ☒ | A description of any assumptions or corrections, such as tests of normality and adjustment for multiple comparisons |
| ☐ | ☒ | A full description of the statistical parameters including central tendency (e.g. means) or other basic estimates (e.g. regression coefficient) AND variation (e.g. standard deviation) or associated estimates of uncertainty (e.g. confidence intervals) |
| ☐ | ☒ | For null hypothesis testing, the test statistic (e.g. *F*, *t*, *r*) with confidence intervals, effect sizes, degrees of freedom and *P* value noted *Give P values as exact values whenever suitable.* |
| ☒ | ☐ | For Bayesian analysis, information on the choice of priors and Markov chain Monte Carlo settings |
| ☒ | ☐ | For hierarchical and complex designs, identification of the appropriate level for tests and full reporting of outcomes |
| ☒ | ☐ | Estimates of effect sizes (e.g. Cohen's *d*, Pearson's *r*), indicating how they were calculated |

*Our web collection on statistics for biologists contains articles on many of the points above.*

## Software and code

Policy information about availability of computer code

| | |
|---|---|
| Data collection | Flow cytometry data collection was performed with CytoFlex (Beckman Coulter). In vivo bioluminescence data collection was performed using a IVIS fluorescence/bioluminescence imaging system (Xenogen). Fluorescence imaging data collection was performed using a using a Elyra7 lattice SIM microscope (ZEISS) and Leica TCS SP8 Confocal with WLL. Bulk RNA sequencing data was collected on NovaSeq X Plus Series (PE150). scRNA sequencing data was collected on a NovaSeq 6000. Bulk ATAC sequencing data was collected using NovaSeqX sequencer. Incucyte data was collected on using the IncuCyte live imaging system (Essen Bioscience). |
| Data analysis | Flow cytometry data were analyzed using Flowjo (version 10.10.0). Statistical analysis was performed using Graphpad Prism (version 9). For bulk RNA sequencing, reads were aligned to the mouse reference genome (mm10) with Rsubread (version 2.18.0). Gene expression was quantified with featureCounts, and DESeq2 (version 1.48.1) was used for this downstream analysis. scRNA sequencing raw data were processed by cellranger mkfastq from 10x Genomics (version 7.1.0) using a custom reference package based on mouse reference genome mm10. The gene expression matrix was processed and analyzed using Seurat (version 5.1.0). Transcription factor (TF) activity prediction was conducted with the pySCENIC (version 0.12.1) docker distribution with default parameter settings. Intercellular communication prediction was performed using LIANA (version 1.5.1). Bulk ATAC sequencing were processed using RSubread (version 2.18.0) using reference genome (hg19). The aligned reads were subjected to peak calling with MACS3 (version 3.0.1). Downstream analyses were performed in R (version 4.4). Peaks were annotated with their genomic locations and associated genes using ChIPseeker (version 1.44.0). The reads aligned with each region were then counted with the summarizeOverlaps function, and the count matrix was analyzed with DESeq2 (version 1.48.1). Differential transcription factor (TF) motif enrichment analysis was performed using chromVAR (version 1.30.1). Cis-regulatory element enrichment analysis were performed with GREAT (version 2.10.0). The R code used to analyze the scRNA-seq data is publicly at Zenodo (DOI: 10.5281/ zenodo.15702063). Analysis details are provided in the Methods section. |

For manuscripts utilizing custom algorithms or software that are central to the research but not yet described in published literature, software must be made available to editors and reviewers. We strongly encourage code deposition in a community repository (e.g. GitHub). See the Nature Portfolio guidelines for submitting code & software for further information.

## Data

Policy information about availability of data

All manuscripts must include a data availability statement. This statement should provide the following information, where applicable:
- Accession codes, unique identifiers, or web links for publicly available datasets
- A description of any restrictions on data availability
- For clinical datasets or third party data, please ensure that the statement adheres to our policy

The raw and processed bulk RNA-seq data are publicly available at GEO under accession code GSE282973 (C57BL/6 wild-type T cells; aligned to the mm10 mouse reference genome). The scRNA-seq data are available at GSE272444 (TCR transgenic pmel T cells in melanoma tumors; mm10 genome). ATAC-seq data generated in this study have been deposited at GEO under accession code GSE272385 (human T cells in melanoma tumors grown in NSG mice; aligned to the hg19 human reference genome). Normalized gene expression matrices and associated sample metadata were downloaded from the Gene Expression Omnibus (https://www.ncbi.nlm.nih.gov/geo/) using the accession numbers above. All other raw data are available on Figshare (DOI: 10.6084/m9.figshare.26322181). Additional supporting information is available from the corresponding author upon reasonable request.

## Research involving human participants, their data, or biological material

Policy information about studies with human participants or human data. See also policy information about sex, gender (identity/presentation), and sexual orientation and race, ethnicity and racism.

| | |
|---|---|
| Reporting on sex and gender | Gender information was not collected. |
| Reporting on race, ethnicity, or other socially relevant groupings | N/A |
| Population characteristics | Human T cells were isolated from buffy coats from anonymous healthy donors (male and female; gender information was not collected) purchased from the Stanford Blood Center. |
| Recruitment | Anonymous healthy donors were recruited by the Stanford Blood Center. |
| Ethics oversight | Ethical approval pertaining to T cell donors was obtained by the Stanford Blood Center. |

Note that full information on the approval of the study protocol must also be provided in the manuscript.

# Field-specific reporting

Please select the one below that is the best fit for your research. If you are not sure, read the appropriate sections before making your selection.

☒ Life sciences          ☐ Behavioural & social sciences          ☐ Ecological, evolutionary & environmental sciences

For a reference copy of the document with all sections, see nature.com/documents/nr-reporting-summary-flat.pdf

# Life sciences study design

All studies must disclose on these points even when the disclosure is negative.

| | |
|---|---|
| Sample size | Group sizes for in vivo validation experiments were selected empirically based on previous results of the intragroup variation of tumor growth upon similar treatments. Similarly, group sizes in vitro were selected on the basis of prior knowledge of variation. Ref: Nature Biotechnology volume 42, pages1693–1704 (2024). doi: 10.1038/s41587-023-02060-8. |
| Data exclusions | Rout outlier tests were run with default parameters (Q = 1%) in Prism on all mouse experimental data due to inherent variability within the model system. |
| Replication | All presented results were repeatable. Replicates were used in all experiments as noted in figure captions or methods. |
| Randomization | Age and sex-matched animals were used for each experiment. Mice were randomized prior to treatment. In the in vitro experiments, samples with same pretreatment conditions were randomly assigned to a treatment group. |
| Blinding | No blinding was performed due to requirements for cage labeling and staffing needs. |

# Reporting for specific materials, systems and methods

We require information from authors about some types of materials, experimental systems and methods used in many studies. Here, indicate whether each material, system or method listed is relevant to your study. If you are not sure if a list item applies to your research, read the appropriate section before selecting a response.

## Materials & experimental systems

| n/a | Involved in the study |
|---|---|
| ☐ | ☒ Antibodies |
| ☐ | ☒ Eukaryotic cell lines |
| ☒ | ☐ Palaeontology and archaeology |
| ☐ | ☒ Animals and other organisms |
| ☒ | ☐ Clinical data |
| ☒ | ☐ Dual use research of concern |
| ☒ | ☐ Plants |

## Methods

| n/a | Involved in the study |
|---|---|
| ☒ | ☐ ChIP-seq |
| ☐ | ☒ Flow cytometry |
| ☒ | ☐ MRI-based neuroimaging |

# Antibodies

**Antibodies used**

The following antibodies or staining reagents were purchased from BioLegend: mouse CD16/32 (93, 101302), mouse CD90.1/Thy-1.1 (OX-7, 202533), mouse CD8b (YTS156.7.7, 126606), mouse CD8b (YTS156.7.7, 126614), mouse CD45.2 (104, 109808), mouse Ki-67 (16A8, 652413), mouse SCA-1 (D7,108142), mouse CD44 (IM7, 103030), mouse CD62L (MEL-14, 104428), mouse PD-1 (29F.1A12, 135220), mouse IFNγ (XMG1.2, 505850), mouse TNF-α (MP6-XT22, 506329), mouse/human Granzyme B (QA16A02, 372214), mouse/human KLRG1 (2F1/KLRG1, 138419), mouse IL-7Rα (A7R34, 135022), mouse CD122 (TM-β1, 123210), mouse SLAMF6 (330-AJ, 134606), mouse CD11b (M1/70, 101228), mouse CD14 (Sa14-2, 123335), mouse CD64 (X54-5/7.1, 139303), mouse LY6G (1A8, 127648), mouse SIRPα (P84, 144011), human EGFR (AY13, 352906), human CD3 (HIT3a, 300308), human CD4 (SK3, 344646), human CD8 (SK1, 344724), human IL-13 (JES10-5A2, 501914), human IL-4 (MP4-25D2, 500845), human IL-5 (TRFK5, 504304), mouse/human GATA3 (W19195B, 386906), human CXCR3 (G025H7, 353737), human CCR4 (L291H4, 359443), human CD62L (DREG-56, 304830), human CD95 (DX2, 305622), human CD45RA (HI100, 304120), human CD27 (O323, 302832), human CCR7 (G043H7, 353214), human CD45RO (UCHL1, 304228), human CD66b (G10F5, 305121), human LAG-3 (11C3C65, 369322), human IFNγ (4S.B3, 502530), human Ki-67 (Ki-67, 350526), human CD39 (A1, 328240), human TIM-3 (F38-2E2, 345026), human CD64 (X54-5/7.1, 139301), Human TruStain FcX™ (422302), anti-mouse CD3ε (145-2C11) and Zombie Violet™ Fixable Viability Kit (423114). The following antibodies or staining reagents were purchased from BD Biosciences: pSTAT3 (4/P-STAT3, 557815), pSTAT4 (38/p-Stat4, 558137), pSTAT5 (47/Stat5, 612599), mouse pSTAT6 (J71-773.58.11, 558252), human pSTAT6 (23/Stat6, 612701), and BD Pharmingen™ APC BrdU Kit (552598). The following antibodies were purchased from Cell Signal: pSTAT1 (58D6, 8009S) and Myc-Tag (9B11, 3739/2233S). The following antibodies were purchased from BioXcell: anti-mouse CD16/32 (93), Anti-human IL-4 (MP4-25D2), Anti-human IL-5 (TRFK5), Anti-human IL-13 (Tralokinumab), anti-mouse CD28 (37.51), anti-human CD3ε (OKT-3) and anti-human CD28 (9.3).For flow cytometry staining, surface marker antibodies were used at a 1:200 dilution, intracellular antibodies at 1:100, and pSTAT antibodies at 1:50.

**Validation**

Here are the links of the websites that show validation of the antibodies used in this study:
mouse CD16/32 (93, 101302): https://www.biolegend.com/en-us/search-results/apc-anti-mouse-cd16-32-antibody-6282
mouse CD90.1/Thy.1 (OX-7, 202533): https://www.biolegend.com/en-us/products/apc-anti-rat-cd90-mouse-cd901-thy11-antibody-5621
mouse CD8b (YTS156.7.7, 126606): https://punchout.biolegend.com/en-us/products/fitc-anti-mouse-cd8b-antibody-4475
mouse CD8b (YTS156.7.7, 126614): https://www.biolegend.com/en-us/products/apc-anti-mouse-cd8b-antibody-9055
mouse CD45.2 (104, 109808): https://www.biolegend.com/en-us/products/pe-anti-mouse-cd45-2-antibody-7?GroupID=BLG7007
mouse Ki-67 (16A8, 652413): https://www.biolegend.com/en-us/products/brilliant-violet-605-anti-mouse-ki-67-antibody-8983
mouse SCA-1 (D7,108142): https://www.biolegend.com/en-us/products/alexa-fluor-700-anti-mouse-ly-6a-e-sca-1-antibody-12078
mouse CD44 (IM7, 103030): https://www.biolegend.com/en-us/products/pe-cyanine7-anti-mouse-human-cd44-antibody-3932
mouse CD62L (MEL-14, 104428): https://www.biolegend.com/en-us/search-results/apc-anti-mouse-cd62l-antibody-381
mouse PD-1 (29F.1A12, 135220): https://www.biolegend.com/en-us/products/brilliant-violet-605-anti-mouse-cd279-pd-1-antibody-7648?GroupID=BLG7927
mouse IFNγ (XMG1.2, 505850): https://www.biolegend.com/en-us/products/apc-cyanine7-anti-mouse-ifn-gamma-antibody-13155
mouse TNF-α (MP6-XT22, 506329): https://www.biolegend.com/en-us/products/brilliant-violet-605-anti-mouse-tnf-alpha-antibody-7682
mouse/human Granzyme B (QA16A02, 372214): https://www.biolegend.com/en-us/products/pe-cyanine7-anti-humanmouse-granzyme-b-recombinant-antibody-15582
mouse/human KLRG1 (2F1/KLRG1, 138419): https://www.biolegend.com/en-us/products/brilliant-violet-605-anti-mouse-human-klrg1-mafa-antibody-9644
mouse IL-7Rα (A7R34, 135022): https://www.biolegend.com/en-us/products/percp-cyanine5-5-anti-mouse-cd127-il-7ralpha-antibody-6196
mouse CD122 (TM-β1, 123210): https://www.biolegend.com/en-us/products/pe-anti-mouse-cd122-il-2rbeta-antibody-4160
mouse SLAMF6 (330-AJ, 134606): https://www.biolegend.com/en-us/products/pe-anti-mouse-ly108-antibody-6016
mouse CD11b (M1/70, 101228): https://www.biolegend.com/en-us/products/percp-cyanine5-5-anti-mouse-human-cd11b-antibody-4257
mouse CD14 (Sa14-2, 123335): https://www.biolegend.com/en-us/products/brilliant-violet-605-anti-mouse-cd14-antibody-20001
mouse CD64 (X54-5/7.1, 139303): https://www.biolegend.com/en-us/products/pe-anti-mouse-cd64-fcgammari-antibody-6691
mouse LY6G (1A8, 127648): https://www.biolegend.com/en-us/products/pe-dazzle-594-anti-mouse-ly-6g-antibody-12246
mouse SIRPα (P84, 144011): https://www.biolegend.com/en-us/products/pe-anti-mouse-cd172a-sirpalpha-antibody-9801
human EGFR (AY13, 352906): https://www.biolegend.com/en-us/products/apc-anti-human-egfr-antibody-7714
human CD3 (HIT3a, 300308): https://www.biolegend.com/en-us/products/pe-anti-human-cd3-antibody-753
human CD4 (SK3, 344646): https://www.biolegend.com/en-us/products/brilliant-violet-605-anti-human-cd4-antibody-15992?GroupID=GROUP28
human CD8 (SK1, 344724): https://www.biolegend.com/en-us/products/alexa-fluor-700-anti-human-cd8-antibody-9062
human IL-13 (JES10-5A2, 501914): https://www.biolegend.com/en-us/products/pe-cyanine7-anti-human-il-13-antibody-13129
human IL-4 (MP4-25D2, 500845): https://www.biolegend.com/en-us/products/brilliant-violet-785-anti-human-il-4-antibody-24058

human IL-5 (TRFK5, 504304): https://www.biolegend.com/en-us/products/pe-anti-mouse-human-il-5-antibody-991
mouse/human GATA3 (W19195B, 386906): https://www.biolegend.com/en-us/products/apc-fire-750-anti-gata3-antibody-24357
human CXCR3 (G025H7, 353737): https://www.biolegend.com/en-us/products/brilliant-violet-785-anti-human-cd183-cxcr3-antibody-12124
human CCR4 (L291H4, 359443):https://www.biolegend.com/en-us/products/alexa-fluor-700-anti-human-cd194-ccr4-antibody-24167
human CD62L (DREG-56, 304830): https://www.biolegend.com/en-us/products/brilliant-violet-785-anti-human-cd62l-antibody-7974
human CD95 (DX2, 305622): https://www.biolegend.com/en-us/products/pe-cyanine7-anti-human-cd95-fas-antibody-6495
human CD45RA (HI100, 304120): https://www.biolegend.com/en-us/products/alexa-fluor-700-anti-human-cd45ra-antibody-3421
human CD27 (O323, 302832): https://www.biolegend.com/en-us/products/brilliant-violet-785-anti-human-cd27-antibody-7970
human CCR7 (G043H7, 353214): https://www.biolegend.com/en-us/products/apc-anti-human-cd197-ccr7-antibody-7536
human CD45RO (UCHL1, 304228): https://www.biolegend.com/en-us/products/apc-cyanine7-anti-human-cd45ro-antibody-7372
human CD66b (G10F5, 305121): https://www.biolegend.com/en-us/products/pe-dazzle-594-anti-human-cd66b-antibody-13979
human LAG-3 (11C3C65, 369322): https://www.biolegend.com/en-us/products/brilliant-violet-785-anti-human-cd223-lag-3-antibody-14877
human IFNγ (4S.B3, 502530): https://www.biolegend.com/en-us/products/apc-cyanine7-anti-human-ifn-gamma-antibody-6965
human Ki-67 (Ki-67, 350526): https://www.biolegend.com/en-us/products/pe-cyanine7-anti-human-ki-67-antibody-9084
human CD39 (A1, 328240): https://www.biolegend.com/en-us/products/brilliant-violet-785-anti-human-cd39-antibody-18339
human TIM-3 (F38-2E2, 345026): https://www.biolegend.com/en-us/products/apc-cyanine7-anti-human-cd366-tim-3-antibody-11928
mouse CD64 (X54-5/7.1, 139301): https://www.biolegend.com/en-gb/products/purified-anti-mouse-cd64-fcgammari-antibody-6690?GroupID=BLG8810
Human TruStain FcX™ (422302): https://www.biolegend.com/en-us/products/human-trustain-fcx-fc-receptor-blocking-solution-6462
anti-mouse CD3ε (145-2C11): https://www.biolegend.com/en-us/products/purified-anti-mouse-cd3epsilon-antibody-28
Zombie Violet™ Fixable Viability Kit (423114): https://www.biolegend.com/en-us/products/zombie-violet-fixable-viability-kit-9341
pSTAT3 (4/P-STAT3, 557815): https://www.bdbiosciences.com/en-us/products/reagents/flow-cytometry-reagents/research-reagents/single-color-antibodies-ruo/alexa-fluor-647-mouse-anti-stat3-py705.557815
pSTAT4 (38/p-Stat4, 558137): https://www.bdbiosciences.com/en-us/products/reagents/flow-cytometry-reagents/research-reagents/single-color-antibodies-ruo/alexa-fluor-647-mouse-anti-stat4-py693.558137
pSTAT5 (47/Stat5, 612599): https://www.bdbiosciences.com/en-us/products/reagents/flow-cytometry-reagents/research-reagents/single-color-antibodies-ruo/alexa-fluor-647-mouse-anti-stat5-py694.612599
mouse pSTAT6 (J71-773.58.11, 558252): https://www.bdbiosciences.com/en-us/products/reagents/flow-cytometry-reagents/research-reagents/single-color-antibodies-ruo/pe-mouse-anti-mouse-stat6-py641.558252
human pSTAT6 (23/Stat6, 612701): https://www.bdbiosciences.com/en-us/products/reagents/flow-cytometry-reagents/research-reagents/single-color-antibodies-ruo/pe-mouse-anti-stat6-py641.612701
BD Pharmingen™ APC BrdU Kit (552598): https://www.bdbiosciences.com/en-us/products/reagents/flow-cytometry-reagents/research-reagents/cell-function-analysis-stains-dyes/apc-brdu-kit.552598
pSTAT1 (58D6, 8009S): https://www.cellsignal.com/products/antibody-conjugates/phospho-stat1-tyr701-58d6-rabbit-mab-alexa-fluor-647-conjugate/8009
Myc-Tag (9B11, 3739/2233S): https://www.cellsignal.com/products/antibody-conjugates/myc-tag-9b11-mouse-mab-pe-conjugate/3739
Anti-human IL-4 (MP4-25D2): https://bioxcell.com/invivomab-anti-human-il-4-be0240
Anti-human IL-5 (TRFK5): https://bioxcell.com/invivomab-anti-mouse-human-il-5-be0198
Anti-human IL-13 (Tralokinumab): https://bioxcell.com/invivosim-anti-human-il-13-tralokinumab-biosimilar-sim0042
anti-mouse CD28 (37.51): https://bioxcell.com/invivomab-anti-mouse-cd28-be0015-1
anti-human CD3ε (OKT-3): https://bioxcell.com/invivomab-anti-human-cd3-be0001-2
anti-human CD28 (9.3): https://bioxcell.com/invivomab-anti-human-cd28-be0248

# Eukaryotic cell lines

Policy information about cell lines and Sex and Gender in Research

| | |
|---|---|
| Cell line source(s) | The B16F10 mouse melanoma cell line, HEK293T cell, Raji cell, RAW 264.7 cell, and J774A.1 cell were originally acquired from the American Type Culture Collection (ATCC). Platinum-E (Plat-E) and Platinum-GP (Plat-GP) retroviral Packaging Cell Line were purchased from Cell Biolabs. A375 and nRFP-M407 human melanoma cells were provided by Prof. Antoni Ribas (University of California Los Angeles). Mouse B cell lymphoma A20 cells were provided by Prof. Robert S. Negrin (Stanford University) and originally sourced from ATCC. The NALM6 cell line was obtained from ATCC. NALM6 cells were modified to express firefly luciferase (NALM6-Luc). |
| Authentication | None of the cell lines were authenticated in these studies. In all studies, cell lines with low passage number were used. |
| Mycoplasma contamination | All cell lines were confirmed mycoplasma negative. |
| Commonly misidentified lines (See ICLAC register) | No commonly misidentified cell lines were used. |

# Animals and other research organisms

Policy information about studies involving animals; ARRIVE guidelines recommended for reporting animal research, and Sex and Gender in Research

| | |
|---|---|
| Laboratory animals | Five- to six-week-old female Thy1.2+ C57BL/6 (C57BL/6J) mice and five- to seven-week-old NOD.Cg-Prkdcscid Il2rgtm1Wjl/SzJ (NSG) mice were purchased from Jackson Laboratory. Five- to six-week-old TCR-transgenic Thy1.1+ pmel-1 (pmel) mice (B6.Cg-Thy1a/Cy Tg(TcraTcrb)8Rest/J) were originally purchased from the Jackson Laboratory and maintained in the Stanford University-Lorry Lokey |

| | |
|---|---|
| | (SIM1) Facility. |
| Wild animals | Study did not involve wild animals. |
| Reporting on sex | Female mice were used for all the experiments. |
| Field-collected samples | Study id not involve field-collected samples. |
| Ethics oversight | Mice were housed in animal facilities approved by the Association for the Assessment and Accreditation of Laboratory Care. Experimental procedures in mouse studies were approved by the Institutional Animal Care and Use Committee (IACUC) at the Stanford University (animal protocol ID 32279) and performed in accordance with the guidelines from the animal facility of Stanford University. |

Note that full information on the approval of the study protocol must also be provided in the manuscript.

# Plants

| | |
|---|---|
| Seed stocks | N/A |
| Novel plant genotypes | N/A |
| Authentication | N/A |

# Flow Cytometry

## Plots

Confirm that:

☒ The axis labels state the marker and fluorochrome used (e.g. CD4-FITC).

☒ The axis scales are clearly visible. Include numbers along axes only for bottom left plot of group (a 'group' is an analysis of identical markers).

☒ All plots are contour plots with outliers or pseudocolor plots.

☒ A numerical value for number of cells or percentage (with statistics) is provided.

## Methodology

| | |
|---|---|
| Sample preparation | For in vivo samples, collected tumors were weighed, mechanically minced, and digested in RPMI-1640 medium supplemented with collagenase type IV (1 mg/ml, Gibco/Thermo Fisher Scientific), dispase II (100 μg ml−1, Sigma-Aldrich), hyalurondase (100 μg ml−1, Sigma-Aldrich), and DNase I (100 μg ml−1, Sigma-Aldrich) at 37 °C for 60 min. RBC lysis was performed on the digested tumor samples with ACK lysing buffer. Tumor infiltrating leukocytes were then enriched by Percoll (Cytiva) density gradient centrifugation, resuspended in PBS with BSA (0.2%, wt/v), stained with indicated antibodies, and analyzed by flow cytometry. Spleens were ground and filtered through a 70-μm strainer (Fisher Scientific). RBC lysis was performed on the spleen samples with ACK lysing buffer (2 ml per spleen, Gibco/Thermo Fisher Scientific) and then resuspended in PBS with BSA (0.2%, wt/v). TDLNs were ground and filtered through a 70-μm strainer (Fisher Scientific) and then resuspended in PBS with BSA (0.2%, wt/v). |
| Instrument | CytoFlex (Beckman Coulter) |
| Software | FlowJo (v 10.10.0) |
| Cell population abundance | Among the enriched live singlet Tumor-Infiltrating Lymphocytes, the proportion of transferred T cells ranges from 0.4% to 5.0% |
| Gating strategy | Pmel CD8+ T cells were gated based on Thy1.1+CD8+, and orthogonal chimeric receptor-engineered pmel CD8+ cells were gated using Thy1.1+YFP+CD8+. For antibody-dependent cellular phagocytosis assays, phagocytic activity was measured as FarRed+ cells within CD8+ pmel T cells and CD14+ RAW264.7 cells after gating for single, live cells and excluding target cells. CD19 CAR T cells were gated based on CD3+Myc+, and orthogonal chimeric receptor-engineered CD19 CAR cells were gated using CD3+YFP+Myc+. NY-ESO-1 TCR-T cells were gated based on CD3+EGFR+, and orthogonal chimeric receptor-engineered NY-ESO-1 TCR-T cells were gated using CD3+YFP+EGFR+. |

☒ Tick this box to confirm that a figure exemplifying the gating strategy is provided in the Supplementary Information.

