## [Peer Review file · Nature]

Expanding the cytokine receptor alphabet reprograms T cells into diverse states

Corresponding Author: Professor K. Christopher Garcia

Version 2:

Reviewer comments:

Referee #1

(Remarks to the Author)

Summary:

This manuscript by Zhao et al investigates the effect of engineered heterodimeric cytokine receptors pairs on T cell signaling, gene expression, and function. To address this question, the authors utilize a previously published orthogonal cytokine receptor platform, which enabled them to pair the common gamma chain that is naturally expressed in T cells with domains from diverse receptor families involving gamma chain cytokines, IL-10 family receptors and homodimeric such as GCSF. Notably, the manuscript heavily focuses on characterizing the anti-tumor potential of adoptive transferred T cells engineered with orthogonal chimeric cytokine receptors. Using a variety of functional approaches, the authors make the following key findings: 1) orthogonal chimeric receptors can promote both natural (eg, TCM phenotype or Tc2/Th2 differentiation) and "synthetic" T cell states (e.g. a myeloid-like state that endows T cells with phagocytic capabilities) and 2) specific receptor combinations strikingly augment the antitumor properties of adoptive transferred T cells in both murine syngeneic cancer models and human cancer cell line xenograft models. This study is significant because 1) broadens the synthetic immunology toolkit by identifying novel chimeric receptors, 2) reveals an unexpected degree of phenotypic plasticity and re-programmability in mature T cells, and 3) offers innovative solutions to address key challenges in cell-based immunotherapies.

This study couples innovation in synthetic immunology (cytokine receptor engineering) with striking phenotypes that are biologically interesting (eg, reprogramming T cells to exhibit myeloid features) and clinically relevant, the latter exemplified by the manuscript's focus on the translational potential of cytokine receptor engineering in antitumor immunity. The authors deserve credit for both 1) creativity and 2) detailed and cross-species characterization of anti-tumor immune phenotypes associated with the receptors described. It is evident that this study would have broad scientific interest and relevance to multiple fields including immunology, synthetic biology and oncology.

However, the manuscript should be improved with deeper mechanistic characterization of the key chimeras of interest and with key experiments that support key claims (eg, γ c-dependence of signaling and phenotypic readouts).

Major claims that need bolstering:

Claim 1: gamma-c-dependent heterodimerization drives observed signaling and functional consequences of orthogonal chimeric receptors.

The authors claim that the downstream signaling events (e.g., JAK/STAT activation patterns) and functional consequences (e.g., cell state transitions and anti-tumor properties) arise specifically from the natural and non-natural heterodimeric

“pairing” with the common gamma chain. However, this study provides no direct proof of ligand-induced heterodimerization of the chimeric receptors with γ_c nor does it test whether observed signaling and functional outcomes are strictly γ_c -dependent. pSTAT readouts, while important, are insufficient to conclude heterodimerization and could reflect alternative signaling mechanisms such as homodimerization. This raises the concern that the observed downstream effects could result from parallel signaling events, such as: 1) Homodimerization events (particularly for receptor families that naturally homodimerize, eg, GCSFR) 2) spurious associations with other endogenous signaling partners. Potential ways to address could be KO of gamma chain or using a ortho-IL2 that cannot engage common gamma chain.

Claim 2: orthogonal GCSFR results in myeloid transdifferentiation of T cells

We commend the authors for identifying and making efforts to characterize what is undoubtedly a fascinating phenotype. This claim is well supported experimentally by 1) expression of myeloid like gene expression program 2) acquisition of myeloid surface markers and 3) functional assays demonstrating bacterial phagocytosis and ADCP. However, I am left with several questions:

1. Relevant to the critique above, can the authors demonstrate that the downstream signaling events and reprogramming of T cells expressing oGCSFR arise from γ_c -heterodimerization (and not a parallel process such as homodimerization?)
2. Are the phenotypes observed in mouse T cells recapitulated in hoGCSFR-expressing CD19 CART cells. Demonstrating myeloid reprogramming in human T cells would strengthen this claim further.
3. To what extent is the enhanced antitumor activity of oGCSFR-transduced T cells related to myeloid reprogramming?

Minor:

1. Typographical/grammatical errors

- Page 13 line 13: “spectrum functions” -> “spectrum of functions”
- Page 4 line 6: “output” -> “outputs”
- Page 4 line 13: “Incudes” -> “induces”
- Page 7 line 12: “landscape” – “landscapes”
- Page 9 line 16: “dimensional reduction” -> “dimensionality reduction”
- Page 9 line 28: “compare” -> “compared”
- Page 10 line 6: “observed approximately a” -> “observed an approximately 2.8 fold”
-

2. The manuscript writing can overall be improved for 1) brevity 2) clarity 3) precision + accuracy

Referee #3

(Remarks to the Author)

In this study, Zhao et al aim to understand how T cell biology is altered with non-natural common gamma chain containing receptors, especially in the context of adoptive cell therapy in cancer. Building off previous work using their orthogonal IL-2 platform, they cleverly switched the intracellular IL-2R β component for receptor alternatives, including those that naturally associate with common gamma chain and non-natural associations. Specifically, they engineered T cells that express receptors in which the IL-2R β chain encompass the wildtype intracellular domain (ICD) as well as ICDs from IL-4R (orthogonal4R) along with multiple other cytokine receptors. Using stimulation with mouse serum albumin fused with IL-2 (MSA-oIL2), receptors with varying ICDs activate STATs that largely align with the expected cognate interleukin. Strikingly, o20R and o225 activate STATs 1-5. oGCSFR activates STAT3 and STAT5. Transcriptomes of T cells activated with oIFNRs are most distinct from other cytokine receptors. Mice inoculated with subcutaneous B16F10 melanomas were treated with intravenous adoptive cell transfer of pme1 T cells engineered using various cytokine receptor ICDs and intraperitoneal MSA-oIL2. Notably, oGCSFR, oEPOR, o20R, o22R, oIFNGR1, oIFNGR1, oIFNLR1, and o4R were superior with respect to tumor size reduction. The authors next employed scRNAseq CD8+ tumor infiltrating lymphocytes following iv adoptive transfer of pme1 T cells and ip MSA-oIL2. O22R cells generated stemlike cells that expressed Lef1, Foxo1. oGCSFR CD8 T cells exhibited myeloid features and o4R expressing T cells displayed Il4, Il13 and Il5. Human NY-ESO-1 TCR T cells were transduced with human orthogonal chimeric IL-4R, which also generated IL-4 /5/13/GATA3+ cells that also expressed IFNg. NSG mice were inoculated with A375 tumors followed by iv adoptive transfer of NY-ESO-1 TCR-T cells expressing ho4R, ho22R and hoGCSFR and ip administration of MSA-hoIL2 and were superior to control T cells in terms of tumor reduction. NSG mice were also inoculated with NALM and Raji cells were treated with CD19 CAR T cells expressing ho22R and hoGCSFR resulting in reduced tumor burden. Chromatin accessibility was assessed using ATAC-seq which revealed less

accessibility of Tox and increased accessibility of Runx1 and Rora in T cells in mice treated with oh22R and oGCSFR expressing cells. This study is thought provoking, creative, and insightful, and addressing the following questions/topics would strengthen its clarity and impact:

1. Transcriptomic data induced by oIFNGR1 appears to be absent in Fig 1h. It is unclear why this was omitted, especially given the relative efficacy shown in Figure 2d. This is of particular interest insofar as IFNg and IFNa/b have been reported to promote exhaustion.
2. The data shown regarding hoGCSFR are provocative and important. However, the data shown in Fig.6f and g and extended data Fig 6h are confusing with respect to survival in response. The data presented in Ext Fig 6f depicts a CR of 3/5. This would seem to be a very important part of the story. In Figure 6 and perhaps other figures, the number of mice should be increased - 5 mice per condition do not provide sufficient power for tumor experiments. Also, the investigators only use female mice. This is common practice; however, the goal of this work is clinical therapeutic efficacy. In this case, it would seem that discerning sex differences are pertinent given the potential therapeutic relevance. Also, Figure 6b x-axis values are absent.
3. The author's measurement of pStat activity downstream of orthogonal receptors is provocative. It looks like the receptors that provide the greatest efficacy (oGMCSF, o22R, o4R) showed reduced pStat5. This is interesting, especially since the authors mention that Stat5 activation has been shown previously to be beneficial in adoptive T cell transfer therapy. More consideration of this point is warranted.
4. In various experiments, mixed CD4 and CD8 T cells are used whereas in other experiments just CD8s are used. This is not always clearly stated. If not pools, the enrichment/sorting protocol in the methods and clarification of the subset needs to be more explicit in the figure legend/text. Mechanistically, the potential impact of the orthogonal receptor signals on CD4s and CD8s may confuse readers; clarification is needed. 5. The rationale behind Figure 4F is unclear. Cell-to-cell interaction analysis would be more informative analyzing cells isolated from the entire tumor, not just the T cells; the therapeutic importance of T cells also relates to their ability to interact with non-T cells.
6. The data provided in Figure 5C is hard to interpret. How did the authors decide where to gate? Can they include a negative control for cytokine production? There The plots would be clearer if a comparator was provided (rather than CD4 and CD8) if both axes were a cytokine, e.g. IFNg vs IL-4.
7. Also unclear is why pSTAT3 data are not provided with IFN stimulation. Including the primary data would be useful with appropriate controls (e.g. Ext Data Fig 1).

Referee #4

(Remarks to the Author)

In this manuscript, the author designed a series of chimeric cytokine receptor based on their previously described orthogonal IL2-IL2RB system to create multiple unnatural /JAK1/2/3 pairing, resulting in unique downstream signaling events that are not native to T cells. This system enables them to activate specific cytokine signaling using an orthogonal IL2 ligand where they showed that integrating orthogonal IL4, IL22 and GCSF chimeric receptor could greatly improve anti-tumor activity of pmel T cells against B16 melanoma. Using scRNA-Seq, they further studied different unique state/fate of T cells when o4R, o22R or oGCSFR are expressed. Interestingly, they found that oGCSFR could endow T cells with myeloid signature and potentially phagocytic activity. Integrating IL4 signaling foster a type 2 state of T cells that is associated with stronger and durable anti-tumor activity in vivo. Lastly, T cells expressing o22R appeared to possess more stem-like features in vivo, leading to best tumor control and long-term persistence of T cells in vivo.

I believe this is a novel and well-designed study with great translational potential (especially with o22R). However, I think there are quite a lot of weaknesses in this manuscript that need to be addressed. Specifically, I have concerns with the reported phagocytic activity of T cells (which I believe is one of the biggest highlights, if true) and with the mechanism of stemness endowed by o22R, which is the key translational point.

Major comments:

1. Conceptually, I wonder how this strategy compared to other semi-orthogonal cytokine signaling strategy (such as the one reported by Naoto Hirano group, Nat. Med 2018 "A novel chimeric antigen receptor containing a JAK-STAT signaling domain mediates superior antitumor effects") where ICD of cytokine receptor is directly coupled to the CAR molecule. For instance, if I add ICD of IL22R or IL4R to the endodomain of a CAR, would that have similar (better or worse) activity in terms of anti-tumor effects and T cell expansion as what's reported here?
2. In all animal experiments, a condition where chimeric receptor is expressed but MSA-oIL2 is not added is missing. While the chimeric receptor function requires oIL2 in vitro, in vivo environments where native IL2 and other cytokines are also present would complicate the interpretation of the results. It's important to demonstrate that the observed anti-tumor activity of chimeric receptor is dependent on the orthogonal ligand instead of other possible ligands in vivo.
3. In ext.figure 4h,i, the authors showed an increased number of granzyme B and IFNG/TNF positive T cells within tumors of oIL4R, oIL22R etc.. I wonder whether the increased density of these effector population is due to increased expression per cell and/or to the increase in total T cell numbers (as shown in ext.fig 3b). Showing FACS plot of granzyme B/IFNG/TNF of YFP+ TILs would be more informative.
4. While C/EBPb is a putative target gene of MafB, are C/EBPb target genes in fig. 4b upregulated in c6 or in oGCSFR T

cells? If not, then I doubt C/EBP β is the TF driving the phenotype in c6 or in oGCSFR T cells, then singling out C/EBP β is irrelevant.

5. In fig. 4g-k authors showed that OE of Nf1c, Maf, MafB is not sufficient to phenocopy oGCSFR and postulate a combination of 2-3 TFs necessary to drive the expression of myeloid genes in T cells. But it is equally likely that other TFs could be involved here. Did you try to OE multiple TFs and measure the phenotype?

6. The authors performed several phagocytic assays to demonstrate the oGCSFR T cells possess phagocytic activity. Relying on mere MFI would over-estimate the effect since the fluorescent cells could be adhering to the cell surface instead of being phagocytosed. The only image is fig.4m but the image of control T cells is missing so it's hard to judge whether control T cells are showing similar phenotype. So, it is imperative to show the confocal image of phagocytic cells of all experiments. Also, blocking experiments should be conducted in parallel, at least with the Ab-opsonized A20 cells where the FcR γ being the clear phagocytic receptor. Since oGCSFR T cells are showing superior anti-tumor activity, do you think the phagocytic activity is required for this observation?

7. The experiment/timepoint where CAR-T cells are collected for ATAC-Seq to study stemness may not be optimal since there is already big difference in tumor burden so it's likely that CAR-T cells in different groups are showing different activation state due to the amount of tumor cells. I believe the presented data provide very little insight to the mechanism of stemness in o22R or oGCSFR T cells. The observed opening of AP-1 region may explain improved functionality but it should also push T cells towards effector state instead of stem-like state. BACH2 could be involved but I would need to see more evidence (e.g. KD BACH2 or demonstrate BACH2 downstream genes related to stemness being upregulated) to be convinced.

Minor comments:

1. In extended figure 1a, o10R-YFP plot appears to have abnormal negative cell populations, please ensure biexponential expression setting is consistent across all plots.

2. Also, I believe it would be more informative to show the chimeric receptor expression instead of YFP expression since IRES-YFP expression only indicates presence of transgene mRNA but missing the information of protein expression. This is important since chimerism may affect protein stability.

3. The notion that CD95 is a stemness marker is inaccurate, CD95 itself does not indicate stemness but the co-expression of CD95 and CD45RA/CCR7 (in human) distinguish TSCM from naïve T cells. Hence, I don't think CD95 itself is a stemness marker. However, I agree with the overall message that o22R pmel T cells have higher percentage of stem-like T cells. This comment also applies to ext.fig.6e/7f, I think it's best to show a representative FACS plot of CD45RA/CD62L/CCR7-CD95 (as in other studies where they define TSCM population in human CAR-T cells).

4. Ext.figure 5a, the y-axis label "proportion of NA" is quite misleading, maybe "proportion of unmapped" is better?

5. Did you evaluate the effectiveness of o4R on CAR-T cells with other tumor models? Do you think the anti-tumor activity of o4R depends on type 2 cytokines or a specific state of T cells that is associated with this type of T cells? For instance, if you can reduce (but not eliminate) GATA3 expression to match what's in control, would that abolish the anti-tumor activity of o4R T cells? Or if we KO IL4/5/13, what would happen? Maybe it's not entirely within the scope of this study but I would like the authors to add in some discussion at least.

6. In fig.6, the authors only showed T cell counts and functional phenotype in A375 model, what about the other two models?

Version 3:

Reviewer comments:

Referee #1

(Remarks to the Author)

The authors have addressed all my concerns. I think the data in the RL-only Fig 2 and 3 is actually important enough to include in the manuscript and should go into the supplemental data somewhere.

Referee #3

(Remarks to the Author)

concerns identified are satisfactorily addressed in the revised manuscript where possible, I suggest that figures provided to reviewers be included to readers

Referee #4

(Remarks to the Author)

The authors have adequately address my previous concerns. With the newly added data, this manuscript present a novel and robust approach that broadens the versatility and application prospects of T-cell based therapy. I have no further comments.

We are grateful for the valuable feedback provided by the editor and reviewers. We have addressed the points raised by the reviewers.

New results are summarized below:

Reviewer#1

- 1) We provided data demonstrating that oIL-2 signals through a chimeric IL-2R β / γ_c receptor heterodimer, and not through other assemblies.
- 2) We showed that MSA-oIL2 is required for the antitumor effect of natural (o4R) and non-natural (oGCSFR) orthogonal receptors (**New Extended Data Fig. 2d**).
- 3) We provide further evidence that hoGCSFR induces a myeloid-like phenotype in human T cells (**New Extended Data Fig. 7h–j**).

Reviewer#3

- 4) We conducted a STAT3 signaling assay on the orthogonal IFN receptor family; all three receptors showed minimal STAT3 activation levels (**New Fig. 1e** and **New Extended Data Fig. 1d**).
- 5) RNA-seq data of oIFNGR1 T cells were added to PCA and predicted STAT activity, showing weak transcriptional programs (**New Fig. 1h,i**).
- 6) We conducted the antitumor efficacy studies using male mice in the Raji and A375 tumor models (**New Extended Data Fig. 8d,e and 10a,b**).

Reviewer#4

- 2) We showed that MSA-oIL2 is required for the antitumor effect of natural (o4R) and non-natural (oGCSFR) orthogonal receptors (**New Extended Data Fig. 2d**).
- 7) We presented representative FACS plots of IFN γ ⁺TNF α ⁺ and granzyme B⁺ cells, demonstrating that the increased density of cytotoxic T cells in tumors by o22R is due to both a higher proportion of cytokine⁺ cells and increased cell counts (**New Extended Data Fig. 4h**).
- 8) We added images showing oGCSFR CD8⁺ T cells with internalized live GFP⁺ Listeria monocytogenes compared to NT controls (**New Extended Data Fig. 5e**).
- 9) We demonstrated that oGCSFR CD8⁺ T cells enhanced tumor fragment uptake compared to NT controls, which is mediated by the Fc receptor (**New Extended Data Fig. 5f–h**).
- 10) We found that *GATA3*, but not secreted type 2 cytokines, is essential for the direct antitumor activity of ho4R NY-ESO-1 TCR-T cells (**New Extended Data Fig. 6f,g**).
- 11) We showed increased frequency of CD45RA⁺CD62L⁺CCR7⁺CD95⁺ in ho22R and hoGCSFR T cells, which confirms their role in promoting T cell stemness (**New Extended Data Fig. 7f,g and 9f**).
- 12) We showed that ho22R exhibited increased expansion and a less-exhausted phenotype in the Raji lymphoma model (**New Extended Data Fig. 8f–j**).
- 13) We found that *BACH2* knockout in ho22R T cells led to the loss of stemness and promoted differentiation toward an effector phenotype (**New Extended Data Fig. 10i–m**).

(Reviewers' comments in black and authors' response in blue)

Referees' comments:

Referee #1 (Remarks to the Author):

Summary:

This manuscript by Zhao et al investigates the effect of engineered heterodimeric cytokine receptors pairs on T cell signaling, gene expression, and function. To address this question, the authors utilize a previously

published orthogonal cytokine receptor platform, which enabled them to pair the common gamma chain that is naturally expressed in T cells with domains from diverse receptor families involving gamma chain cytokines, IL-10 family receptors and homodimeric such as GCSF. Notably, the manuscript heavily focuses on characterizing the anti-tumor potential of adoptive transferred T cells engineered with orthogonal chimeric cytokine receptors. Using a variety of functional approaches, the authors make the following key findings: 1) orthogonal chimeric receptors can promote both natural (eg, TCM phenotype or Tc2/Th2 differentiation) and “synthetic” T cell states (e.g. a myeloid-like state that endows T cells with phagocytic capabilities) and 2) specific receptor combinations strikingly augment the antitumor properties of adoptive transferred T cells in both murine syngeneic cancer models and human cancer cell line xenograft models. This study is significant because 1) broadens the synthetic immunology toolkit by identifying novel chimeric receptors, 2) reveals an unexpected degree of phenotypic plasticity and re-programmability in mature T cells, and 3) offers innovative solutions to address key challenges in cell-based immunotherapies.

This study couples innovation in synthetic immunology (cytokine receptor engineering) with striking phenotypes that are biologically interesting (eg, reprogramming T cells to exhibit myeloid features) and clinically relevant, the latter exemplified by the manuscript’s focus on the translational potential of cytokine receptor engineering in antitumor immunity. The authors deserve credit for both 1) creativity and 2) detailed and cross-species characterization of anti-tumor immune phenotypes associated with the receptors described. It is evident that this study would have broad scientific interest and relevance to multiple fields including immunology, synthetic biology and oncology.

However, the manuscript should be improved with deeper mechanistic characterization of the key chimeras of interest and with key experiments that support key claims (eg, γ_c -dependence of signaling and phenotypic readouts).

We appreciate the reviewer’s consideration. Below, we have addressed the points raised.

Major claims that need bolstering:

Claim 1: gamma-c-dependent heterodimerization drives observed signaling and functional consequences of orthogonal chimeric receptors.

The authors claim that the downstream signaling events (e.g., JAK/STAT activation patterns) and functional consequences (e.g., cell state transitions and anti-tumor properties) arise specifically from the natural and non-natural heterodimeric “pairing” with the common gamma chain. However, this study provides no direct proof of ligand-induced heterodimerization of the chimeric receptors with γ_c nor does it test whether observed signaling and functional outcomes are strictly γ_c -dependent. pSTAT readouts, while important, are insufficient to conclude heterodimerization and could reflect alternative signaling mechanisms such as homodimerization. This raises the concern that the observed downstream effects could result from parallel signaling events, such as: 1) Homodimerization events (particularly for receptor families that naturally homodimerize, eg, GCSFR) 2) spurious associations with other endogenous signaling partners. Potential ways to address could be KO of gamma chain or using a ortho-IL2 that cannot engage common gamma chain.

In our previous work , we developed mouse and human orthogonal IL-2 system by mutating “site 1” of IL-2, which is responsible for engaging IL-2R β , together with a compensatory mutation in the IL-2R β binding site¹⁻⁵ (**RL-only Fig. 1, a,b**). In this study, we kept the extracellular domain (ECD) of orthogonal IL-2 and IL-2R β and swapped the intracellular domain (ICD) with selected signaling receptors (**RL-only Fig. 1, c**). Thus, oIL-2 only binds to the ECD of ortho-IL-2R β , neither binds to their wild-type partners, conferring site 1 orthogonality for oIL-2/ortho-IL-2R β . Aside from the mutation at “site 1”, oIL-2 is identical to IL-2. Thus, oIL-2 retains the natural “site 2” binding site of IL-2 for the ECD of γ_c , which together with site 1

engagement of the ortho-IL-2R β ECD, compels the receptors to form a heterodimer. The structural constraints imposed by the cytokine and the extracellular domains of the IL-2 receptors⁶, and the requirement for transactivation by the JAK1/JAK3 heterodimer⁷ collectively ensure cytokine-dependent heterodimerization. To address the concern raised by the reviewer, we present the following experimental data to support the conclusion that the observed signaling and functional outcomes are strictly dependent on heterodimerization with γ_c :

1. **Signaling dependence on orthogonal ligand:** STAT signaling curves from our experiments show that signaling is strictly dependent on the orthogonal ligand concentration (Extended Data Fig. 1c–f). In the absence of the ligand, receptor dimerization does not occur. This shows that the observed signaling events are ligand-dependent and not from auto-homodimerization in the absence of the ligand.
2. **γ_c knockout in YT-1 cells:** To validate the role of γ_c in orthogonal signaling, we engineered the human NK cell line YT-1 to express CD25 (which enhances the interaction of the oIL-2 /ortho-IL-2R β / γ_c complex) and various orthogonal receptors (including ho2R, ho10R, ho2R, and hoGCSFR) (RL-only Fig. II, a), and consistently observed a STAT signaling response in an MSA-hoIL2-dose-dependent manner (RL-only Fig. II, b–d). However, in the absence of γ_c (RL-only Fig. II, a), no signaling was detected (RL-only Fig. II, e–g), demonstrating that γ_c is essential for orthogonal receptor activation.
3. **Orthogonal IL-2 with γ_c interface mutations fails to activate STAT:** An orthogonal IL-2 variant (oIL-2-TR) carrying mutations at key residues in the human IL-2– γ_c interface (Q126T and S130R)⁸ failed to induce pSTAT5 signaling in o2R-expressing CD25⁺ YT-1 cells (RL-only Fig. III). In contrast, wild-type orthogonal IL-2 elicited dose-dependent STAT5 activation, suggesting that heterodimerization with γ_c is required for pSTAT signaling.
4. **Anti-tumor efficacy in the absence of orthogonal ligand:** We also demonstrated that orthogonal receptor transduced pmel CD8⁺ T cells (oGCSFR, o4R) lose their anti-tumor activity in the absence of the orthogonal ligand in the B16F10 tumor model, ruling out the possibility of signaling through endogenous ligands (New Extended Data Fig. 2d).

Together, these experiments confirm that the downstream signaling events and functional outcomes observed in our study are dependent on γ_c and ligand-induced heterodimerization.

RL-only Fig. I. Schematic of wild type IL-2/IL-2R β (a), ortho-IL-2/ortho-IL-2R β (b) or ortho-IL-2/ortho-IL-2R β chimeric receptor (c). Schematics created using BioRender.com.

RL-only Fig. II. STAT signaling response in γ_c knockout YT-1 cells. **a**, Representative FACS histogram showing CD25 and CD132 (γ_c) expression levels. **b–g**, Orthogonal chimeric receptors were introduced into YT-1 cells using YFP-encoding retroviral vectors. Cells were stimulated with MSA-hoIL2 for 20 minutes and analyzed for pSTAT via flow cytometry. Data are presented as the MFI of pSTAT from YFP⁺ T cells, plotted against the log₁₀ concentration of MSA-hoIL2. **b–d**, pSTAT-1 (**b**), pSTAT-3 (**c**), and pSTAT-5 (**d**) signaling dose-response curves for ho2R, ho10R, ho22R, and hoGCSFR-transduced YT-1 (CD25⁺) cells. **e–g**, pSTAT-1 (**e**), pSTAT-3 (**f**), and pSTAT-5 (**g**) signaling dose-response curves for ho2R, ho10R, ho22R, and hoGCSFR-transduced γ_c knockout (KO) YT-1 cells. All data represent mean \pm s.e.m.

RL-only Fig. III. pSTAT signaling response in CD25⁺ YT-1 cells. human o2R were introduced into CD25⁺ YT-1 cells using YFP-encoding retroviral vectors. Cells were stimulated with orthogonal IL-2 (oIL-2) or oIL-2-TR (carrying

the Q126T and S130R mutations) for 20 minutes and analyzed for pSTAT via flow cytometry. Data are presented as the MFI of pSTAT from YFP⁺ T cells. All data represent mean \pm s.e.m.

New Extended Data Fig. 2d. The experimental setting is described in **Fig. 2a** ($n = 5$). Shown are average tumor growth curves. All data represent mean \pm s.e.m. and are analyzed by two-way ANOVA with Tukey's post-test (**d**).

Claim 2: orthogonal GCSFR results in myeloid transdifferentiation of T cells

We commend the authors for identifying and making efforts to characterize what is undoubtedly a fascinating phenotype. This claim is well supported experimentally by 1) expression of myeloid like gene expression program 2) acquisition of myeloid surface markers and 3) functional assays demonstrating bacterial phagocytosis and ADCP. However, I am left with several questions:

1. Relevant to the critique above, can the authors demonstrate that the downstream signaling events and reprogramming of T cells expressing oGCSFR arise from γ_c -heterodimerization (and not a parallel process such as homodimerization?)

As detailed in our response to the query above, the signaling events and functional outcomes observed in our study are strictly dependent on ortho-IL-2 ligand-induced receptor heterodimerization of orthogonal IL-2R β and endogenous γ_c . The orthogonal IL-2R β -GCSFR chimera contains the same extracellular domains as all of the other ortho-chimeras (RL-only Fig. I), and therefore does not signal as a homodimer. Specifically, we show that the presence of the orthogonal ligand is necessary for both signaling and anti-tumor efficacy (**Extended Data Fig. 1c-f** and **New Extended Data Fig. 2d**). The interaction between the ligand and γ_c is also required for signaling (**RL-only Fig. II** and **RL-only Fig. III**). These results confirm that the observed effects are a consequence of heterodimeric pairing of the orthogonal IL-2R β -GCSFR chimera and γ_c .

2. Are the phenotypes observed in mouse T cells recapitulated in hoGCSFR-expressing CD19 CART cells. Demonstrating myeloid reprogramming in human T cells would strengthen this claim further.

We assessed the phenotype of hoGCSFR-expressing human T cells treated with MSA-hoIL2. Aligned with mouse receptors, hoGCSFR also induces a myeloid-like T cell phenotype (Mac-1⁺, CD66b⁺) in human T cells (**New Extended Data Fig. 7h-j**).

New Extended Data Fig. 7h–j. NT and oGCSFR CD8⁺ human T cells were stimulated with MSA-hoIL2 (500 nM) for 72 hours, followed by flow cytometry analysis (n = 6). **h**, Representative flow cytometry plots of Mac-1⁺ and CD66b⁺ cells among CD8⁺ T cells. **i,j**, Shown are the frequencies of Mac-1⁺ (**i**) and CD66b⁺ (**j**) cells among CD8⁺ T cells. All data represent mean ± s.e.m. and are analysed by two-tailed Student’s t-test (**i,j**).

3. To what extent is the enhanced antitumor activity of oGCSFR-transduced T cells related to myeloid reprogramming?

Fc receptor-induced antitumor activity by oGCSFR T cells requires tumor-targeted antibodies. To investigate this, we co-cultured mouse CD19-expressing B16F10 cells with NT and oGCSFR pmel CD8⁺ T cells, in the presence or absence of mCD19 antibodies. We found that oGCSFR enhanced antitumor killing compared to NT controls under both conditions (**RL-only Fig. IV**). In the presence of CD19 antibodies, oGCSFR increased tumor killing from ~57.8% to ~70.8%, while NT controls showed no significant changes. In the in vivo scenario, where endogenous antibodies tag B16F10 cells, we expect the anti-tumor effect to be outweighed by direct T cell cytotoxicity. However, oGCSFR adoptive T cell therapy may synergize with tumor-targeting antibody-based therapies.

RL-only Fig. IV. oGCSFR pmel CD8⁺ T cells were preserved in MSA-hoIL2 (500 nM) for 24 hours, while NT pmel CD8⁺ cells were cultured in MSA-IL-2 (100 U/mL). mouse CD19-expressing B16F10 tumor cells (B16F10-mCD19) were co-cultured with NT and oGCSFR pmel CD8⁺ T cells at a 1:1 E:T ratio with MSA-hoIL2 (200 nM), in the presence or absence of mCD19 antibodies (2 µg/mL) for 48 hours. Cells were collected for flow cytometry analysis. Shown are the percentages of B16F10-mCD19 cell killing. All data represent mean ± s.e.m. and were analyzed by one-way ANOVA with Tukey’s post-test.

Minor:

1. Typographical/grammatical errors

- Page 13 line 13: “spectrum functions” -> “spectrum of functions”
- Page 4 line 6: “output” -> “outputs”
- Page 4 line 13: “Incudes” -> “induces”
- Page 7 line 12: “landscape” – “landscapes”
- Page 9 line 16: “dimensional reduction” -> “dimensionality reduction”
- Page 9 line 28: “compare” -> “compared”
- Page 10 line 6: “observed approximately a” -> “observed an approximately 2.8 fold”

We thank the reviewer for pointing out the errors. We have reviewed and corrected the manuscript accordingly.

2. The manuscript writing can overall be improved for 1) brevity 2) clarity 3) precision + accuracy

We thank the reviewer for the suggestions. We have revised the results section for improved readability and accuracy.

Referee #3 (Remarks to the Author):

In this study, Zhao et al aim to understand how T cell biology is altered with non-natural common gamma chain containing receptors, especially in the context of adoptive cell therapy in cancer. Building off previous work using their orthogonal IL-2 platform, they cleverly switched the intracellular IL-2Rb component for receptor alternatives, including those that naturally associate with common gamma chain and non-natural associations. Specifically, they engineered T cells that express receptors in which the IL-2Rb chain encompass the wildtype intracellular domain (ICD) as well as ICDs from IL-4R (orthogonal4R) along with multiple other cytokine receptors. Using stimulation with mouse serum albumin fused with IL-2 (MSA-oIL2), receptors with varying ICDs activate STATs that largely align with the expected cognate interleukin. Strikingly, o20r and o225 activate STATs 1-5. oGCSFR activates STAT3 and STAT5. Transcriptomes of T cells activated with oIFNRs are most distinct from other cytokine receptors. Mice inoculated with subcutaneous B16F10 melanomas were treated with intravenous adoptive cell transfer of pme1 T cells engineered using various cytokine receptor ICDs and intraperitoneal MSA-oIL2. Notably, oGCSFR, oEPOR, o20R, o22R, oIFNGR1, oIFNGR1, oIFNLR1, and o4R were superior with respect to tumor size reduction. The authors next employed scRNAseq CD8+ tumor infiltrating lymphocytes following iv adoptive transfer of pme1 T cells and ip MSA-oIL2. O22R cells generated stemlike cells that expressed Lef1, Foxo1. oGCSFR CD8 T cells exhibited myeloid features and o4R expressing T cells displayed Il4, Il13 and Il5. Human NY-ESO-1 TCR T cells were transduced with human orthogonal chimeric IL-4R, which also generated IL-4 /5/13/GATA3+ cells that also expressed IFNg. NSG mice were inoculated with A375 tumors followed by iv adoptive transfer of NY-ESO-1 TCR-T cells expressing ho4R, ho22R and

hoGCSFR and ip administration of MSA-hoIL2 and were superior to control T cells in terms of tumor reduction. NSG mice were also inoculated with NALM and Raji cells were treated with CD19 CAR T cells expressing ho22R and hoGCSFR resulting in reduced tumor burden. Chromatin accessibility was assessed using ATAC-seq which revealed less accessibility of Tox and increased accessibility of Runx1 and Rora in T cells in mice treated with oh22R and oGCSFR expressing cells. This study is thought provoking, creative, and insightful, and addressing the following questions/topics would strengthen its clarity and impact:

We thank the reviewer for the favorable remarks. We have addressed all the concerns as shown below in detail.

1. Transcriptomic data induced by oIFNGR1 appears to be absent in Fig 1h. It is unclear why this was omitted, especially given the relative efficacy shown in Figure 2d. This is of particular interest insofar as IFN γ and IFN α/β have been reported to promote exhaustion.

In RNA-seq, oIFNGR1 shows surprisingly weak transcriptional features in T cells. This is consistent with our unpublished observations that primary T cells show very weak responsiveness to oIFNGR1 at 6 hours post-stimulation (**RL-only Fig. V**). Canonical STAT1 target genes such as *Cd274* and *Ifitm1* were strongly induced only at 48 hours. Since our RNA-seq was performed at 6 hours, it may not capture the peak transcriptional response of all receptors. We have included this data in the PCA analysis and predicted STAT activity (**New Fig. 1h,i**), which could provide more reference for early IFN γ signaling in T cells.

RL-only Fig. V. Normalized mRNA expression levels of indicated genes in primary mouse T cells transduced with NoICD or oIFNGR1, following 6- or 48-hour stimulation (unstimulated [NS] or stimulated with MSA-oIL-2). All data represent mean \pm s.e.m.

2. The data shown regarding hoGCSFR are provocative and important. However, the data shown in Fig.6f and g and extended data Fig 6h are confusing with respect to survival in response. The data presented in Ext Fig 6f depicts a CR of 3/5. This would seem to be a very important part of the story. In Figure 6 and perhaps other figures, the number of mice should be increased - 5 mice per condition do not provide sufficient power for tumor experiments. Also, the investigators only use female mice. This is common practice; however, the goal of this work is clinical therapeutic efficacy. In this case, it would seem that

discerning sex differences are pertinent given the potential therapeutic relevance. Also, Figure 6b x-axis values are absent.

We apologize for the confusion. We omitted the individual growth curves for Fig. 6f,g. The previous Extended Data Fig. 6h (now Extended Data Fig. 8c in the revised manuscript) shows the individual tumor growth curves for Fig. 6d. To improve readability, we have added the individual growth curves for Fig. 6f,g in the New Extended Data Fig. 9g in the revised manuscript. All correspondence between average and individual tumor growth curves is described in the figure caption.

We agree that the antitumor efficacy is a crucial part of the study. We have tested the antitumor efficacy of the receptors across one syngeneic tumor model (with 8-9 mice per group) and three xenograft tumor models (with 5 mice per group), variation within group is very small. In Fig. 6 (xenograft tumor models), we observed minimal variation within each group. In addition, all experiments were first tested in a pilot study to explore conditions. However, we agree that increasing the sample size will provide more statistical power. As requested, we have conducted two independent experiments using male mice in the Raji and A375 models. In the Raji model, both ho22R and hoGCSFR CD19 CAR-T cells completely eradicated the tumor, with 100% of mice remaining tumor-free (New Extended Data Fig. 8d,e). In the A375 model, both hoGCSFR and ho22R NY-ESO-1 TCR-T cells demonstrated sustained tumor control in all treated mice. ho22R NY-ESO-1 TCR-T cells cleared tumors in 4 out of 5 mice, while hoGCSFR TCR-T cells achieved tumor clearance in 2 mice (New Extended Data Fig. 10a,b). The revised manuscript now includes corresponding statements describing these new experiments (page 14, lines 27-29 and page 15, lines 20-23).

X-axis values in Fig. 6b have been added to the revised manuscript.

New Extended Data Fig. 9g. NSG female mice were inoculated (s.c.) with HLA*0201⁺ NY-ESO-1⁺ A375 melanoma cells (5×10^5) and received an i.v. adoptive transfer of 3×10^6 NT, ho2R, ho22R or hoGCSFR CD3⁺ NY-ESO-1 TCR-T cells on Day 6, followed by i.p. administration of MSA-hoIL2 (1×10^5 unit) or PBS every other day until day 20 (n = 5 animals). Shown are the individual tumor growth curves. Indicated are the numbers of tumor-free mice per total number of mice in each group. All data represent mean \pm s.e.m.

New Extended Data Fig. 8d,e. NSG male mice were inoculated (s.c.) with Raji cells (1×10^6) and received an i.v. adoptive transfer of 1×10^6 NT, ho22R, or hoGCSFR CD3⁺ CD19 CAR T cells on Day 7, followed by i.p. administration of MSA-hoIL2 (2.5×10^4 unit) or PBS every other day until day 29 ($n = 5$ animals). **d**, Average tumor growth curves. **e**, Survival curves. Indicated are the numbers of tumor-free mice per total number of mice in each group. All data represent mean \pm s.e.m. and are analyzed by two-way (**d**) ANOVA with Tukey's post-test, or log-rank (Mantel-Cox) test (**e**).

New Extended Data Fig. 10a,b. NSG male mice were inoculated (s.c.) with HLA*0201⁺ NY-ESO-1⁺ A375 melanoma cells (5×10^5) and received an i.v. adoptive transfer of 3×10^6 NT, ho2R, ho22R or hoGCSFR CD3⁺ NY-ESO-1 TCR-T cells on Day 6, followed by i.p. administration of MSA-hoIL2 (1×10^5 unit) or PBS every other day until day 20 ($n = 5$ animals). **a**, Average tumor growth curves. **b**, Survival curves. Indicated are the numbers of tumor-free mice per total number of mice in each group. All data represent mean \pm s.e.m. and are analysed by two-way (**a**) ANOVA with Tukey's post-test, or log-rank (Mantel-Cox) test (**b**).

3. The author's measurement of pStat activity downstream of orthogonal receptors is provocative. It looks like the receptors that provide the greatest efficacy (oGMCSF, o22R, o4R) showed reduced pStat5. This is interesting, especially since the authors mention that Stat5 activation has been shown previously to be beneficial in adoptive T cell transfer therapy. More consideration of this point is warranted.

We thank for reviewer for the thoughtful comments. In our previous study, we demonstrated that STAT5 activation via both mouse and human orthogonal IL-2R is beneficial in adoptive T cell transfer therapy^{1,2}. However, o2R alone showed suboptimal efficacy in B16F10 solid tumors^{1,3}, although it improved antitumor effects in hematological leukemia models (which are generally considered easier models compared to solid tumors)². T cell exhaustion is a major challenge in solid tumors, various cytokines have been explored, since IL-2 alone is insufficient to overcome this issue. As described in the discussion (page 19, lines 8-17),

STAT3 has been reported to play a critical role in T cell stemness and resistance to exhaustion⁹⁻¹³. We observed that both oGCSFR and o22R show higher STAT3 and moderate levels of STAT5, suggesting that this STAT profile may be crucial for optimizing T cell responses in tumors. In contrast, o4R operates through a different mechanism, as it is the only receptor that solely activates STAT6 signaling. This enhanced antitumor effect aligns with recent reports suggesting that STAT6 plays a key role in promoting T cell function and attenuate exhaustion¹⁴. Thus, we cannot conclude that the enhanced antitumor efficacy of o4R is due to reduced STAT5 activation. We specifically discuss the tumor response to o4R in the discussion (page 17, lines 12 - page 18, line 6). Our data primarily provide a comparison of different STAT profiles. While STAT5 also has its benefits, the study may inform the rationale for combining it with other STATs for improved therapeutic outcomes.

4. In various experiments, mixed CD4 and CD8 T cells are used whereas in other experiments just CD8s are used. This is not always clearly stated. If not pools, the enrichment/sorting protocol in the methods and clarification of the subset needs to be more explicit in the figure legend/text. Mechanistically, the potential impact of the orthogonal receptor signals on CD4s and CD8s may confuse readers; clarification is needed.

In the syngeneic model, we used CD8⁺ pmel T cells as described in the manuscript. In the xenograft model, since adoptive T cell therapies use a mixture of CD4⁺ and CD8⁺ T cells in the clinical and preclinical studies^{15,16}, we chose to omit this distinction. To avoid confusion, we have explicitly stated "CD3⁺ T cells" in the relevant figure captions in the revised manuscript for clarity.

5. The rationale behind Figure 4F is unclear. Cell-to-cell interaction analysis would be more informative analyzing cells isolated from the entire tumor, not just the T cells; the therapeutic importance of T cells also relates to their ability to interact with non-T cells.

We agree and this panel is not critical to our conclusions, but nevertheless the rationale behind Fig. 4f is to explore the characteristics of myeloid-like T cells, which is not directly related to therapeutic relevance and more of interest for the concept of T cell reprogramming. We recognize that cell-to-cell interaction analysis is typically performed on cells isolated from the entire tumor, which include interactions between myeloid cells and T cells. To assess whether myeloid-like T cells (C6) induced by oGCSFR exhibit functional similarities to myeloid cells, we conducted this analysis. We found that these cells displayed elevated expression of several myeloid related ligands compared to other clusters (Fig. 4f), suggesting their potential for intercellular communication. This further supports their myeloid-like characteristics. In addition, these data provide insights into the ligand profiles of C6 and its potential interactions with other clusters in tumors.

6. The data provided in Figure 5C is hard interpret. How did the authors decide where to gate? Can they include a negative control for cytokine production? There The plots would be clearer if a comparator was provided (rather than CD4 and CD8) if both axes were a cytokine, e.g. IFN γ vs IL-4.

We gated cytokine production based on unstimulated controls, as shown below. Due to space limitations, we have included this figure in the New Supplementary Figure 1. Gating Strategy. As suggested, we have updated both axes to display cytokine comparisons (IL-4 vs. IFN γ , IL-5 vs. IFN γ , IL-13 vs. IFN γ) in the New Fig. 5c.

New Supplementary Figure 1. Gating Strategy. d. Gating Strategy for cytokine profiles of ho4R NY-ESO-1 TCR-T cells in Fig. 5c–g, Extended Data Fig. 6b.

7. Also unclear is why pSTAT3 data are not provided with IFN stimulation. Including the primary data would be useful with appropriate controls (e.g. Ext Data Fig 1).

The IFN receptor family typically does not activate STAT3. As suggested, we have included this data in New Fig. 1e and New Extended Data Fig. 1d for clarity.

New Fig. 1e. Heatmap of relative pSTAT-1, -3, and -5 MFI in WT T cells transduced with o2R, oIFNAR2, oIFNGR1, and oIFNLR1, treated with MSA-oIL2 (10 μM), normalized to YFP⁻ controls. All data represent mean ± s.e.m

New Extended Data Fig. 1d. pSTAT-1, -3, and -5 signaling dose-response curves of orthogonal IL-2Rβ ECD-ICD from oIFNAR2, oIFNGR1, and oIFNLR1 transduced T cells. All data represent mean ± s.e.m.

Referee #4 (Remarks to the Author):

In this manuscript, the author designed a series of chimeric cytokine receptor based on their previously described orthogonal IL2-IL2RB system to create multiple unnatural /JAK1/2/3 pairing, resulting in unique downstream signaling events that are not native to T cells. This system enables them to activate specific cytokine signaling using an orthogonal IL2 ligand where they showed that integrating orthogonal IL4, IL22 and GCSF chimeric receptor could greatly improve anti-tumor activity of pmel T cells against B16 melanoma. Using scRNA-Seq, they further studied different unique state/fate of T cells when o4R, o22R or oGSCFR are expressed. Interestingly, they found that oGSCFR could endow T cells with myeloid signature and potentially phagocytic activity. Integrating IL4 signaling foster a type 2 state of T cells that is associated with stronger and durable anti-tumor activity in vivo. Lastly, T cells expressing o22R appeared to possess more stem-like features in vivo, leading to best tumor control and long-term persistence of T cells in vivo.

I believe this is a novel and well-designed study with great translational potential (especially with o22R). However, I think there are quite a lot of weaknesses in this manuscript that need to be addressed. Specifically, I have concerns with the reported phagocytic activity of T cells (which I believe is one of the biggest highlights, if true) and with the mechanism of stemness endowed by o22R, which is the key translational point.

We thank the reviewer for the positive comments and thoughtful assessment. We have thoroughly addressed the points as shown below in detail.

Major comments:

1. Conceptually, I wonder how this strategy compared to other semi-orthogonal cytokine signaling strategy (such as the one reported by Naoto Hirano group, Nat. Med 2018 “A novel chimeric antigen receptor containing a JAK–STAT signaling domain mediates superior antitumor effects”) where ICD of cytokine receptor is directly coupled to the CAR molecule. For instance, if I add ICD of IL22R or IL4R to the endodomain of a CAR, would that have similar (better or worse) activity in terms of anti-tumor effects and T cell expansion as what’s reported here?

Cytokine receptors are structurally optimized for rapid and robust JAK-STAT activation upon ligand-induced dimerization and transactivation of the JAKs⁷. In contrast, the CAR-T induced JAK-STAT signaling observed in Hirano et al. occurs by a mechanism that is not clear (i.e. how is STAT3 being phosphorylated?). The STAT3 phosphorylation by the CAR was delayed and inefficient, with 28z-ΔIL2RB-z(YXXQ) inducing only moderate STAT3 activation between 120 and 240 minutes¹⁷. Notably, this pSTAT3 activation included basal signaling from CD28z, which can by itself induce STAT3 activation.

In response to the reviewers' suggestion, we attempted to incorporate the ICD of IL-22R into the endodomain of a CAR. IL-22R mediates non-canonical, tyrosine-independent STAT3 activation, where STAT3 can pre-associate with IL-22R α through a mechanism independent of receptor phosphorylation by binding to the receptor's C-terminal domain (CTD)¹⁸. We fused the CTD of IL-22R¹⁹ to the C-terminus of a murine CD19 CAR (1D3 clone) with CD28z (CD19_28z_22R CTD) and compared its pSTAT3 signaling in T cells to that of the mCD19-CAR containing the YRHQ mutation (CD19_28z_YRHQ) described by Hirano et al (**RL-only Fig. VI. a**).

CD19_28z_22R CTD supported pSTAT3 signaling at levels comparable to or slightly higher than the Hirano YRHQ format but exhibited similarly delayed kinetics, when compared to IL-21-mediated pSTAT3 at 30 minutes (**RL-only Fig. VI. b**). Consistently, control CAR (CD19_28z) also induced some pSTAT3, suggesting that the pSTAT3 signaling in this approach is partially driven by the CAR domain alone. When compared to strong STAT3 signaling induced by each of the orthogonal receptors at 20 minutes post-activation (**Extended Data Fig. 1**), the STAT3 signaling from YRHQ and the 22R CTD was less effective.

Thus, adding an ICD/motif to a CAR terminus is insufficient for achieving robust and timely STAT activation via cytokine receptor-mediated signaling.

RL-only Fig. VI. a. Schematic depicting the CD19_28z, CD19_28z_YRHQ, and CD19_28z_22R CTD; scFv, single-chain variable fragment; TM, transmembrane domain; ICD, intracellular domain, CTD, C-terminal domain. **b.** MFI of phosphorylated STAT3 over time, as assessed by flow cytometry, in CD19 CAR T cells transduced with the indicated CAR-encoding construct (n = 2). CAR T cells were stimulated with CD19⁺ B cells or treated with IL-21 (50 ng/mL) served as a control. All data represent mean ± s.e.m. and were analyzed by one-way ANOVA with Tukey's post-test (**b**).

2. In all animal experiments, a condition where chimeric receptor is expressed but MSA-oIL2 is not added is missing. While the chimeric receptor function requires oIL2 in vitro, in vivo environments where native IL2 and other cytokines are also present would complicate the interpretation of the results. It's important to demonstrate that the observed anti-tumor activity of chimeric receptor is dependent on the orthogonal ligand instead of other possible ligands in vivo.

We have previously demonstrated that orthogonal receptor activation is strictly dependent on ortho IL-2 in several studies, native IL-2 cannot activate the orthogonal receptors containing the chimeric ICDs¹⁻³. In response to the reviewer's suggestion, we have added new in vivo antitumor studies using natural (o4R) and non-natural (oGCSFR) receptors as representatives. Our results indicated that ACT with o4R and oGCSFR pmel CD8⁺ T cells showed slight tumor inhibition (**New Extended Data Fig. 1d,e**), comparable to levels observed with pmel CD8⁺ T cells alone in **Extended Data Fig. 2**. In contrast, with MSA-oIL2 i.p. administration, o4R and oGCSFR pmel CD8⁺ T cells exhibited substantially enhanced antitumor activity

(New Extended Data Fig. 1d,e). Thus, the observed anti-tumor activity of the chimeric receptor is dependent on the orthogonal ligand rather than other potential ligands *in vivo*.

New Extended Data Fig. 2d. The experimental setting is described in Fig. 2a ($n = 5$). Shown are average tumor growth curves. All data represent mean \pm s.e.m. and are analysed by two-way ANOVA with Tukey's post-test (**d**).

3. In ext.figure 4h,i, the authors showed an increased number of granzyme B and IFNG/TNF positive T cells within tumors of oIL4R, oIL22R etc.. I wonder whether the increased density of these effector population is due to increased expression per cell and/or to the increase in total T cell numbers (as shown in ext.fig 3b). Showing FACS plot of granzyme B/IFNG/TNF of YFP⁺ TILs would be more informative.

The increased density of these effector populations is due to both increased expression per cell (**RL-only Fig. VII**) and an increase in total T cell numbers (Extended Data, Fig. 3b). As suggested, we have added a FACS plot of IFN γ ⁺TNF α ⁺ and granzyme B⁺ in YFP⁺ TILs, shown in the New Extended Data Fig. 4h.

RL-only Fig. VII. The experimental setting is described in Fig. 3a. **a**, Frequencies of IFN γ ⁺TNF α ⁺ among Thy1.1⁺CD8⁺ T cells in tumors ($n = 5$ animals). **b**, Frequencies of granzyme B⁺ among Thy1.1⁺CD8⁺ T cells in tumors ($n = 5$ animals). All data represent mean \pm s.e.m. and were analyzed by one-way ANOVA with Tukey's post-test (**a,b**).

New Extended Data Fig. 4h. Representative flow cytometry plots showing IFN γ ⁺TNF α ⁺ (top) and Granzyme B⁺ (bottom) subpopulations among Thy1.1⁺CD8⁺ pmel T cells in tumors.

4. While C/EBP β is a putative target gene of MafB, are C/EBP β target genes in fig. 4b upregulated in c6 or in oGCSFR T cells? If not, then I doubt C/EBP β is the TF driving the phenotype in c6 or in oGCSFR T cells, then singling out C/EBP β is irrelevant.

C/EBP β target genes in **Fig. 4b**, including *Cd14* and *Plaur*, were upregulated in C6 cells (**Fig. 4c,d**). In flow cytometry, CD14 was also detected in oGCSFR T cells (**Fig. 4g**). Since C/EBP β is a putative target gene of MafB, we hypothesize that oGCSFR may induce these myeloid markers via the MafB–C/EBP β –targeted genes pathway. While this is a plausible mechanism, we present this data here as C/EBP β has been studied in cell transdifferentiation studies^{20,21}, which could provide a reference for this observation.

5. In fig. 4g-k authors showed that OE of Nfic, Maf, MafB is not sufficient to phenocopy oGCSFR and postulate a combination of 2-3 TFs necessary to drive the expression of myeloid genes in T cells. But it is equally likely that other TFs could be involved here. Did you try to OE multiple TFs and measure the phenotype?

We apologize for the imprecise statement. Our intention was to describe how multiple TFs, not limited to Maf, MafB, and Nfic, are regulated by oGCSFR and work together to induce a myeloid phenotype. To clarify, we have rephrased the statement as follows: “Indeed, Nfic, Maf, and MafB-overexpressing pmel CD8⁺ T cells partially acquired myeloid features (Fig. 4g–k), albeit less profoundly than oGCSFR+MSA-oIL2, suggesting that the coordinated gene regulation orchestrated by oGCSFR, rather than a single TF, is required for the most prominent T-myeloid trans-differentiation” (**Page 11, lines 10-14**).

To address the reviewer’s question, we overexpressed three TFs (Nfic, Maf, MafB) and assessed the phenotype by flow cytometry. Our results show that the combination of these three TFs is insufficient to induce a myeloid phenotype (**RL-only Fig. VIII**), indicating that other TFs, or even additional factors such as signaling networks and epigenetic regulators, are likely involved in the reprogramming process. Neither a single TF (Nfic/Maf/MafB) nor this trio of TFs can replicate the intricate gene regulation induced by oGCSFR.

RL-only Fig. VIII. NT pmel CD8⁺ T cells and those transduced with Nfic-Maf-MafB, or oGCSFR were cultured for 72h, with oGCSFR pmel CD8⁺ T cells stimulated with MSA-oIL2 (500 nM) during this period, followed by flow cytometry analysis (n = 7). **a**, Representative flow cytometry plots showing the frequencies of Mac-1⁺, CD14⁺, CD64⁺, and Gr-1⁺ cells among pmel CD8⁺ T cells. **b–e**, Shown are the frequencies of Mac-1⁺ (**b**), CD14⁺ (**c**), CD64⁺ (**d**), and Gr-1⁺ (**e**) cells among pmel CD8⁺ T cells.

6. The authors performed several phagocytic assays to demonstrate the oGCSFR T cells possess phagocytic activity. Relying on mere MFI would over-estimate the effect since the fluorescent cells could be adhering to the cell surface instead of being phagocytosed. The only image is fig.4m but the image of control T cells is missing so it's hard to judge whether control T cells are showing similar phenotype. So, it is imperative to show the confocal image of phagocytic cells of all experiments. Also, blocking experiments should be conducted in parallel, at least with the Ab-opsonized A20 cells where the FcRg being the clear phagocytic receptor. Since oGCSFR T cells are showing superior anti-tumor activity, do you think the phagocytic activity is required for this observation?

For some studies, we used flow cytometry based on MFI for analysis. For the Vybran Phagocytosis Assay Kit (Extended Data Fig. 5c,d), the reagent is specifically designed for flow cytometry. Importantly, when assessing MFI, NT and oGCSFR CD8⁺ T cells underwent the same procedure, so we do not believe the observed signal is due to non-specific adherence. In addition, we used live bacteria for further validation. At the highest concentration of GFP⁺ *L. monocytogenes*, we observed significantly enhanced GFP signal in oGCSFR CD8⁺ cells (Fig. 4l). As the reviewer suggests, we added confocal images of co-cultures with GFP⁺ *L. monocytogenes* at the highest concentration. Most GFP⁺ *L. monocytogenes* in NT T cells remained outside the cells, with minimal GFP signaling detected inside, while a proportion of oGCSFR T cells efficiently internalized GFP⁺ *L. monocytogenes* (New Extended Data Fig. 5e). Therefore, combining flow cytometry data and confocal visualization, we confirmed that oGCSFR CD8⁺ T cells possess enhanced capacity to internalize bacteria.

For the antibody-dependent cellular phagocytosis (ADCP) assay in **Fig. 4n,o**, we followed the published protocol and gating strategy (**Supplementary Fig. 1b**) specifically designed for flow cytometry²². In response to the reviewers' request, we also performed confocal imaging for this experiment, along with Fc receptor blocking. We observed that, unlike RAW 264.7 macrophages, which internalize tumor fragments via phagocytosis more efficiently, the increased FarRed⁺ cells in oGCSFR T cells result from capturing tumor membrane fragments on the surface and internalizing tumor-derived components (**New Extended Data Fig. 5f**). In contrast, NT cells exhibited no tumor fragments on the surface or internalized. This process is mediated by Fc receptors (**New Extended Data Fig. 5g,h**). We thank the reviewer for the suggestion and have clarified this point in the revised manuscript (**Page 12, lines 3-6**).

Fc receptor-induced antitumor activity by oGCSFR T cells requires tumor-targeted antibodies. To investigate this, we co-cultured mouse CD19-expressing B16F10 cells with NT and oGCSFR pmel CD8⁺ T cells, in the presence or absence of mCD19 antibodies. We found that oGCSFR enhanced antitumor killing compared to NT controls under both conditions (**RL-only Fig. III**). In the presence of CD19 antibodies, oGCSFR increased tumor killing from ~57.8% to ~70.8%, while NT controls showed no significant changes. In the *in vivo* scenario, where endogenous antibodies tag B16F10 cells, we expect the anti-tumor effect to be outweighed by direct T cell cytotoxicity. However, oGCSFR adoptive T cell therapy may synergize with tumor-targeting antibody-based therapies.

New Extended Data Fig. 5e. The experimental setup is described in Fig. 4l. Shown are the representative fluorescence images of GFP⁺ *L. monocytogenes* at the highest concentration internalized by NT or oGCSFR CD8⁺ T cells.

New Extended Data Fig. 5f. The experimental setting is described in Fig. 4n,o. Representative fluorescence images of CSFE⁺ (green) effector cells (NT, oGCSFR CD8⁺ T cells, or RAW264.7) cocultured with FarRed⁺ (red) A20 target cells at a 1:1 E:T ratio.

New Extended Data Fig. 5g,h. NT or oGCSFR-transduced pmel CD8⁺ T cells (\pm Fc receptor blocking antibody cocktail) were cocultured for 2 hours with FarRed⁺ A20 cells pretreated with anti-mouse CD19 antibody, followed by flow cytometry analysis (n = 6). Shown are the Representative flow cytometry plots (**g**) and average frequencies (**h**) of FarRed⁺CD8⁺ cells. All data represent mean \pm s.e.m. and are analysed by two-tailed Student's t-test (h).

RL-only Fig. III. oGCSFR pmel CD8⁺ T cells were preserved in MSA-hoIL2 (500 nM) for 24 hours, while NT pmel CD8⁺ cells were cultured in MSA-IL-2 (100 U/mL). mouse CD19-expressing B16F10 tumor cells (B16F10-mCD19) were co-cultured with NT and oGCSFR pmel CD8⁺ T cells at a 1:1 E:T ratio with MSA-hoIL2 (200 nM), in the presence or absence of mCD19 antibodies (2 μg/mL) for 48 hours. Cells were collected for flow cytometry analysis. Shown are the percentages of B16F10-mCD19 cell killing. All data represent mean ± s.e.m. and were analyzed by one-way ANOVA with Tukey's post-test.

7. The experiment/timepoint where CAR-T cells are collected for ATAC-Seq to study stemness may not be optimal since there is already big difference in tumor burden so it's likely that CAR-T cells in different groups are showing different activation state due to the amount of tumor cells. I believe the presented data provide very little insight to the mechanism of stemness in o22R or oGSCFR T cells. The observed opening of AP-1 region may explain improved functionality but it should also push T cells towards effector state instead of stem-like state. BACH2 could be involved but I would need to see more evidence (e.g. KD BACH2 or demonstrate BACH2 downstream genes related to stemness being upregulated) to be convinced.

Our study provided multiple lines of evidence that o22R promotes T cell stemness under chronic tumor antigen stimulation (Fig. 3g-i, 3j, 6h,i, and Extended Data Fig. 9f). These findings suggest that o22R sustains stemness within tumors, which is critical for its antitumor function. To explore this further, we collected TCR-T cells in tumors for ATAC-seq after six doses of orthogonal IL-2 injection. While there is no single optimal time point that balances comparable tumor burden, sufficient TILs expansion for analysis, and adequate exposure to orthogonal IL-2 for stem-like properties to manifest, a 1-2 week post-treatment window is commonly used in similar studies^{14,23,24} and should allow for robust profiling of tumor-infiltrating T cells.

We agree that the presented data provide limited mechanistic insight into the stemness of o22R/oGSCFR T cells. However, as a synthetic biology and reprogramming study, it is beyond the scope of this paper to fully delineate the mechanisms for all receptors examined. While the AP-1 region contributes to enhanced functionality by driving T cells toward an effector state, effective tumor clearance requires engagement of effector programs. We propose that ho22R T cells upregulate BACH2, which may counterbalance AP-1, reinforcing transcriptional and epigenetic programs that sustain a stem-like state while preventing terminal differentiation. As suggested, we disrupted BACH2 expression and found that its knockout in ho22R T cells resulted in a loss of stemness characteristics (CD62L, TCF-1, and partial FOXO1) and a shift toward a CD44^{hi} effector phenotype upon TCR stimulation (New Extended Data Fig. 10i-m).

New Extended Data Fig. 5i–m, NT, ho22R, and ho22R (BACH2 knockout) CD3⁺ human T cells were restimulated with soluble CD3 antibodies at 1 μ g/mL in the presence of MSA-hoIL2 (200 nM) for 48 hours (n = 4). Cells were then collected for quantitative PCR and flow cytometry analysis. **i**, Relative expression of BACH2 (n = 4). **j–m**, MFI of CD62L (**j**), CD44 (**k**), TCF-1 (**l**), FOXO1 (**m**). All data represent mean \pm s.e.m. and are analysed by one-way ANOVA with Tukey’s post-test (**j–m**).

Minor comments:

1. In extended figure 1a, o10R-YFP plot appears to have abnormal negative cell populations, please ensure biexponential expression setting is consistent across all plots.

We have updated the o10R-YFP histogram in New Extended Data Fig. 1a, to ensure consistent biexponential scaling across all plots.

2. Also, I believe it would be more informative to show the chimeric receptor expression instead of YFP expression since IRES-YFP expression only indicates presence of transgene mRNA but missing the information of protein expression. This is important since chimerism may affect protein stability.

We agree that receptor expression is more informative than YFP. However, our data suggest that the orthogonal receptor is stable *in vivo*. With MSA-oIL2 treatment, we observed a sustained response throughout the therapy window (until day 14 post-ACT). If the receptor were unstable, we would expect a diminished response over time.

3. The notion that CD95 is a stemness marker is inaccurate, CD95 itself does not indicate stemness but the co-expression of CD95 and CD45RA/CCR7 (in human) distinguish TSCM from naïve T cells. Hence, I don’t think CD95 itself is a stemness marker. However, I agree with the overall message that o22R pmel T cells have higher percentage of stem-like T cells. This comment also applies to ext. fig. 6e/7f, I think it’s best to show a representative FACS plot of CD45RA/CD62L/CCR7-CD95 (as in other studies where they define TSCM population in human CAR-T cells).

We agree that many stemness markers are shared between memory T cells and naïve T cells. Using a combination of markers for gating can provide a more precise definition of T_{SCM}. As suggested, we have included a representative FACS plot of CD45RA/CD62L/CCR7/CD95 and updated the analysis to show frequencies of CD45RA⁺CD62L⁺CCR7⁺CD95⁺ cells in New Extended Data Fig. 7f,g and New Extended Data Fig. 9f.

New Extended Data Fig. 7f,g. **f**, Representative flow cytometry plots showing the gating strategy for CD45RA⁺CD62L⁺CCR7⁺CD95⁺ subpopulations in NT, hoGCSFR, and ho22R CD3⁺ NY-ESO-1 TCR-T cells. **g**, Frequencies of CD45RA⁺CD62L⁺CCR7⁺CD95⁺ cells.

New Extended Data Fig. 7f. Frequencies of CD45RA⁺CD62L⁺CCR7⁺CD95⁺ among CD3⁺ NY-ESO-1 TCR-T cells (n = 3).

4.Ext.figure 5a, the y-axis label “proportion of NA” is quite misleading, maybe “proportion of unmapped” is better?

We thank the reviewer for pointing that out. We have updated the y-axis label to 'proportion of unmapped' in New Extended Data Fig. 5a.

5.Did you evaluate the effectiveness of o4R on CAR-T cells with other tumor models? Do you think the anti-tumor activity of o4R depends on type 2 cytokines or a specific state of T cells that is associated with this type of T cells? For instance, if you can reduce (but not eliminate) GATA3 expression to match what’s in control, would that abolish the anti-tumor activity of o4R T cells? Or if we KO IL4/5/13, what would happen? Maybe it’s not entirely within the scope of this study but I would like the authors to add in some discussion at least.

We thank the reviewer for raising this point. We evaluated the effectiveness of o4R in pmel T cells in the B16F10 tumor model and in NY-ESO-1 TCR-T cells in the A375 tumor model, but not yet in a CAR-T model. Regarding whether the anti-tumor activity of o4R is dependent on type 2 cytokines, we propose two potential mechanisms: 1) Secreted type 2 cytokines may act on o4R-induced Th2/Tc2 cells, which express higher levels of the receptors compared to other T cells, forming a positive feedback loop. 2) We observed that o4R shows more potent tumor inhibition in syngeneic tumor model (**Fig. 2b,c and Extended Data Fig. 2c**) compared to xenograft model (**Fig. 5i,j**), suggesting that type 2 cytokines may also act on endogenous immune populations (e.g., eosinophils, macrophages) in a paracrine manner to boost the anti-tumor response.

In the in vitro experiments, blocking IL-4, IL-5, IL-13, or their combination with antibodies did not diminish the anti-tumor efficacy of o4R TCR-T cells, and in fact, slightly enhanced it (**New Extended Data Fig. 6f,g**). While tumor cells may express these cytokine receptors, which could introduce complexity, the results generally indicate that secreted cytokines are not the primary drivers of o4R's anti-tumor effect. However, upon disrupting GATA3 in o4R TCR-T cells, we observed a significant reduction in anti-tumor activity, suggesting the essential role of GATA3 in enhancing tumor killing (**New Extended Data Fig. 6f,g**). This suggests that GATA3's function extends beyond regulating Th2 cytokines, likely influencing T cell effector programs as well. As shown in our scRNA-seq data, the Tc2 cluster also exhibits higher expression of *Gzma* and *Gzmb* (**Extended Data Fig. 4e,f**), these cytotoxic molecules may play a role in Tc2 cell-mediated tumor killing and could be regulated by GATA3²⁵⁻²⁷. As suggested, we have added a discussion in the revised manuscript (**Page 17, Line 23 – 28**).

New Extended Data Fig. 6f,g. **f**, Representative flow cytometry histograms showing GATA-3 expression level. **g**, NT or ho4R NY-ESO-1 TCR-T cells (unmodified, GATA3 knockout, or treated with the indicated antibodies) were co-cultured with A375 cells at an E:T ratio of 1:2 for 48 hours (n = 3). Shown are the counts of viable A375 tumor cells. All data represent mean \pm s.e.m. and are analysed by one-way ANOVA with Tukey's post-test (**g**).

6. In fig.6, the authors only showed T cell counts and functional phenotype in A375 model, what about the other two models?

We also showed T cell counts in the NALM6-Luc leukemia model. In peripheral blood, we observed a 72.6-fold increase in T cell engraftment in ho22R CAR T-treated mice compared to PBS controls at day 6 post-ACT (**Extended Data Fig. 8b**). In addition, we have now included T cell profiling data in the Raji lymphoma model. On day 26 post-tumor inoculation, ho22R CAR T cells demonstrated enhanced CD19 CAR T cell enrichment within tumors and a reduction in exhaustion markers, as evidenced by lower PD-1 and LAG-3 expression. Notably, ho22R also preserved IFN γ and granzyme B secretion (**New Extended Data Fig. 8f-j**).

New Extended Data Fig. 8f–j. NSG female mice were inoculated (s.c.) with Raji cells (2×10^6) and received an i.v. adoptive transfer of 1×10^6 NT, ho22R, or hoGCSFR CD3⁺ CD19 CAR T cells (with Myc tag) on Day 11, followed by i.p. administration of MSA-hoIL2 (2.5×10^4 unit) or PBS every other day until day 25 ($n = 6$ animals). On day 26, the mice were sacrificed, and tumors were collected for flow cytometry analysis. **f**, The experimental timeline. **g**, Average tumor growth curves. **h**, Counts of CD3⁺Myc⁺ CD19 CAR T cells in tumors. **i**, MFI of PD-1 (left) and LAG-3 (right) among CD3⁺Myc⁺ CD19 CAR T cells in tumors. **j**, MFI of Granzyme B (left) and IFN γ (right) among CD3⁺Myc⁺ CD19 CAR T cells in tumors. All data represent mean \pm s.e.m. and are analyzed by one-way ANOVA with Tukey's post-test (**h–j**). Schematics in **f** created using BioRender.com.

References

- 1 Sockolovsky, J. T. *et al.* Selective targeting of engineered T cells using orthogonal IL-2 cytokine-receptor complexes. *Science* 359, 1037-1042 (2018). <https://doi.org:10.1126/science.aar3246>
- 2 Zhang, Q. *et al.* A human orthogonal IL-2 and IL-2R β system enhances CAR T cell expansion and antitumor activity in a murine model of leukemia. *Science Translational Medicine* 13, eabg6986 <https://doi.org:10.1126/scitranslmed.abg6986>
- 3 Kalbasi, A. *et al.* Potentiating adoptive cell therapy using synthetic IL-9 receptors. *Nature* 607, 360-365 (2022). <https://doi.org:10.1038/s41586-022-04801-2>
- 4 Hirai, T. *et al.* Selective expansion of regulatory T cells using an orthogonal IL-2/IL-2 receptor system facilitates transplantation tolerance. *J Clin Invest* 131 (2021). <https://doi.org:10.1172/jci139991>
- 5 Beltra, J. C. *et al.* Stat5 opposes the transcription factor Tox and rewires exhausted CD8(+) T cells toward durable effector-like states during chronic antigen exposure. *Immunity* 56, 2699-2718.e2611 (2023). <https://doi.org:10.1016/j.immuni.2023.11.005>
- 6 Wang, X., Rickert, M. & Garcia, K. C. Structure of the quaternary complex of interleukin-2 with its alpha, beta, and gammac receptors. *Science* 310, 1159-1163 (2005). <https://doi.org:10.1126/science.1117893>
- 7 Caveney, N. A. *et al.* Structural basis of Janus kinase trans-activation. *Cell Rep* 42, 112201 (2023). <https://doi.org:10.1016/j.celrep.2023.112201>
- 8 Mitra, S. *et al.* Interleukin-2 Activity Can Be Fine Tuned with Engineered Receptor Signaling Clamps. *Immunity* 42, 826-838 (2015). <https://doi.org:https://doi.org/10.1016/j.immuni.2015.04.018>
- 9 Zhao, Y. *et al.* IL-10-expressing CAR T cells resist dysfunction and mediate durable clearance of solid tumors and metastases. *Nature Biotechnology* (2024). <https://doi.org:10.1038/s41587-023-02060-8>
- 10 Sun, Q. *et al.* STAT3 regulates CD8+ T cell differentiation and functions in cancer and acute infection. *Journal of Experimental Medicine* 220, e20220686 (2023). <https://doi.org:10.1084/jem.20220686>
- 11 Guo, Y. *et al.* Metabolic reprogramming of terminally exhausted CD8(+) T cells by IL-10 enhances anti-tumor immunity. *Nat Immunol* 22, 746-756 (2021). <https://doi.org:10.1038/s41590-021-00940-2>
- 12 Cui, W., Liu, Y., Weinstein, J. S., Craft, J. & Kaech, S. M. An interleukin-21-interleukin-10-STAT3 pathway is critical for functional maturation of memory CD8+ T cells. *Immunity* 35, 792-805 (2011). <https://doi.org:10.1016/j.immuni.2011.09.017>
- 13 Laidlaw, B. J. *et al.* Production of IL-10 by CD4+ regulatory T cells during the resolution of infection promotes the maturation of memory CD8+ T cells. *Nature Immunology* 16, 871-879 (2015). <https://doi.org:10.1038/ni.3224>
- 14 Feng, B. *et al.* The type 2 cytokine Fc-IL-4 revitalizes exhausted CD8+ T cells against cancer. *Nature* 634, 712-720 (2024). <https://doi.org:10.1038/s41586-024-07962-4>
- 15 Neelapu Sattva, S. *et al.* Axicabtagene Ciloleuce CAR T-Cell Therapy in Refractory Large B-Cell Lymphoma. *New England Journal of Medicine* 377, 2531-2544 <https://doi.org:10.1056/NEJMoa1707447>
- 16 Lynn, R. C. *et al.* c-Jun overexpression in CAR T cells induces exhaustion resistance. *Nature* 576, 293-300 (2019). <https://doi.org:10.1038/s41586-019-1805-z>
- 17 Kagoya, Y. *et al.* A novel chimeric antigen receptor containing a JAK-STAT signaling domain mediates superior antitumor effects. *Nature Medicine* 24, 352-359 (2018). <https://doi.org:10.1038/nm.4478>
- 18 Saxton, R. A. *et al.* The tissue protective functions of interleukin-22 can be decoupled from pro-inflammatory actions through structure-based design. *Immunity* 54, 660-672.e669 (2021). <https://doi.org:https://doi.org/10.1016/j.immuni.2021.03.008>

- 19 Dumoutier, L., de Meester, C., Tavernier, J. & Renauld, J.-C. New Activation Modus of STAT3: A TYROSINE-LESS REGION OF THE INTERLEUKIN-22 RECEPTOR RECRUITS STAT3 BY INTERACTING WITH ITS COILED-COIL DOMAIN*. *Journal of Biological Chemistry* 284, 26377-26384 (2009). <https://doi.org/10.1074/jbc.M109.007955>
- 20 Laiosa, C. V., Stadtfeld, M., Xie, H., de Andres-Aguayo, L. & Graf, T. Reprogramming of Committed T Cell Progenitors to Macrophages and Dendritic Cells by C/EBP α and PU.1 Transcription Factors. *Immunity* 25, 731-744 (2006). <https://doi.org/10.1016/j.immuni.2006.09.011>
- 21 Xie, H., Ye, M., Feng, R. & Graf, T. Stepwise Reprogramming of B Cells into Macrophages. *Cell* 117, 663-676 (2004). [https://doi.org/10.1016/S0092-8674\(04\)00419-2](https://doi.org/10.1016/S0092-8674(04)00419-2)
- 22 Stanganello, E. *et al.* in *Methods in Cell Biology* Vol. 173 (eds Clément Thomas & Lorenzo Galluzzi) 109-120 (Academic Press, 2023).
- 23 Zhao, Y. *et al.* IL-10-expressing CAR T cells resist dysfunction and mediate durable clearance of solid tumors and metastases. *Nature Biotechnology* 42, 1693-1704 (2024). <https://doi.org/10.1038/s41587-023-02060-8>
- 24 Miller, B. C. *et al.* Subsets of exhausted CD8⁺ T cells differentially mediate tumor control and respond to checkpoint blockade. *Nature Immunology* 20, 326-336 (2019). <https://doi.org/10.1038/s41590-019-0312-6>
- 25 O'Hara, J. *et al.* Notch dependent chromatin remodeling enables Gata3 binding and drives lineage specific CD8(+) T cell function. *Immunol Cell Biol* (2025). <https://doi.org/10.1111/imcb.70002>
- 26 Nguyen, M. L. T. *et al.* Dynamic regulation of permissive histone modifications and GATA3 binding underpin acquisition of granzyme A expression by virus-specific CD8⁺ T cells. *European Journal of Immunology* 46, 307-318 (2016). <https://doi.org/10.1002/eji.201545875>
- 27 Wan, Y. Y. GATA3: a master of many trades in immune regulation. *Trends in Immunology* 35, 233-242 (2014). <https://doi.org/10.1016/j.it.2014.04.002>